# Federated Linear Contextual Bandits

**Ruiquan Huang**
The Pennsylvania State University
rzh5514@psu.edu

**Weiqiang Wu**
Facebook
weiqiang.wwu@gmail.com

**Jing Yang**
The Pennsylvania State University
yangjing@psu.edu

**Cong Shen**
University of Virginia
cong@virginia.edu

## Abstract

This paper presents a novel federated linear contextual bandits model, where individual clients face different $K$-armed stochastic bandits coupled through common global parameters. By leveraging the geometric structure of the linear rewards, a collaborative algorithm called `Fed-PE` is proposed to cope with the heterogeneity across clients without exchanging local feature vectors or raw data. `Fed-PE` relies on a novel multi-client G-optimal design, and achieves near-optimal regrets for both disjoint and shared parameter cases with logarithmic communication costs. In addition, a new concept called collinearly-dependent policies is introduced, based on which a tight minimax regret lower bound for the disjoint parameter case is derived. Experiments demonstrate the effectiveness of the proposed algorithms on both synthetic and real-world datasets.

## 1 Introduction

Federated learning (FL) (McMahan et al., 2017) is an emerging distributed machine learning (ML) paradigm where massive number of clients collaboratively learn a shared prediction model while keeping all the training data on local devices. Compared with standard centralized machine learning, FL has the following characteristics (Kairouz et al., 2021):

- **Heterogeneous local datasets.** The local datasets, which are often generated at edge devices, are likely drawn from non-independent and identically distributed (non-IID) distributions.
- **Communication efficiency.** The communication cost scales with the number of clients, which is one of the primary bottlenecks of FL. It is critical to minimize the communication cost while maintaining the learning accuracy.
- **Privacy.** FL protects local data privacy by only sharing model updates instead of the raw data.

While the main focus of the state-of-the-art FL is on the supervised learning setting, recently, a few researchers begin to extend FL to the multi-armed bandits (MAB) framework (Lai and Robbins, 1985; Auer et al., 2002; Bubeck and Cesa-Bianchi, 2012; Agrawal and Goyal, 2012, 2013a). In the canonical setting of MAB, a player chooses to play one arm from a set of arms at each time slot. An arm, if played, will offer a reward that is drawn from its distribution which is unknown to the player. With all previous observations, the player needs to decide which arm to pull each time in order to maximize the cumulative reward. MAB thus represents an online learning model that naturally captures the intrinsic exploration-exploitation tradeoff in many sequential decision-making problems.

Extending FL to the MAB framework is naturally motivated by a corpus of applications, such as recommender systems, clinical trials, and cognitive radio. In those applications, the sequential decision making involves multiple clients and is distributed by nature. While classical MAB models assume immediate access to the sequentially generated data at the learning agent, under the new

realm of FL, local datasets can be stored and analyzed at the clients, thus reducing the communication load and potentially protecting the data privacy.

Despite the potential benefits of FL, the sequential decision making and bandit feedback bring new challenges to the design of FL algorithms in the MAB setting. Different from the supervised learning setting where static datasets are collected beforehand, under the MAB setting, data is generated sequentially as decisions are made, actions are taken, and observations are collected. In order to maximize the cumulative reward and minimize the corresponding learning regret, it thus requires sophisticated coordination of the actions of the clients. The heterogeneous reward distributions across clients make the coordination process even more convoluted and challenging. Besides, the data privacy and communication efficiency requirements result in significant challenges for efficient information exchange and aggregation between local clients and the central server.

In this work, we attempt to address those challenges in a federated linear contextual bandits framework. This particular problem is motivated by the following exemplary applications.

- **Personalized content recommendation.** For content (arm) recommendation in web-services, user engagement (reward) depends on the profile of a user (context). The central server may deploy a recommender system on each user' local device (client) in order to personalize recommendations without knowing the personal profile or behavior of the user.

- **Personalized online education.** In order to maximize students performances (reward) in online learning, the education platform (central server) needs to personalize teaching methods (arms) based on the characteristics of individual students (context). With the online learning software installed at local devices (client), it is desirable to personalize the learning experiences without allowing the platform to access students' characteristics or scores.

In those examples, the reward of pulling the same arm at different clients follows different distributions dependent on the context as in contextual bandits (Auer, 2003; Langford and Zhang, 2008). We note that conventional contextual bandits is defined with respect to a single player, where the time-varying context can be interpreted as different incoming user profiles. In contrast, we consider a multi-client model, where each client is associated with a fixed user profile. The variation of contexts is captured over clients as opposed to over time. Although the set of clients remains fixed through the learning process, the reward of pulling the same arm still varies across clients. Such a model naturally takes data heterogeneity into consideration. Besides, we adopt a linear reward model, which has been widely studied in contextual bandits (Li et al., 2010; Agrawal and Goyal, 2013b).

**Main contributions.** Our main contributions are summarized as follows.

First, we propose a new federated linear contextual bandits model that takes the diverse user preferences and data heterogeneity into consideration. Such a model naturally bridges local stochastic bandits with linear contextual bandits, and is well poised to capture the tradeoffs between communication efficiency and learning performances in the federated bandits setting.

Second, we design a novel algorithm named `Fed-PE` and further develop its variants to solve the federated linear contextual bandits problem. Under `Fed-PE`, clients only upload their local estimates of the global parameters without sharing their local feature vectors or raw observations. It not only keeps the personal information private, but also reduces the upload cost. We explicitly show that `Fed-PE` and its variants achieve near-optimal regret performances for both disjoint and shared parameter cases with logarithmic communication costs.

Third, we generalize the G-optimal design from the single-player setting (Lattimore and Szepesvári, 2020) to the multi-client setting. We develop a block coordinate ascent algorithm to solve the generalized G-optimal design efficiently with convergence guarantees. Such a multi-client G-optimal design plays a vital role in `Fed-PE`, and may find broad applications in related multi-agent setups.

Finally, we introduce a novel concept called *collinearly-dependent policy* and show that the celebrated LinUCB type of policies (Li et al., 2010), Thompson sampling based policies with Gaussian priors (Agrawal and Goyal, 2013b), and least squared estimation based policies, such as `Fed-PE`, are all in this category. By utilizing the property of collinearly-dependent policies, we are able to characterize a tight minimax regret lower bound in the disjoint parameter setting. We believe that this concept may be of independent interest for the study of bandits with linear rewards.

Table 1: Performance comparison

| Model | Algorithm | Regret | Communication cost |
|---|---|---|---|
| Linear | DELB | $O(d\sqrt{MT\log(T)})$ | $O((Md + d\log\log d)\log T)$ |
| Linear contextual (shared parameter) | FedUCB[1] | $O(\sqrt{dMT}\log T)$ | $O(Md^2\log T)$ |
| | Fed-PE (this work) | $O(\sqrt{dMT\log(KMT)})$ | $O(M(d^2 + dK)\log T)$ |
| | Lower bound | $\Omega(\sqrt{dMT})$ | N/A |
| Linear contextual (disjoint parameter) | Centralized[2] | $O(\sqrt{dKMT\log^3(KMT)})$ | $O(Md^2KT)$ |
| | Fed-PE (this work) | $O(\sqrt{dKMT\log(KMT)})$ | $O(Md^2K\log T)$ |
| | Lower bound (this work) | $\Omega(\sqrt{dKMT})$ | N/A |

$M$: number of clients; $K$: number of arms; $T$: time horizon; $d$: ambient dimension of the feature vectors.

**Notations.** Throughout this paper, we use $\|x\|_V$ to denote $\sqrt{x^{\mathsf{T}}Vx}$. The *range* of a matrix $A$, denoted by $\mathrm{range}(A)$, is the subspace spanned by the column vectors of $A$. We use $A^{\dagger}$ and $\mathrm{Det}(A)$ to denote the pseudo-inverse and pseudo-determinant of square matrix $A$, respectively. The specific definitions can be found in Appendix B of the supplementary material.

## 2 Related Works

**Collaborative and distributed bandits.** Our model is closely related to the collaborative and distributed bandits when action collision is not considered. Landgren et al. (2016, 2018) and Martínez-Rubio et al. (2019) study distributed bandits in which multiple agents face the same MAB instance, and the agents collaboratively share their estimates over a fixed communication graph in order to design consensus-based distributed estimation algorithms to estimate the mean of rewards at each arm. Szorenyi et al. (2013) considers a similar setup where in each round an agent is able to communicate with a few random peers. Korda et al. (2016) considers the case where clients in different unknown clusters face independent bandit problems, and every agent can communicate with only one other agent per round. The communication and coordination among the clients in those works are fundamentally different from our work.

Wang et al. (2020) investigates communication-efficient distributed linear bandits, where the agents can communicate with a server by sending and receiving packets. It proposes two algorithms, namely, DELB and DisLinUCB, for fixed and time-varying action sets, respectively. The fixed action set setting is similar to our setup, except that it assumes that all agents face the same bandits model, which does not take data heterogeneity into consideration.

**Federated bandits.** A few recent works have touched upon the concept of federated bandits. With heterogeneous reward distributions at local clients, Shi and Shen (2021) and Shi et al. (2021) investigate efficient client-server communication and coordination protocols for federated MAB without and with personalization, respectively. Agarwal et al. (2020) studies regression-based contextual bandits as an example of the federated residual learning framework, where the reward of a client depends on both a global model and a local model. Li et al. (2020) and Zhu et al. (2021) focus on differential privacy based local data privacy protection in federated bandits. While the linear contextual bandit model considered in Dubey and Pentland (2020) is similar to this work, it focuses on federated differential privacy and proposes a LinUCB-based FedUCB algorithm, which incurs a higher regret compared with our result for the shared parameter case. A regret and communication cost comparison between Fed-PE and other baseline algorithms is provided in Table 1.

## 3 Problem Formulation

**Clients and local bandits model.** We consider a federated linear contextual bandits setting where there are $M$ clients pulling the same set of $K$ items (arms) denoted as $[K] := \{1, 2, \ldots, K\}$. At each time $t$, each client $i \in [M]$ pulls an arm $a_{i,t} \in [K]$ based on locally available information. The incurred reward $y_{i,t}$ is given by $y_{i,t} = r_{i,a_{i,t}} + \eta_{i,t}$, where $\eta_{i,t}$ is a random noise, and $r_{i,a_{i,t}}$ is

---

[1]Results adapted from Dubey and Pentland (2020) by letting the privacy budget $1/\epsilon$ go to 0.

[2]Results adapted from the single-player linear contextual bandits studied in Dimakopoulou et al. (2017) by assuming instantaneous information exchange and sequential decision-making at the central server.

the unknown expected reward by pulling arm $a_{i,t}$. We note that without additional assumptions or interaction among the clients, each local model is a standard single-player stochastic MAB, where classic algorithms such as UCB (Auer and Ortner, 2010) and Thompson sampling (Agrawal and Goyal, 2012) are known to achieve order-optimal regret.

**Linear reward structure with global parameters.** In order to capture the inherent correlation between rewards of pulling the same arm by different clients, we assume $r_{i,a}$ has a linear structure, i.e., $r_{i,a} = x_{i,a}^\mathsf{T} \theta_a$, where $x_{i,a} \in \mathbb{R}^d$ is the feature vector associated with client $i$ and arm $a$, and $\theta_a \in \mathbb{R}^d$ is a fixed but unknown parameter vector for each $a \in [K]$. Here we use $x^\mathsf{T}$ to denote the transpose of vector $x$. The same arm $a$ may have different reward distributions for different clients, due to potentially varying $x_{i,a}$ across clients. Such a linear model naturally captures the heterogeneous data distributions at the clients, yet admits possible collaborations among clients due to the common parameters $\{\theta_a\}_{a\in[K]}$. When $\theta_a$ varies for different arm $a$, it is called the *disjoint parameter* case; when $\theta_a$ is known to be a constant across the arms, it is the *shared parameter* case. We investigate both cases in Sections 4 and 5, respectively.

**Communication model.** We assume there exists a central server in the system, and similar to FL, the clients can communicate with the server periodically with zero latency. Specifically, the clients can send "local model updates" to the central server, which then aggregates and broadcasts the updated "global model" to the clients. (We will specify these components later.) Note that just as in FL, communication is one of the major bottlenecks and the algorithm has to be conscious about its usage. Similar to Wang et al. (2020), we define the communication cost of an algorithm as the number of scalars (integers or real numbers) communicated between server and clients. We also make the assumption that clients and server are fully synchronized (McMahan et al., 2017).

**Data privacy concerns.** Similar to Dubey and Pentland (2020), our contextual bandit problem involves two sets of information that are desirable to be kept private to client $i$: the feature vectors $\{x_{i,a}\}_{a\in[K]}$ and the observed rewards $\{y_{i,t}\}_{t\in[T]}$. Different from the differential privacy mechanism adopted in Dubey and Pentland (2020), in this work, we aim to communicate estimated global model parameters $\{\theta_a\}_a$ between the clients and the server. This is consistent with the FL framework, where only model updates are communicated instead of the raw data.

**Assumption 1** *We make the following assumptions throughout the paper:*

*1)* **Bounded parameters:** *For any $i \in [M]$, $a \in [K]$, we have $\|\theta_a\|_2 \le s$, $0 < \ell \le \|x_{i,a}\|_2 \le L$.*
*2)* **Independent 1-subgaussian noise:** *$\eta_{i,t}$ is a 1-subgaussian noise parameter sampled independently at each time for each client with $\mathbb{E}[\eta_{i,t}] = 0$, $\mathbb{E}[\exp(\lambda \eta_{i,t})] \le \exp(\frac{\lambda^2}{2})$ for any $\lambda > 0$.*

Assumption 1.1 is a standard assumption in the bandit literature, which ensures that the maximum regret at any step is bounded. We emphasize that our work does not make any assumption on the knowledge of suboptimality gaps, nor do we assume the existence of a unique optimal arm at each client.

Our objective is to minimize the expected cumulative regret among all clients, defined as:

$$\mathbb{E}[R(T)] = \mathbb{E}\left[\sum_{i=1}^{M}\sum_{t=1}^{T}\left(x_{i,a_i^*}^\mathsf{T}\theta_{a_i^*} - x_{i,a_{i,t}}^\mathsf{T}\theta_{a_{i,t}}\right)\right], \tag{1}$$

where $a_i^* \in [K]$ is an optimal arm for client $i$: $\forall b \ne a_i^*$, $x_{i,a_i^*}^\mathsf{T}\theta_{a_i^*} - x_{i,b}^\mathsf{T}\theta_b \ge 0$.

## 4 Federated Linear Contextual Bandits: Disjoint Parameter Case

### 4.1 Challenges

Solving the federated linear contextual bandits model faces several new challenges. The first challenge is due to the constraint that only locally estimated parameters $\{\theta_a\}_{a\in[K]}$ are uploaded to the central server. While this is not an issue for stochastic MAB where the $\{\theta_a\}_{a\in[K]}$ are scalars (Shi and Shen, 2021; Shi et al., 2021), this brings significant challenges for the aggregation of the local estimates into a "global model" in our setup. This is because under the linear reward structure, the locally received rewards $\{y_{i,t}\}_{t\in[T]}$ only contain the projection of $\theta_a$ along the direction of $x_{i,a}$, while the

portion of information lying outside $\mathrm{range}(x_{i,a})$ is not captured in $\{y_{i,t}\}_{t\in[T]}$. Thus, by utilizing $\{y_{i,t}\}_{t\in[T]}$, the locally estimated $\theta_a$, denoted as $\hat{\theta}_{i,a}$, cannot provide any information of $\theta_a$ beyond $\mathrm{range}(x_{i,a})$. Since $\{x_{i,a}\}_{i\in[M]}$ are different for the same arm $a$, the locally estimated $\{\hat{\theta}_{i,a}\}_{i\in[M]}$ are essentially lying in different subspaces. The central server thus needs to take such geometric structure into account when aggregating $\{\hat{\theta}_{i,a}\}$ to construct the global estimate of $\theta_a$.

The geometric structure of the local rewards also brings another challenge for the coordination of actions of local clients. Intuitively, in order to help client $i$ accurately estimate the expected reward by pulling arm $a$, it suffices to obtain an accurate projection of $\theta_a$ on $\mathrm{range}(x_{i,a})$; any part of $\theta_a$ lying outside this subspace is irrelevant. Therefore, if two clients $i$ and $j$ have $x_{i,a}$ and $x_{j,a}$ orthogonal to each other, exchanging the local estimates $\hat{\theta}_{i,a}$ and $\hat{\theta}_{j,a}$ does not help the other client improve her own local estimation. On the other hand, if $x_{i,a}$ and $x_{j,a}$ are completely aligned with each other, $\hat{\theta}_{i,a}$ and $\hat{\theta}_{j,a}$ can be aggregated directly to improve the local estimation accuracy of both. With $M$ possible subspaces spanned by $\{x_{i,a}\}_i$, it is highly likely that different clients receive different amounts of *relevant* information (i.e., information lying in $\mathrm{range}(x_{i,a})$) through information exchange facilitated by the central server. Therefore, in order to reduce the overall regret, it is necessary to coordinate the actions of clients in a sophisticated fashion.

Third, since the exact knowledge of the local feature vectors $\{x_{i,a}\}$ are kept from the central server, the server may not have an accurate estimation of the uncertainty level of the local estimates at each client, or how much the coordination would help individual clients. This would make efficient and effective coordination even more challenging.

## 4.2 Federated Phased Elimination (Fed-PE) Algorithm

To address the aforementioned challenges, we propose the Federated Phased Elimination (Fed-PE) algorithm. The Fed-PE algorithm works in phases, where the length of phase $p$ is $f^p + K$. It contains a client side subroutine (Algorithm 1) and a server side subroutine (Algorithm 2). Throughout the paper we use superscript $p$ to indicate phase $p$ barring explicit explanation. We use $\mathcal{A}_i^p \subset [K]$ to denote the subset of active arms at client $i$ in phase $p$, $\mathcal{A}^p := \cup_{i=1}^M \mathcal{A}_i^p$, $\mathcal{R}_a^p := \{i : a \in \mathcal{A}_i^p\}$, and define $\mathcal{T}_{i,a}^p$ as the time indices at which client $i$ pulls arm $a$ during the collaborative exploration step in phase $p$. Then, the algorithm works as follows.

---

**Algorithm 1** Fed-PE : client $i$

---

**Input:** $T$, $M$, $K$, $\alpha$, $f^p$
1: **Initialization:** Pull each arm $a \in [K]$ and receive reward $y_{i,a}$; $\hat{\theta}_{i,a}^0 \leftarrow \frac{y_{i,a}x_{i,a}}{\|x_{i,a}\|^2}$; Send $\{\hat{\theta}_{i,a}^0\}_a$ to the server; $\mathcal{A}_i^0 \leftarrow [K]$; $p \leftarrow 1$.
2: **while** not reaching the time horizon $T$ **do**
3:     Receive $\{(\hat{\theta}_a^p, V_a^p)\}_{a \in \mathcal{A}^{p-1}}$ from the server.           ▷ Arm elimination
4:     **for** $a \in \mathcal{A}_i^{p-1}$ **do**
$$\hat{r}_{i,a}^p \leftarrow x_{i,a}^\mathsf{T}\hat{\theta}_a^p, \qquad u_{i,a}^p \leftarrow \alpha \|x_{i,a}\|_{V_a^p} / \ell. \tag{2}$$
5:     **end for**
6:     $\hat{a}_i^p \leftarrow \arg\max_{a \in \mathcal{A}_i^{p-1}} \hat{r}_{i,a}^p$,   $\mathcal{A}_i^p \leftarrow \left\{a \in \mathcal{A}_i^{p-1} \mid \hat{r}_{i,a}^p + u_{i,a}^p \geq \hat{r}_{i,\hat{a}_i^p}^p - u_{i,\hat{a}_i^p}^p\right\}$.
7:     Send $\mathcal{A}_i^p$ to the central server.           ▷ Active arm set updating
8:     Receive $f_{i,a}^p$ for all $a \in \mathcal{A}_i^p$.
9:     **for** $a \in \mathcal{A}_i^p$ **do**           ▷ Collaborative exploration
10:         Pull arm $a$ for $f_{i,a}^p$ times and receive rewards $\{y_{i,t}\}_{t \in \mathcal{T}_{i,a}^p}$.
$$\hat{\theta}_{i,a}^p \leftarrow \left(\frac{1}{f_{i,a}^p} \sum_{t \in \mathcal{T}_{i,a}^p} y_{i,t}\right) \frac{x_{i,a}}{\|x_{i,a}\|^2}. \tag{3}$$
11:     **end for**
12:     Send $\{\hat{\theta}_{i,a}^p\}_{a \in \mathcal{A}_i^p}$ to the server; Pull $\hat{a}_i^p$ until phase length equals $f^p + K$.
13:     $p \leftarrow p + 1$.
14: **end while**

---

At the initialization phase, each client $i$ pulls every arm $a \in [K]$ once and receives a reward $y_{i,a}$, based on which it obtains an estimate of the projection of $\theta_a$. These estimates are sent to the server to construct a preliminary global estimate of $\theta_a$ for each $a$. The global estimates $\{\hat{\theta}_a^1\}_a$ and the potential matrices $\{V_a^1\}_a$ are then broadcast to all clients, after which phase $p = 1$ begins. Note that after receiving $\{\hat{\theta}_{i,a}^0\}_{i,a}$, the central server will keep a unit vector $\bar{e}_{i,a} = \hat{\theta}_{i,a}^0 / \|\hat{\theta}_{i,a}^0\|$ for all $i \in [M], a \in [K]$. Since $\hat{\theta}_{i,a}^0$ is a scaled version of $x_{i,a}$, $\bar{e}_{i,a}$ lies in $\text{range}(x_{i,a})$, and will be utilized to coordinate the arm pulling process (coined as *collaborative exploration*), as elaborated in Section 4.3.

At the beginning of phase $p$, after receiving the broadcast $\{\hat{\theta}_a^p\}_a$ and $\{V_a^p\}_a$ from the server, each client $i$ will utilize the $(\hat{\theta}_a^p, V_a^p)$ pair to estimate the expected rewards $r_{i,a}$ and obtain the confidence level according to Eqn. (2) for each $a \in \mathcal{A}_i^{p-1}$. Based on the constructed confidence interval, client $i$ then eliminates some arms in $\mathcal{A}_i^{p-1}$ and obtains $\mathcal{A}_i^p$.

Next, each client $i$ sends the newly constructed active arm set $\mathcal{A}_i^p$ to the server. The server then decides $f_{i,a}^p$, the number of times client $i$ pulling arm $a$ during the collaborative exploration step in phase $p$ for each $i \in [M]$ and $a \in \mathcal{A}_i^p$. The specific mechanism to decide $f_{i,a}^p$ is elaborated in Section 4.3.

After the collaborative exploration step, client $i$ performs least-square estimation (LSE) for each arm $a \in \mathcal{A}_i^p$ based on *local* observations collected in the current phase according to Eqn. (3) and then sends it to the server for global aggregation. Note that although $\hat{\theta}_{i,a}^p$ lies in $\text{range}(x_{i,a})$, the exact value of $x_{i,a}$ is not revealed to the server, thus preserving the privacy to certain extent.

---

**Algorithm 2** Fed-PE : Central server

**Input:** $T, M, K, \alpha, f^p$

1: **Initialization:** Receive $\{\hat{\theta}_{i,a}^0\}_{i,a}$; $\bar{e}_{i,a} \leftarrow \frac{\hat{\theta}_{i,a}^0}{\|\hat{\theta}_{i,a}^0\|}$ for all $i \in [M], a \in [K]$; $V_a^1 \leftarrow \left( \sum_{i \in [M]} \frac{\hat{\theta}_{i,a}^0 (\hat{\theta}_{i,a}^0)^\intercal}{\|\hat{\theta}_{i,a}^0\|} \right)^\dagger$,

$\hat{\theta}_a^1 \leftarrow V_a^1 \left( \sum_{i \in [M]} \hat{\theta}_{i,a}^0 \right)$ for all $a \in [K]$; Broadcast $\{\hat{\theta}_a^1, V_a^1\}_{a \in [K]}$; $p \leftarrow 1$.

2: **while** not reaching the time horizon $T$ **do**
3:     Receive $\{\mathcal{A}_i^p\}_{i \in [M]}$; Set $\mathcal{A}^p \leftarrow \cup_{i=1}^M \mathcal{A}_i^p$; Set $\mathcal{R}_a^p \leftarrow \{i : a \in \mathcal{A}_i^p\}$.
4:     Solve the multi-client G-optimal design in (6), and obtain solution $\pi^p = \{\pi_{i,a}^p\}_{i \in [M], a \in \mathcal{A}_i^p}$.
5:     For every client $i$, send $\{f_{i,a}^p := \lceil \pi_{i,a}^p f^p \rceil\}_{a \in \mathcal{A}_i^p}$.
6:     Receive $\{(a, \hat{\theta}_{i,a}^p)\}_{a \in \mathcal{A}_i^p}$ from each client $i$.
7:     **for** $a \in \mathcal{A}^p$ **do**                               ▷ Global aggregation

$$V_a^{p+1} \leftarrow \left( \sum_{i \in \mathcal{R}_a^p} f_{i,a}^p \frac{\hat{\theta}_{i,a}^p (\hat{\theta}_{i,a}^p)^\intercal}{\|\hat{\theta}_{i,a}^p\|^2} \right)^\dagger, \quad \hat{\theta}_a^{p+1} \leftarrow V_a^{p+1} \left( \sum_{i \in \mathcal{R}_a^p} f_{i,a}^p \hat{\theta}_{i,a}^p \right). \tag{4}$$

8:     **end for**
9:     Broadcast $\{(\hat{\theta}_a^{p+1}, V_a^{p+1})\}_{a \in \mathcal{A}^p}$ to all clients.
10:    $p \leftarrow p + 1$.
11: **end while**

---

### 4.3 Multi-client G-optimal Design

In this subsection, we elaborate the core design of Fed-PE, the collaborative exploration step. There are three main design objectives we aim to achieve: 1) As explained in Section 4.1, one of the main challenges in our federated linear contextual bandits setting is that, each client may benefit differently from the information exchange through the central server. To minimize the overall regret, for each arm $a \in \mathcal{A}^p$, it is desirable to ensure that after the global aggregation following the collaboration exploration in phase $p$, the uncertainty in $\hat{r}_{i,a}^{p+1}$ *across the clients* is balanced. 2) For each client $i$, in order to eliminate the sub-optimal arms efficiently, it is also important to guarantee that the uncertainty in $\hat{r}_{i,a}^{p+1}$ *across the arms* in $\mathcal{A}_i^p$ is balanced. 3) Finally, in order to ensure synchronized model updating, we aim to have each client perform the same number of arm pulling in each phase.

Motivated by those objectives, we propose a multi-client G-optimal design to coordinate the exploration of all clients. Specifically, we define $\pi_i^p : \mathcal{A}_i^p \to [0, 1]$ as a distribution on the active arm set

$\mathcal{A}_i^p$ for each $i$, and denote $\pi^p := (\pi_1^p, \ldots, \pi_M^p)$ as a vector in $\mathbb{R}^{\sum_{i \in [M]} |\mathcal{A}_i^p|}$. Let $e_{i,a} := x_{i,a}/\|x_{i,a}\|$. We note that $e_{i,a}$ equals to either $\bar{e}_{i,a}$ or $-\bar{e}_{i,a}$. Then, we define a feasible set $\mathcal{C}^p \subset \mathbb{R}^{\sum_{i \in [M]} |\mathcal{A}_i^p|}$ as follows:

$$\mathcal{C}^p = \left\{ \pi^p \left| \begin{array}{l} \pi_{i,a}^p \geq 0, \forall i \in [M], a \in \mathcal{A}_i^p, \\ \sum_{a \in \mathcal{A}_i^p} \pi_{i,a}^p = 1, \forall i \in [M], \\ \text{rank}(\{\pi_{i,a}^p e_{i,a}\}_{i \in \mathcal{R}_a^p}) = \text{rank}(\{e_{i,a}\}_{i \in \mathcal{R}_a^p}), \forall a \in \mathcal{A}^p \end{array} \right. \right\}. \tag{5}$$

We can verify that $\mathcal{C}^p$ is a convex set. The first two conditions ensure that $\{\pi_{i,a}^p\}_a$ form a valid distribution for each client $i$. We name the last condition as the "*rank-preserving*" condition. We note that the subspace spanned by the LHS of the rank-preserving condition is always a subset of that spanned by the RHS. Thus, once the rank is preserved, the subspaces spanned by the LHS and the RHS are the same. Thus, this condition ensures that every dimension of $\theta_a$ lying in $\text{range}(\{x_{i,a}\}_{i \in \mathcal{R}_a^p})$ will be explored under the collaborative exploration. Any violation of the "rank-preserving" condition will lead to information missing along the unexplored dimensions, which shall be prevented in order to reduce the uncertainty level regarding arm $a$ at every client $i \in \mathcal{R}_a^p$.

Then, we formulate the so called multi-client G-optimal design problem as follows:

$$\text{minimize } G(\pi) = \sum_{i=1}^{M} \max_{a \in \mathcal{A}_i^p} e_{i,a}^\mathsf{T} \left( \sum_{j \in \mathcal{R}_a^p} \pi_{j,a}^p e_{j,a} e_{j,a}^\mathsf{T} \right)^\dagger e_{i,a} \quad \text{s.t. } \pi^p \in \mathcal{C}^p. \tag{6}$$

We note that $e_{i,a}^\mathsf{T} \left( \sum_{j \in \mathcal{R}_a^p} \pi_{j,a}^p e_{j,a} e_{j,a}^\mathsf{T} \right)^\dagger e_{i,a}$ can be interpreted as an *approximate* measure of the uncertainty level along dimension $x_{i,a}$ if the arms are explored locally according to distributions $\{\pi_i^p\}_i$. Thus, the objective function is an approximate measure of the total uncertainties in the least explored arms at each of the clients. By solving (6), the aforementioned three design objectives can be met. We point out that although the server does not known $e_{i,a}$, the objective function remains the same when $e_{i,a}$ is replaced by $\bar{e}_{i,a}$. Thus, the server can simply use $\bar{e}_{i,a}$ to solve (6).

After solving (6) and obtaining $\{\pi_{i,a}^p\}$, the server would set $\{f_{i,a}^p := \lceil \pi_{i,a}^p f^p \rceil\}_{a \in \mathcal{A}_i^p}$ and send it to client $i$. Note that after taking the ceiling function, $\sum_{a \in \mathcal{A}_i^p} f_{i,a}^p$ may be greater than $f^p$. To ensure synchronized updating, each client $i$ would keep pulling the estimated best arm $\hat{a}_i^p$ until the phase length equals $f^p + K$.

We note that the multi-client G-optimal design formulated in (6) is related to the G-optimal design for the single-player linear bandits problem discussed in Lattimore and Szepesvári (2020), and the DELB algorithm for the distributed linear bandits in Wang et al. (2020). However, for such cases, the player(s) faces a single bandit problem, thus the objective is to simply obtain a distribution over a so-called core set of arms in order to minimize the maximum uncertainty *across the arms*. In contrast, due to the multiple clients involved in the federated bandits setting and the heterogeneous reward distributions, we are essentially solving $M$ *coupled* G-design problems, one associated with each client. Such coupling effect fundamentally changes the nature of the problem, leading to very different characterization of the problem and numerical approaches.

In Appendix C.1 of the supplementary material, we analyze an equivalent problem of the multi-client G-optimal design. We note that such equivalence essentially generalizes the equivalence between the original G-optimal design and D-optimal design in Lattimore and Szepesvári (2020) to the coupled design case. While the original G-design problem can be approximately solved through the Frank-Wolfe algorithm under an appropriate initialization (Todd, 2016), solving the multi-client G-optimal design problem is numerically non-trivial. In Appendix C.2, we propose a block coordinate ascent algorithm to solve the equivalent problem of (6) efficiently with guaranteed convergence.

## 4.4 Theoretical Analysis of `Fed-PE`

We now characterize the performance of the `Fed-PE` algorithm.

**Theorem 1** *Under Assumption 1, with probability at least $1 - \delta$, the cumulative regret under `Fed-PE` scales in $O\left( \sqrt{dKMT(\log(K(\log T)/\delta) + \min\{d, \log M\})} \right)$ and the communication cost scales in $O(Md^2 K \log T)$.*

The complete version of Theorem 1 and its proof can be found in Appendix D in the supplementary material. For the communication cost, at each phase $p$, client $i$ uploads at most $K$ local estimates with dimension $d$ and downloads at most $K$ global estimates and potential matrices with dimension $d$ and $d^2$, respectively. Thus, the upload cost is $O(MdK \log T)$ and the download cost is $O(Md^2K \log T)$.

**Remark 1** *When we set $\delta = O(\sqrt{dK/MT})$, the overall regret scales in $O(\sqrt{dKMT \log(MKT)})$, and the per-client regret scales in $O(\sqrt{dKT \log(MKT)/M})$. While the minimax lower bound for standard stochastic MAB scales in $\Omega(\sqrt{KT})$, the collaborative learning induced by* `Fed-PE` *leads to $\sqrt{d/M}$-fold reduction of the per-client regret. We also note that for single-player linear contextual bandits with disjoint parameters, the best known upper bound scales in $\tilde{O}(\sqrt{dKMT})$ (Dimakopoulou et al., 2017) over $MT$ arm pulls, which indicates that the regret of* `Fed-PE` *is close to the state-of-the-art centralized algorithms at a communication cost in $O(\log T)$.*

### 4.5 Enhanced Fed-PE

The original `Fed-PE` algorithm requires exponentially increasing $f^p$ in order to achieve the regret upper bound in Theorem 1. This is because in each phase $p$, we only utilize the rewards collected in phase $p-1$ to estimate $\theta_a$. While this simplifies the analysis, the measurements collected in earlier phases cannot be utilized. In order to overcome this limitation, we propose an `Enhanced Fed-PE` algorithm by leveraging all historical information. `Enhanced Fed-PE` achieves different tradeoffs between communication cost and regret performance by adjusting $f^p$. The detailed description and analysis of `Enhanced Fed-PE` for different selection of $f^p$ can be found in Appendix E in the supplementary material.

### 4.6 Lower Bound

To derive a tight lower bound, we focus on a set of representative policies defined as follows.

**Definition 1 (Collinearly-dependent policy)** *Two clients $i$ and $j$ are called* collinear *if there exist an arm $a \in [K]$ and a subset $\mathcal{S} \subset [M]$ such that the following conditions are satisfied: 1) $x_{i,a} \notin \text{span}(\{x_{m,a}|m \in \mathcal{S}\})$; and 2) $x_{i,a} \in \text{span}(\{x_{m,a}|m \in \mathcal{S}\} \cup \{x_{j,a}\})$. For any two clients $i$ and $j$ that are not collinear, if the action of client $i$ is independent of the action of $j$ under a policy $\pi$, then, the policy is called a collinearly-dependent policy.*

We note that the definition of collinearly-dependent policies is actually quite natural. Intuitively, for two clients that are not collinear, their local observations on any arm $a$ cannot be utilized to improve each other's knowledge of their own local models. As a result, they should not affect each other's decision-making process. As shown in the supplementary material, we can verify that the most celebrated ridge regression based LinUCB type of policies (Li et al., 2010), Thompson sampling based polices with Gaussian priors (Agrawal and Goyal, 2013b), and least-square estimation based policies, including `Fed-PE`, all fall in this category.

**Theorem 2** *For any collinearly-dependent policy, there exists an instance of the federated linear contextual bandits such that the regret is lower bounded as $R(T) = \Omega(\sqrt{dKMT})$.*

**Remark 2** *Theorem 2 essentially shows that, even if raw data transmission and instantaneous communication are allowed and other collinearly-dependent policies are adopted, we cannot improve the order of the regret summarized in Theorem 1 much, i.e.,* `Fed-PE` *is order-optimal up to $\sqrt{\log(KMT)}$.*

The proof of Theorem 2 relies on the construction of a special instance of the federated linear contextual bandits where the clients can be divided into $d$ groups. Clients in each group face the same $K$-armed stochastic bandits model locally, while clients from two distinct groups are not collinear. Analyzing the regret bound in each individual group, we can show that it is lower bounded by $\Omega(\sqrt{KMT/d})$. Then, by utilizing the property of collinearly-dependent policies, we can show that the overall regret is lower bounded by $\Omega(\sqrt{dKMT})$ for this scenario. More discussions on the collinearly-dependent policies and the complete proof of Theorem 2 can be found in Appendix F in the supplementary material.

# 5 Federated Linear Contextual Bandits: Shared Parameter Case

The `Fed-PE` algorithm can be slightly modified for the shared parameter case where $\theta_a = \theta, \forall a \in [K]$. While the client side operation stays the same, the global aggregation step at the server side in (4) can be changed by letting the "potential matrix" $V^p$ be $(\sum_{a \in [K]} (V_a^p)^\dagger)^\dagger$, and the global estimator $\hat{\theta}^p$ be $V^p \left( \sum_{i \in [M]} \sum_{a \in \mathcal{A}_i^{p-1}} f_{i,a}^{p-1} \hat{\theta}_{i,a}^{p-1} \right)$. Below, we present the main result for this case and leave the detailed algorithm description and regret analysis in the supplementary material.

**Theorem 3** *Under Assumption 1, with probability at least $1-\delta$, the regret of the adapted `Fed-PE` for the shared parameter case is upper bounded by $O\left( \sqrt{dMT(\log((\log T)/\delta) + \min\{d, \log MK\})} \right)$, and the communication cost scales in $O((KdM + d^2M)\log T)$.*

**Remark 3** *By setting $\delta = O(\sqrt{1/MT})$, we can show that the overall regret scales in $O(\sqrt{dMT\log(MTK)})$. We note that this bound improves the regret bound for the no differential privacy guarantee case in Dubey and Pentland (2020) by a factor of $\sqrt{\log T}$, although our settings are slightly different. By assuming all clients face the same linear bandits in this shared parameter setting, the regret in the federated setting over horizon $T$ must be worse than the linear bandits over horizon $MT$. Since the latter is lowered bounded by $\Omega(\sqrt{dMT})$ (Chu et al., 2011), the minimax regret for the shared parameter setting is bounded by $\Omega(\sqrt{dMT})$ as well. Thus, the modified `Fed-PE` is near-optimal for this case.*

*In terms of the communication cost, the uploading cost stays the same as in the disjoint parameter case, while the broadcast cost is reduced by a factor of $K$, since only one potential matrix needs to be broadcast for the shared parameter case.*

# 6 Experiments

Experiment results using both synthetic and real-world datasets are reported in this section to evaluate `Fed-PE` and the proposed enhancement. Additional experimental details and more experimental results can be found in the supplementary material. We consider four different algorithms, namely, `Fed-PE`, `Enhanced Fed-PE`, local UCB without communication, and a modified `Fed-PE` algorithm with full information exchange after collaborative exploration in each phase (coined as 'Collaborative' in Figure 1). For all experiments, we set $T = 2^{17}$, $f^p = 2^p, p \in \{1, 2, \ldots, 16\}$, and run 10 trials. For `Fed-PE` and its variants, we choose $\delta = 0.1$. Note that other values of $\delta$ may further improve the regret. We evaluate the algorithms on both synthetic and MovieLens-100K datasets.

**Synthetic Dataset**: We first set $M = 100, K = 10$, and $d = 3$. We set $\{\theta_a\}$ as the canonical basis of $\mathbb{R}^3$. The feature vectors $x_{i,a}$ are generated randomly ensuring that the suboptimality reward gaps lie in $[0.2, 0.4]$ and $\ell = 0.5, L = 1$. The per-client cumulative regret as a function of $T$ is plotted in Figure 1(a). We see that `Enhanced Fed-PE` outperforms `Fed-PE` while being slightly worse than 'Collaborative'. This indicates that keeping feature vectors $x_{i,a}$ private to clients does not impact the learning performance significantly. All `Fed-PE` related algorithms outperform local UCB when $T$ is sufficiently large, demonstrating the effectiveness of communication in improving learning locally. We also set $K = 10, d = 4$, and vary the number of clients $M$. The performance of `Enhanced Fed-PE` is plotted in Figure 1(b). We note that the per-client regret monotonically decreases as $M$ increases, corroborating the theoretical results.

**Movielens Dataset**: We then use the MovieLens-100K dataset (Harper and Konstan, 2015) to evaluate the performances. Motivated by Bogunovic et al. (2021), we first complete the rating matrix $R = [r_{i,a}] \in \mathbb{R}^{943 \times 1682}$ through collaborative filtering (Morabia, 2019), and then use non-negative matrix factorization with 3 latent factors to get $R = WH$, where $W \in \mathbb{R}^{943 \times 3}, H \in \mathbb{R}^{3 \times 1682}$. Let $x_{i,a}$ be the $i$th row vector of $W$. We apply the $k$-means algorithm to the row vectors of $H$ to produce $K = 30$ groups (arms), and let $\theta_a$ be the center of the $a$-th group. Finally, we randomly choose $M = 100$ users' feature vectors. We observe that $0.4 \le \|x_{i,a}\|^2 \le 0.8$, and the suboptimality gaps lie in $[0.01, 0.8]$. The regret performances of the algorithms are plotted in Figure 1(c). The curves show similar characteristics as in Figure 1(a). These results demonstrate the effectiveness of collaborative learning in the federated bandits setting.

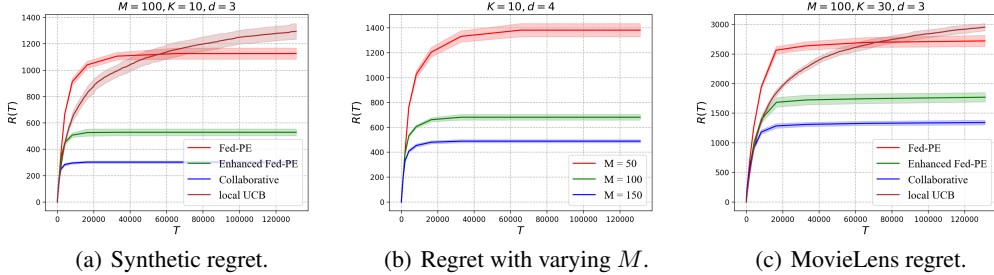

Figure 1: Pseudo-regret over $T$. Shaded area indicates the standard deviation.

# 7    Discussion and Conclusion

In this work, we have considered a novel federated linear contextual bandits model, which naturally connects local stochastic MAB models with linear contextual bandits through common global parameters. While each client can only observe a projection of each global parameter in its own subspace, `Fed-PE` utilizes the geometric structure of the local estimates to reconstruct the global parameters and guide efficient collaborative exploration. Theoretical analysis indicates that `Fed-PE` achieves near-optimal regret for both disjoint and shared parameter cases with a communication cost in the order of $O(\log T)$.

An interesting open question is whether we can further reduce the communication cost without downgrading the regret performance. In particular, we note the the original single-player G-optimal design allows for a sparse solution whose support is of size $d(d+1)/2$. Our numerical results indicate that such sparse solutions exist for the multi-client G-optimal design as well. Utilizing the sparsity of the solution may reduce the communication cost significantly. Theoretical characterization of the existence of such sparse solutions is our next step.

Another possible direction to explore is to incorporate the differential privacy mechanism to the `Fed-PE` framework. Although local feature vectors $\{x_{i,a}\}$ are kept private under `Fed-PE`, local estimate $\hat{\theta}_{i,a}$ lies in $\mathrm{range}(x_{i,a})$, thus revealing the direction of $x_{i,a}$ to the central server. We aim to add certain perturbation on $\hat{\theta}_{i,a}$ in order to obfuscate the direction information without significantly affecting the regret performance.

## Acknowledgments and Disclosure of Funding

The work of RH and JY was supported by the US National Science Foundation under Grants CNS-1956276, CNS-2003131, CNS-2114542, and ECCS-2030026. CS acknowledges the funding support by the US National Science Foundation under Grants ECCS-2029978, ECCS-2033671, and CNS-2002902. WW's work was done before he joined Facebook.

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
