# Supplementary Material for "Federated Linear Contextual Bandits"

**Ruiquan Huang, Weiqiang Wu, Jing Yang, and Cong Shen**

## A  Related Works

**Linear contextual bandits.** The reward model considered in this work is similar to that of linear contextual bandits in the literature. The single-agent setting is first introduced in Auer (2003) through the LinRel algorithm and is subsequently improved through the OFUL algorithm in Dani et al. (2008) and the LinUCB algorithm in Li et al. (2010). Rusmevichientong and Tsitsiklis (2010) extend the work of Dani et al. (2008) by considering both optimistic and explore-then-commit strategies. A modified version of LinUCB, named SupLinUCB, is considered in Chu et al. (2011), which is further improved by Valko et al. (2013). This line of literature typically allows for sequential stochastic (Goldenshluger and Zeevi, 2013; Bastani and Bayati, 2015) or adversarial (Raghavan et al., 2018; Kannan et al., 2018) context arrivals, which is different from the fixed context setting and distributed nature of the federated linear contextual bandits considered in this work.

**Batched and parallel bandits.** Batched bandits is the setting where the time axis is partitioned into batches, and the decision at each time $t$ depends only on observations from batches strictly prior to the current one (Perchet et al., 2016). While Perchet et al. (2016) study the two-armed stochastic bandits, Gao et al. (2019) and Han et al. (2020) extend it to the $K$-armed MAB and the linear contextual bandits, respectively. The parallel linear contextual bandits studied in Chan et al. (2021) is essentially similar to the batched bandits setting in which $P$ distinct processors perform simultaneous queries in batches. We note that the phased decision processes in our setting is similar to batched bandits, as the decision in a phase only depends on observations collected in previous phases. However, the distributed network structure and the associated communication and privacy concerns are not considered in the batched bandits setting.

**Collaborative and distributed bandits.** The collaborative and distributed bandits with multiple agents has gained growing interest recently. One research direction is the multi-player multi-armed bandits (MP-MAB) problem (Liu and Zhao, 2010; Anandkumar et al., 2011), where collision occurs when two players simultaneously play the same arm. Without explicit communication among players, the main focus of MP-MAP is to avoid (Rosenski et al., 2016; Besson and Kaufmann, 2018; Avner and Mannor, 2014) or exploit (Boursier and Perchet, 2019; Shi et al., 2020) such collisions in order to maximize the collective cumulative rewards.

When action collision is not considered, Landgren et al. (2016, 2018) and Martínez-Rubio et al. (2019) study distributed bandits in which multiple agents face the same MAB instance, and the agents collaboratively share their estimates over a fixed communication graph in order to design consensus-based distributed estimation algorithms to estimate the mean of rewards at each arm. Szorenyi et al. (2013) considers a similar setup where in each round an agent is able to communicate with a few random peers. Korda et al. (2016) considers the case where clients in different unknown clusters face independent bandit problems, and every agent can communicate with only one other agent per round. Similar approaches have been extended to the contextual bandits and recommender systems, where user/context similarities are exploited to improve sample efficiency of online recommendation (Cesa-Bianchi et al., 2013; Wu et al., 2016; Gentile et al., 2014, 2017). The communication and coordination among the clients in those works are fundamentally different from our work.

Wang et al. (2020) investigates communication-efficient distributed linear bandits, where the agents can communicate with a server by sending and receiving packets. It proposes two algorithms, namely, DELB and DisLinUCB, for fixed and time-varying action sets, respectively. The fixed action set setting is quite similar to our setup, except that it assumes that all agents face the same bandits problem, which does not take data heterogeneity into consideration.

**Federated bandits.** A few recent works have touched upon the concept of federated bandits. With heterogeneous reward distributions at local clients, Shi and Shen (2021) and Shi et al. (2021) investigate efficient client-server communication and coordination protocols for federated MAB without and with personalization, respectively. Agarwal et al. (2020) studies regression-based contextual bandits as an example of the federated residual learning framework, where the reward of a client depends on both a global model and a local model. Li et al. (2020) and Zhu et al. (2021)

focus on differential privacy based local data privacy protection in federated bandits. While the linear contextual bandit model considered in Dubey and Pentland (2020) is similar to the shared parameter case studied this work, it focuses on federated differential privacy and proposes a LinUCB-based algorithm, which incurs a much higher regret compared with our result.

# B  Preliminaries

## B.1  Notations

Throughout this paper, we use $\|x\|_V$ to denote $\sqrt{x^\intercal V x}$. The *range* of a matrix $A$, denoted by $\mathrm{range}(A)$, is the subspace spanned by the column vectors of $A$. Occasionally we use $\mathrm{span}(A)$ to denote $\mathrm{range}(A)$ as well. We use $\mathrm{rank}(\{x_i\}_i)$ to denote the maximum number of linearly independent vectors in $\{x_i\}_i$, and $\mathrm{span}(\{x_i\}_i)$ to denote the subspace spanned by them. For any positive semi-definite matrices $X$ and $Y$ of the same size, $X \succeq Y$ implies that $X - Y$ is positive semi-definite. $\mathbb{1}\{\cdot\}$ is the indicator function while $I_d$ is a $d \times d$ identify matrix.

## B.2  Matrix Analysis

**Definition 2 (Pseudo-inverse of matrices)** *Given a matrix $A$, the pseudo-inverse of $A$ is a unique matrix, denoted by $A^\dagger$, that satisfies the following properties:*

$$AA^\dagger A = A, \quad A^\dagger AA^\dagger = A^\dagger, \quad (AA^\dagger)^\intercal = AA^\dagger, \quad (A^\dagger A)^\intercal = A^\dagger A.$$

**Definition 3 (Pseudo-determinant of matrices)** *For a square matrix $A \in \mathbb{R}^{n\times n}$, the pseudo-determinant is defined as*

$$\mathrm{Det}(A) = \lim_{\epsilon \to 0} \frac{\det(A + \epsilon I)}{\epsilon^{n-\mathrm{rank}(A)}},$$

*and the generalized pseudo-determinant with respect to a degree $s$ is defined as*

$$\mathrm{Det}_s(A) = \lim_{\epsilon \to 0} \frac{\det(A + \epsilon I)}{\epsilon^{n-s}}.$$

**Remark 4** *Note that $\mathrm{Det}_s(A) = 0$ when $s > \mathrm{rank}(A)$, and $\mathrm{Det}_s(A) = \infty$ if $s < \mathrm{rank}(A)$. When $s = \mathrm{rank}(A)$, $\mathrm{Det}_{\mathrm{rank}(A)}(A)$ becomes $\mathrm{Det}(A)$, the standard pseudo-determinant of matrix $A$. We introduce the definition of generalized pseudo-determinant in order to handle cases when the rank of $A$ is uncertain.*

We first characterize some important properties of the pseudo-inverse of a matrix.

**Proposition 1** *Suppose $A \in \mathbb{R}^{n\times n}$ is a symmetric matrix, and $v \in \mathrm{range}(A)$. Then, the following limit relations hold:*

*1) $A^\dagger = \lim_{\epsilon \to 0}(A + \epsilon I)^{-1}A(A + \epsilon I)^{-1}$.*
*2) $A = \lim_{\epsilon \to 0} A(A + \epsilon I)^{-1}A$.*
*3) $A^\dagger v = \lim_{\epsilon \to 0}(A + \epsilon I)^{-1}v$.*

*Proof.* Since $A$ is a symmetric matrix, we can decompose it as $A = U\Lambda U^\intercal$, where $UU^\intercal = I$ and $\Lambda = \mathrm{diag}(\lambda_1, \ldots, \lambda_n)$. We assume $|\lambda_1| \geq |\lambda_2| \geq \ldots \geq |\lambda_n|$ are the eigenvalues of $A$. If $\mathrm{rank}(A) = d \leq n$, then $|\lambda_d| > 0 = |\lambda_{d+1}|$. There must exist $d$ constants $c_1, \ldots, c_d$ such that $v = \sum_{s=1}^d c_s u_s$, where $u_s$ is the $s$th column vector of matrix $U$.

Note that when $A$ is symmetric,

$$A^\dagger = U\mathrm{diag}\left(\frac{1}{\lambda_1}, \ldots, \frac{1}{\lambda_d}, 0, \ldots, 0\right)U^\intercal.$$

For the first two relations, note that $A + \epsilon I = U\Lambda U^\intercal + \epsilon UU^\intercal = U\mathrm{diag}(\lambda_1 + \epsilon, \ldots, \lambda_n + \epsilon)U^\intercal$. Thus, for sufficiently small $\epsilon$, we have

$$(A + \epsilon I)^{-1} = U\mathrm{diag}\left(\frac{1}{\lambda_1 + \epsilon}, \ldots, \frac{1}{\lambda_n + \epsilon}\right)U^\intercal. \tag{7}$$

Substituting (7) into 1) and 2), the first two relations can be readily obtained.

For 3), we express $v = Uc$, where $c = (c_1, \ldots, c_d, 0, \ldots, 0)^\mathsf{T}$. Then we have

$$\lim_{\epsilon \to 0} (A + \epsilon I)^{-1} v = \lim_{\epsilon \to 0} U \mathrm{diag}\left( \frac{1}{\lambda_1 + \epsilon}, \ldots, \frac{1}{\lambda_n + \epsilon} \right) U^\mathsf{T} U c \tag{8}$$

$$= \lim_{\epsilon \to 0} U \left( \frac{c_1}{\lambda_1 + \epsilon}, \ldots, \frac{c_d}{\lambda_d + \epsilon}, 0, \ldots, 0 \right)^\mathsf{T} \tag{9}$$

$$= U \mathrm{diag}\left( \frac{1}{\lambda_1}, \ldots, \frac{1}{\lambda_d}, 0, \ldots, 0 \right) U^\mathsf{T} U c \tag{10}$$

$$= A^\dagger v. \tag{11}$$

∎

Next, we present several useful lemmas that are needed in the analysis of the generalized G-optimal design discussed in Appendix C.

**Lemma 1 (Jacobi's formula (Magnus and Neudecker, 1999))** *If $A : \mathbb{R} \to \mathbb{R}^{n \times n}$ is differentiable in its domain, and $A(t)$ is invertible, then*

$$\frac{d \det(A(t))}{dt} = \mathrm{trace}\left( \mathrm{adj}(A(t)) \cdot \frac{dA(t)}{dt} \right),$$

*where $\mathrm{adj}(A(t))$ is the adjugate of $A(t)$, i.e. $\mathrm{adj}(A(t)) A(t) = \det(A(t)) \cdot I_n$.*

Lemma 2 below explicitly identifies the derivative of $\log \mathrm{Det}(\cdot)$. We note that the general derivative of pseudo-determinant is studied in Holbrook (2018).

**Lemma 2** *For a given index set $\mathcal{I}$ and a set of vectors $v_s \in \mathbb{R}^n$ where $s \in \mathcal{I}$, consider the following function in terms of $\pi := \{\pi_s\}_{s \in \mathcal{I}}$:*

$$F(\pi) = \log \mathrm{Det}\left( \sum_{s \in \mathcal{I}} \pi_s v_s v_s^\mathsf{T} \right)$$

*defined over $\{\pi | \pi_s \geq 0, \forall s \in \mathcal{I}, \mathrm{rank}\left( \sum_{s \in \mathcal{I}} \pi_s v_s v_s^\mathsf{T} \right) = \mathrm{rank}(\{v_s\}_{s \in \mathcal{I}}) := D\}$.*

*Then, the $\iota$-th coordinate of the gradient of $F(\pi)$ satisfies*

$$(\nabla F)_\iota = v_\iota^\mathsf{T} \left( \sum_{s \in \mathcal{I}} \pi_s v_s v_s^\mathsf{T} \right)^\dagger v_\iota. \tag{12}$$

*Proof.* Denote $A := \sum_{s \in \mathcal{I}} \pi_s v_s v_s^\mathsf{T}$. Then, according to the definition of pseudo-determinant in Definition 3, we have

$$\frac{\partial F(\pi)}{\partial \pi_\iota} = \frac{1}{\mathrm{Det}(A)} \frac{\partial \mathrm{Det}(A)}{\partial \pi_\iota} \tag{13}$$

$$= \lim_{\epsilon \to 0} \frac{1}{\mathrm{Det}(A)} \frac{\partial \det(A + \epsilon I)}{\epsilon^{n-D} \partial \pi_\iota} \tag{14}$$

$$= \lim_{\epsilon \to 0} \frac{1}{\mathrm{Det}(A) \epsilon^{n-D}} \mathrm{trace}(\mathrm{adj}(A + \epsilon I) v_\iota v_\iota^\mathsf{T}) \tag{15}$$

$$= \lim_{\epsilon \to 0} \frac{1}{\mathrm{Det}(A) \epsilon^{n-D}} \mathrm{trace}(\det(A + \epsilon I)(A + \epsilon I)^{-1} v_\iota v_\iota^\mathsf{T}) \tag{16}$$

$$= \lim_{\epsilon \to 0} v_\iota^\mathsf{T} (A + \epsilon I)^{-1} v_\iota \tag{17}$$

$$= v_\iota^\mathsf{T} \left( \sum_{s \in \mathcal{I}} \pi_s v_s v_s^\mathsf{T} \right)^\dagger v_\iota, \tag{18}$$

where Eqn. (15) follows from Lemma 1, (16) follows from the definition of adjugate, and (17) follows from the definition of $\mathrm{Det}(A)$ in Definition 3 and the fact that $\mathrm{trace}(AB) = \mathrm{trace}(BA)$ when their dimensions match. Eqn. (18) is due to Proposition 1-3) since $v_\iota \in \mathrm{range}(A)$. ∎

The following lemma establishes the concavity of $\log \mathrm{Det}(\cdot)$, which generalizes the result for $\log \det(\cdot)$ (Boyd and Vandenberghe, 2004) to the pseudo-determinant case.

**Lemma 3** *Let $S$ be a subspace of $\mathbb{R}^n$. Then, $\log \mathrm{Det}(A)$ is a concave function in any convex subset of $\mathcal{M}_S = \{A | A = A^\mathsf{T}, \mathrm{range}(A) = S, A \succeq 0\}$, i.e., the collection of symmetric positive semi-definite matrices with range $S$.*

*Proof.* We show the concavity by considering an arbitrary line, defined by $X = Y + tZ$, where $X, Y, Z$ are matrices lying in a convex subset of $\mathcal{M}_S$. Assume $\mathrm{rank}(X) = k$.

Let $g(t) = \log \mathrm{Det}(Y + tZ)$. Then,

$$
\begin{aligned}
g(t) &= \log \lim_{\epsilon \to 0} \frac{\det(Y + tZ + \epsilon I)}{\epsilon^{n-k}} \\
&= \lim_{\epsilon \to 0} \log \det(Y + \epsilon I + tZ) - (n - k) \log \epsilon \\
&= \lim_{\epsilon \to 0} \log \det \left( (Y + \epsilon I)^{\frac{1}{2}} \left( I + t(Y + \epsilon I)^{-\frac{1}{2}} Z (Y + \epsilon I)^{-\frac{1}{2}} \right) (Y + \epsilon I)^{\frac{1}{2}} \right) - (n - k) \log \epsilon \\
&= \lim_{\epsilon \to 0} \log \det(Y + \epsilon I) + \log \det \left( I + t(Y + \epsilon I)^{-\frac{1}{2}} Z (Y + \epsilon I)^{-\frac{1}{2}} \right) - (n - k) \log \epsilon \\
&= \lim_{\epsilon \to 0} \log \frac{\det(Y + \epsilon I)}{\epsilon^{n-k}} + \log \det \left( I + t(Y + \epsilon I)^{-\frac{1}{2}} Z (Y + \epsilon I)^{-\frac{1}{2}} \right) \\
&= \log \mathrm{Det}\, Y + \lim_{\epsilon \to 0} \log \det \left( I + t(Y + \epsilon I)^{-\frac{1}{2}} Z (Y + \epsilon I)^{-\frac{1}{2}} \right) \\
&= \log \mathrm{Det}\, Y + \sum_{i=1}^{k} \log \det(1 + t\lambda_i),
\end{aligned}
$$

where $\lambda_1, \lambda_2, \ldots, \lambda_k$ are eigenvalues of $\lim_{\epsilon \to 0} (Y + \epsilon I)^{-\frac{1}{2}} Z (Y + \epsilon I)^{-\frac{1}{2}}$.

Taking the second derivative of $g(t)$, we have

$$
\frac{d^2 g(t)}{dt^2} = -\sum_{i=1}^{k} \frac{\lambda_i^2}{(1 + t\lambda_i)^2} < 0. \tag{19}
$$

Therefore, $g(t)$ is concave, and the proof is complete. ∎

### B.3 Important Inequalities

In this subsection, we present two important inequalities that will be utilized in Appendix D and Appendix E, respectively.

**Lemma 4** *Let $\zeta \in \mathbb{R}^n$ be a 1-sub-Gaussian random vector conditioned on $\mathcal{F}_p$ and $A \in \mathbb{R}^{n \times n}$ be a $\mathcal{F}_p$-measurable matrix. Let $\lambda > 0$ and $\det(I_n - 2\lambda AA^\mathsf{T}) > 0$. Then, we have*

$$
\mathbb{E}\left[ e^{\lambda \|A\zeta\|^2} \middle| \mathcal{F}_p \right] \leq \sqrt{\frac{1}{\det(I_n - 2\lambda AA^\mathsf{T})}}. \tag{20}
$$

*Proof.* Assume $x \sim \mathcal{N}(0, 2\lambda I_n)$ is an independent Gaussian random vector. Then, we have

$$
\mathbb{E}_{x, \zeta}\left[ e^{x^\mathsf{T} A\zeta} \middle| \mathcal{F}_p \right] = \mathbb{E}_\zeta \left[ \mathbb{E}_x \left[ e^{x^\mathsf{T} A\zeta} | A, \zeta \right] \middle| \mathcal{F}_p \right] = \mathbb{E}\left[ e^{\lambda \|A\zeta\|^2} \middle| \mathcal{F}_p \right]. \tag{21}
$$

On the other hand,

$$
\mathbb{E}_{x, \zeta}\left[ e^{x^\mathsf{T} A\zeta} \middle| \mathcal{F}_p \right] = \mathbb{E}_x \left[ \mathbb{E}_\zeta \left[ e^{x^\mathsf{T} A\zeta} | x \right] \middle| \mathcal{F}_p \right] \tag{22}
$$

$$\leq \mathbb{E}\left[ e^{\|A^\mathsf{T} x\|^2/2} \middle| \mathcal{F}_p \right] \tag{23}$$

$$= \int \sqrt{\frac{1}{(4\pi\lambda)^n}} e^{\frac{1}{2}x^\mathsf{T} AA^\mathsf{T} x - \frac{1}{4\lambda}x^\mathsf{T} x} dx \tag{24}$$

$$= \sqrt{\frac{1}{(2\lambda)^n \det(\frac{1}{2\lambda}I_n - AA^\mathsf{T})}} \tag{25}$$

$$= \sqrt{\frac{1}{\det(I_n - 2\lambda AA^\mathsf{T})}}, \tag{26}$$

where (23) is due to the property of sub-Gaussian random vectors. The result then follows by combining Eqn. (21) and Eqn. (26). ∎

We note that similar but slightly more complicated versions of the inequalities for standard Gaussian vectors and sub-Gaussian vectors have been shown in Laurent and Massart (2000) and Hsu et al. (2012), respectively.

The following result is a simplified version of the classical self-normalized bound (Theorem 1 in Abbasi-Yadkori et al. (2011)) with one-dimensional random variables. We recover the proof for completeness.

**Lemma 5** *Suppose $\sigma_q$ is $\mathcal{F}_q$-measurable, and $\{X_q\}_{q \geq 1}$ is an $\mathcal{F}$-adapted $\sigma_q$-sub-Gaussian random variable, i.e., $\mathbb{E}[\exp(\lambda X_q)|\mathcal{F}_q] \leq \exp(\lambda^2 \sigma_q^2/2)$. Then, for any $\sigma, \delta > 0$:*

$$\mathbb{P}\left[ \exists p \text{ s.t. } \left| \sum_{q=1}^p X_q \right| \geq \sqrt{\left( \sigma^2 + \sum_{q=1}^p \sigma_q^2 \right) \log \frac{\sigma^2 + \sum_{q=1}^p \sigma_q^2}{\sigma^2 \delta^2}} \right] \leq \delta.$$

*Proof.* Let $M_p(x) := \exp\left( x \sum_{q=1}^p X_q - \frac{1}{2}x^2 \sum_{q=1}^p \sigma_q^2 \right)$. We can verify that $\mathbb{E}[M_p(x)|\mathcal{F}_p] \leq M_{p-1}(x)$, and $\bar{M}_p := \int \frac{\exp(-\sigma^2 x^2/2)}{\sqrt{2\pi/\sigma^2}} M_p(x) dx$ is a super-martingale. Besides,

$$\bar{M}_p = \int \exp\left( \frac{(\sum_{q=1}^p X_q)^2}{2\sigma^2 + 2\sum_{q=1}^p \sigma_q^2} \right) \exp\left( -\frac{\sigma^2 + \sum_{q=1}^p \sigma_q^2}{2} \left( x - \frac{\sum_{q=1}^p X_q}{\sigma^2 + \sum_{q=1}^p \sigma_q^2} \right)^2 \right) \frac{\sigma dx}{\sqrt{2\pi}}$$

$$= \sqrt{\frac{\sigma^2}{\sigma^2 + \sum_{q=1}^p \sigma_q^2}} \exp\left( \frac{(\sum_{q=1}^p X_q)^2}{2\sigma^2 + 2\sum_{q=1}^p \sigma_q^2} \right).$$

By the maximal inequality (Theorem 3.9 in Lattimore and Szepesvári (2020)), and the fact that $\mathbb{E}[\bar{M}_1] \leq 1$, we have

$$\mathbb{P}\left[ \exists p \text{ s.t. } \sqrt{\frac{\sigma^2}{\sigma^2 + \sum_{q=1}^p \sigma_q^2}} \exp\left( \frac{(\sum_{q=1}^p X_q)^2}{2\sigma^2 + 2\sum_{q=1}^p \sigma_q^2} \right) \geq \frac{1}{\delta} \right] = \mathbb{P}\left[ \max_p \bar{M}_p \geq \frac{1}{\delta} \right] \leq \delta \mathbb{E}[\bar{M}_1] \leq \delta.$$

∎

## C Generalized G-optimal Design

### C.1 Analysis of the Multi-client G-optimal Design

After extending the techniques in (Lattimore and Szepesvári, 2020) to the multi-constraint and pseudo-determinant case, we establish an important property of the optimization problems (6) and (27), as stated in the following lemma.

**Lemma 6** *Given sets $\mathcal{A}_i^p \subset [K]$ and $\mathcal{R}_a^p \subset [M]$ under* `Fed-PE` *in phase $p$, consider the following optimization problem:*

$$\textit{maximize} \quad F(\pi) = \sum_{a \in [K]} \log \mathrm{Det}\left(\sum_{j \in \mathcal{R}_a^p} \pi_{j,a}^p e_{j,a} e_{j,a}^\mathsf{T}\right) \quad \textit{s.t. } \pi^p \in \mathcal{C}^p. \tag{27}$$

*Denote $d_a^p := \mathrm{rank}(\{e_{i,a}\}_{i \in \mathcal{R}_a^p}), \forall a \in [K]$. Then, we have the following equivalent statements:*

1) *$\pi^*$ is a maximizer of $F(\pi)$.*
2) *$\pi^*$ is a minimizer of $G(\pi)$ defined in Eqn. (6).*
3) *$G(\pi^*) = \sum_{a \in [K]} d_a^p$.*

**Remark 5** *Lemma 6 is surprisingly similar to the original equivalence between G-optimal design and D-optimal design introduced in Theorem 21.1 of Lattimore and Szepesvári (2020), which was first studied in Kiefer and Wolfowitz (1960). It provides an alternative method to solve (6) by maximizing $F(\pi)$. We note that a determinant maximization problem similar to (27) with two linear constraints has been studied in Harman and Benková (2014), while a generalized experimental design similar to (6) has been investigated in Harman et al. (2014). However, those works do not establish the equivalence between the two problems. Although an algorithm is proposed in Harman et al. (2014) to solve the generalized experimental design problem, it does not provide any convergence guarantees.*

*The generalized multi-client G-optimal design problem studied in this work can be treated as a particularized version of the generalized experimental design in Harman et al. (2014). As we will show below, the pseudo-determinant in the objective function in (27) can be viewed as the determinant of a block diagonal matrix, and the client-wise constraints essentially are imposed on individual blocks. Such special structure enables us to establish the equivalence between the generalized multi-agent G-optimal design in (6) and the optimization problem in (27), as well as to design a novel block coordinate ascent method to solve (27) efficiently without violating the constraints.*

In order to prove Lemma 6, we prove a generalized version of it instead. Before we proceed, we introduce the following notations. First, we omit the phase index $p$ in this subsection without causing any ambiguity. Next, we embed each normalized feature vector $e_{i,a} := x_{i,a}/\|x_{i,a}\|$ into a higher dimensional space as follows: we construct a vector $v_{i,a} \in \mathbb{R}^{dK}$ as:

$$v_{i,a} = (0, \ldots, 0, \underbrace{e_{i,a}^\mathsf{T}}_{a\text{-th block}}, 0, \ldots, 0)^\mathsf{T} \in \mathbb{R}^{dK},$$

i.e., the $[(a-1)d + k]$-th coordinate of $v_{i,a}$ is the $k$-th coordinate of $e_{i,a}$ for all $k \in [d]$ and all other coordinates of $v_{i,a}$ are 0.

We can verify that

$$e_{i,a}^\mathsf{T}\left(\sum_{j \in \mathcal{R}_a} \pi_{j,a} e_{j,a} e_{j,a}^\mathsf{T}\right)^\dagger e_{i,a} = v_{i,a}^\mathsf{T}\left(\sum_{a \in [K]}\sum_{j \in \mathcal{R}_a} \pi_{j,a} v_{j,a} v_{j,a}^\mathsf{T}\right)^\dagger v_{i,a},$$

and

$$\prod_{a \in [K]} \mathrm{Det}\left(\sum_{j \in \mathcal{R}_a} \pi_{j,a} e_{j,a} e_{j,a}^\mathsf{T}\right) = \mathrm{Det}\left(\sum_{a \in [K]}\sum_{j \in \mathcal{R}_a} \pi_{j,a} v_{j,a} v_{j,a}^\mathsf{T}\right),$$

due to the fact that $\sum_{a \in [K]}\sum_{j \in \mathcal{R}_a} \pi_{j,a} v_{j,a} v_{j,a}^\mathsf{T}$ is a block diagonal matrix, and the $a$-th diagonal block is exactly the matrix $\sum_{j \in \mathcal{R}_a} \pi_{j,a} e_{j,a} e_{j,a}^\mathsf{T}$. Meanwhile, the following relation also holds:

$$\sum_{a \in [K]} \mathrm{rank}\left(\sum_{j \in \mathcal{R}_a} \pi_{j,a} e_{j,a} e_{j,a}^\mathsf{T}\right) = \mathrm{rank}\left(\sum_{a \in [K]}\sum_{j \in \mathcal{R}_a} \pi_{j,a} v_{j,a} v_{j,a}^\mathsf{T}\right).$$

With those notations, we present a generalized version of Lemma 6 below. It is a "generalized version" in the sense that, the vectors $\{v_{i,a}\}_{i,a}$ involved in Lemma 7 can be arbitrary real vectors, which allows

more versatility in the formulation. Besides, in Lemma 7, $\sum_{a \in \mathcal{A}_i} \pi_{i,a}$ may be different for different client $i \in [M]$, which implies that the clients can work "asynchronously" instead of "synchronously" as under Fed-PE. Note that we change the order of the double summation, which removes $\mathcal{R}_a$ from the expressions.

**Lemma 7 (Generalized version of Lemma 6)** *Given any $v_{i,a} \in \mathbb{R}^n$, where $i \in [M], a \in \mathcal{A}_i$, and $\{\mathcal{A}_i\}$ are index sets. Consider the following optimization problems* (28) *and* (29)*:*

$$
\begin{cases}
\text{maximize } \bar{F}(\pi) = \log \text{Det} \left( \sum_{j \in [M]} \sum_{a \in \mathcal{A}_j} \pi_{j,a} v_{j,a} v_{j,a}^\mathsf{T} \right), \\
\text{subject to } \pi \in \bar{\mathcal{C}}.
\end{cases}
\tag{28}
$$

$$
\begin{cases}
\text{minimize } \bar{G}(\pi) = \sum_{i=1}^{M} f_i \max_{a \in \mathcal{A}_i} v_{i,a}^\mathsf{T} \left( \sum_{j \in [M]} \sum_{a \in \mathcal{A}_j} \pi_{j,a} v_{j,a} v_{j,a}^\mathsf{T} \right)^\dagger v_{i,a}, \\
\text{subject to } \pi \in \bar{\mathcal{C}}.
\end{cases}
\tag{29}
$$

*where the feasible set $\bar{\mathcal{C}} \subset \mathbb{R}^{\sum_{i \in [M]} |\mathcal{A}_i|}$ contains all $\pi$ that satisfy:*

$$
\bar{\mathcal{C}} = \left\{ \pi \, \middle| \, \begin{array}{l} \pi_{i,a} \geq 0, \forall i \in [M], a \in \mathcal{A}_i \\ \sum_{a \in \mathcal{A}_i} \pi_{i,a} = f_i, \forall i \in [M], \\ \text{rank}(\{\pi_{i,a} v_{i,a}\}_{i \in [M], a \in \mathcal{A}_i}) = \text{rank}(\{v_{i,a}\}_{i \in [M], a \in \mathcal{A}_i}) \end{array} \right\}.
\tag{30}
$$

*Then, we have the following equivalent statements:*

1) *$\pi^*$ is a maximizer of (28).*
2) *$\pi^*$ is a minimizer of (29).*
3) *$\bar{G}(\pi^*) = \text{rank}(\{v_{i,a}\}_{i \in [M], a \in \mathcal{A}_i}) := D$.*

*Proof.* 1) $\implies$ 2): Since $\bar{\mathcal{C}}$ is a convex set, and its image under a linear transformation is still convex, $\bar{F}(\pi)$ is a concave function over $\bar{\mathcal{C}}$ according to Lemma 3. Using the first-order optimality, we have

$$
\left\langle \nabla \bar{F}, \pi - \pi^* \right\rangle \leq 0
$$

holds for any $\pi \in \bar{\mathcal{C}}$. Therefore, based on Lemma 2, we have

$$
0 \geq \sum_{i \in [M]} \sum_{a \in \mathcal{A}_i} (\pi_{i,a} - \pi_{i,a}^*) v_{i,a}^\mathsf{T} \left( \sum_{j \in [M]} \sum_{a \in \mathcal{A}_j} \pi_{j,a}^* v_{j,a} v_{j,a}^\mathsf{T} \right)^\dagger v_{i,a}
\tag{31}
$$

$$
= \sum_{i \in [M]} \sum_{a \in \mathcal{A}_i} \pi_{i,a} v_{i,a}^\mathsf{T} \left( \sum_{j \in [M]} \sum_{a \in \mathcal{A}_j} \pi_{j,a}^* v_{j,a} v_{j,a}^\mathsf{T} \right)^\dagger v_{i,a}
$$

$$
- \text{trace} \left( \sum_{i \in [M]} \sum_{a \in \mathcal{A}_i} \pi_{i,a}^* v_{i,a} v_{i,a}^\mathsf{T} \left( \sum_{j \in [M]} \sum_{a \in \mathcal{A}_j} \pi_{j,a}^* v_{j,a} v_{j,a}^\mathsf{T} \right)^\dagger \right)
\tag{32}
$$

$$
= \sum_{i \in [M]} \sum_{a \in \mathcal{A}_i} \pi_{i,a} v_{i,a}^\mathsf{T} \left( \sum_{j \in [M]} \sum_{a \in \mathcal{A}_j} \pi_{j,a}^* v_{j,a} v_{j,a}^\mathsf{T} \right)^\dagger v_{i,a} - D.
\tag{33}
$$

Let $a_i = \arg\max_b v_{i,b}^\mathsf{T} \left( \sum_{j \in \mathcal{R}_b} \pi_{j,b}^* v_{j,b} v_{j,b}^\mathsf{T} \right)^\dagger v_{i,b}$, and set $\hat{\pi}_{i,a_i} = f_i$ and $\hat{\pi}_{i,b} = 0$ for any $b \neq a_i$. Then, define a sequence $\{\pi^m\}_{m=1}^{\infty} \in \bar{\mathcal{C}}$ with $\lim_{m \to \infty} \pi_m = \hat{\pi}$. Substituting $\pi^m$ into Eqn. (33) and taking the limit of $m$, we have

$$
\lim_{m \to \infty} \sum_{i \in [M]} \sum_{a \in \mathcal{A}_i} \pi_{i,a}^m v_{i,a}^\mathsf{T} \left( \sum_{j \in [M]} \sum_{a \in \mathcal{A}_j} \pi_{j,a}^* v_{j,a} v_{j,a}^\mathsf{T} \right)^\dagger v_{i,a} - D
\tag{34}
$$

$$
= \sum_{i \in [M]} \sum_{a \in \mathcal{A}_i} \hat{\pi}_{i,a} v_{i,a}^\mathsf{T} \left( \sum_{j \in [M]} \sum_{a \in \mathcal{A}_j} \pi_{j,a}^* v_{j,a} v_{j,a}^\mathsf{T} \right)^\dagger v_{i,a} - D
\tag{35}
$$

$$
= \sum_{i \in [M]} \hat{\pi}_{i,a_i} \max_{a \in \mathcal{A}_i} v_{i,a}^\mathsf{T} \left( \sum_{j \in [M]} \sum_{a \in \mathcal{A}_j} \pi_{j,a}^* v_{j,a} v_{j,a}^\mathsf{T} \right)^\dagger v_{i,a} - D \leq 0,
\tag{36}
$$

i.e.,

$$\sum_{i\in[M]} f_i \max_{a\in\mathcal{A}_i} v_{i,a}^\mathsf{T} \left(\sum_{j\in[M]}\sum_{a\in\mathcal{A}_j} \pi_{j,a}^* v_{j,a} v_{j,a}^\mathsf{T}\right)^\dagger v_{i,a} \leq D. \tag{37}$$

On the other hand, for any feasible point $\pi$, we have

$$D = \text{trace}\left(\sum_{i\in[M]}\sum_{a\in\mathcal{A}_i} \pi_{i,a} v_{i,a} v_{i,a}^\mathsf{T} \left(\sum_{j\in[M]}\sum_{a\in\mathcal{A}_j} \pi_{j,a} v_{j,a} v_{j,a}^\mathsf{T}\right)^\dagger\right) \tag{38}$$

$$= \sum_{i\in[M]}\sum_{a\in\mathcal{A}_i} \pi_{i,a} v_{i,a}^\mathsf{T} \left(\sum_{j\in[M]}\sum_{a\in\mathcal{A}_j} \pi_{j,a} v_{j,a} v_{j,a}^\mathsf{T}\right)^\dagger v_{i,a} \tag{39}$$

$$\leq \sum_{i\in[M]} f_i \max_{a\in\mathcal{A}_i} v_{i,a}^\mathsf{T} \left(\sum_{j\in[M]}\sum_{a\in\mathcal{A}_j} \pi_{j,a} v_{j,a} v_{j,a}^\mathsf{T}\right)^\dagger v_{i,a}. \tag{40}$$

Combining Eqns. (37) and (40), we conclude that $\pi^*$ is also a minimizer of $\bar{G}(\pi)$, and $\bar{G}(\pi^*) = D$.

2) $\implies$ 3) This can be seen from the above argument.

3) $\implies$ 1) For any feasible point $\pi \in \bar{\mathcal{C}}$, we have:

$$\langle \nabla \bar{F}, \pi - \pi^* \rangle \tag{41}$$

$$= \sum_{i\in[M]}\sum_{a\in\mathcal{A}_i} \pi_{i,a} v_{i,a}^\mathsf{T} \left(\sum_{j\in[M]}\sum_{a\in\mathcal{A}_j} \pi_{j,a}^* v_{j,a} v_{j,a}^\mathsf{T}\right)^+ v_{i,a} - D \tag{42}$$

$$\leq \bar{G}(\pi^*) - D = 0. \tag{43}$$

Thus, based on the concavity of $\bar{F}$, we conclude that $\pi^*$ is a maximizer of $\bar{F}(\pi)$. ∎

### C.2  Block Coordinate Ascent for Generalized G-optimal Design

In this subsection, we provide an efficient block coordinate ascent algorithm to solve the optimization problem in (27). We keep the same notations as in Appendix C.1, e.g., omitting the phase index $p$.

The block coordinate ascent algorithm inherits the idea from (Todd, 2016), which leverages the low-rank updating formula (Corollary A.10 in Todd (2016)). While only nonsingular matrices are considered in Todd (2016), we extend it to the case where pseudo-inverse and pseudo-determinant are involved. Such extension is necessary, because under Fed-PE, $\text{rank}(\{x_{i,a}\}_{i\in\mathcal{R}_a^p})$ is decreasing as phase $p$ progresses in general, making singular matrices unavoidable. As elaborated in this subsection, such extension is technically non-trivial.

**Lemma 8** *Assume $A \in \mathbb{R}^{n\times n}$ is a positive semi-definite matrix and vector $u \in \text{range}(A)$. Let $\lambda \in \mathbb{R}$ such that $\text{range}(A + \lambda uu^T) = \text{range}(A)$. Then, $\text{Det}_s(A + \lambda uu^\mathsf{T}) = (1 + \lambda u^\mathsf{T} A^\dagger u)\text{Det}_s(A)$ holds for any $s \in \{0, 1, \ldots, n\}$. Moreover,*

$$(A + \lambda uu^\mathsf{T})^\dagger = A^\dagger - \frac{\lambda A^\dagger uu^\mathsf{T} A^\dagger}{1 + \lambda u^\mathsf{T} A^\dagger u}.$$

*Proof.* Corollary A.10 in (Todd, 2016) states that for nonsingular matrix $B$ and any vector $u$ and $\lambda$ such that $B + \lambda uu^\mathsf{T}$ is also nonsingular, the following updating rules hold:

$$\det(B + \lambda uu^\mathsf{T}) = (1 + \lambda u^\mathsf{T} B^{-1} u)\det(B), \tag{44}$$

$$(B + \lambda uu^\mathsf{T})^{-1} = B^{-1} - \frac{\lambda B^{-1}uu^\mathsf{T} B^{-1}}{1 + \lambda u^\mathsf{T} B^{-1} u}. \tag{45}$$

For a general positive semi-definite matrix $A \in \mathbb{R}^{n\times n}$, we let $B := A + \epsilon I_n$, where $\epsilon > 0$ is an arbitrarily small number, and insert it in Eqn. (44). We have

$$\det(A + \epsilon I_n + \lambda uu^\mathsf{T}) = (1 + \lambda u^\mathsf{T}(A + \epsilon I_n)^{-1} u)\det(A + \epsilon I_n). \tag{46}$$

Dividing both sides by $\epsilon^{n-s}$ and letting $\epsilon$ go to 0, we have

$$\lim_{\epsilon \to 0} \frac{\det(A + \epsilon I_n + \lambda uu^\mathsf{T})}{\epsilon^{n-s}} = \lim_{\epsilon \to 0} \frac{(1 + \lambda u^\mathsf{T}(A + \epsilon I_n)^{-1}u)\det(A + \epsilon I_n)}{\epsilon^{n-s}}. \tag{47}$$

Given the fact that $u \in \text{range}(A + \lambda uu^\mathsf{T}) = \text{range}(A)$ and Proposition 1, we then have

$$\text{Det}_s(A + \lambda uu^\mathsf{T}) = (1 + \lambda u^\mathsf{T} A^\dagger u)\,\text{Det}_s(A). \tag{48}$$

In order to obtain the second part of Lemma 8, we first have

$$(A + \lambda uu^T)^\dagger = \lim_{\epsilon \to 0}(A + \epsilon I_n + \lambda uu^\mathsf{T})^{-1}(A + \lambda uu^\mathsf{T})(A + \epsilon I_n + \lambda uu^\mathsf{T})^{-1} \tag{49}$$

based on Proposition 1. Then, we use (45) to obtain

$$(A + \epsilon I_n + \lambda uu^\mathsf{T})^{-1} = (A + \epsilon I_n)^{-1} - \frac{\lambda(A + \epsilon I_n)^{-1}uu^\mathsf{T}(A + \epsilon I_n)^{-1}}{1 + \lambda u^\mathsf{T}(A + \epsilon I_n)^{-1}u}. \tag{50}$$

Inserting (45) in (49), using Proposition 1 to remove the limit, and manipulating the expression with the associative property of matrix multiplication, we finally have

$$(A + \lambda uu^\mathsf{T})^\dagger = A^\dagger - \frac{\lambda A^\dagger uu^\mathsf{T} A^\dagger}{1 + \lambda u^\mathsf{T} A^\dagger u}. \tag{51}$$

∎

The next two lemmas will be utilized to demonstrate that the coordinate ascent algorithm (Algorithm 3) described below does not violate the "rank-preserving" condition defined in Eqn. (5).

**Lemma 9** *Let* $A := \sum_{s=1}^d \lambda_s u_s u_s^\mathsf{T}$ *with* $u_s \in \mathbb{R}^n$, $\lambda_s > 0$ *for* $s \in [d]$. *Then,* $\text{range}(A) = \text{span}(\{u_s\}_{s \in [d]})$, *and* $\text{rank}(A) = \text{rank}(A^\dagger) = \text{rank}(\{\lambda_s u_s\}_{s \in [d]}) = \text{rank}(\{u_s\}_{s \in [d]})$.

*Proof.* It suffices to show the column space of $A$ is the same as the space spanned by $\{u_s, s \in [d]\}$. Note that

$$\text{range}(A) = \{v \in \mathbb{R}^n | v = Ac, c \in \mathbb{R}^n\},$$

and

$$\text{span}(\{u_s\}_{s \in [d]}) = \left\{ v \in \mathbb{R}^n \,\middle|\, v = \sum_{s \in [d]} u_s c_s, c_s \in \mathbb{R} \right\}.$$

Then, for any $v \in \text{range}(A)$, there must exist a $c \in \mathbb{R}^n$ such that $v = Ac = \sum_{s \in [d]} \lambda_s u_s u_s^\mathsf{T} c = \sum_{s \in [d]} (\lambda_s u_s^\mathsf{T} c)u_s \in \text{span}(\{u_s\}_{s \in [d]})$. Thus, we have $\text{range}(A) \subseteq \text{span}(\{u_s\}_{s \in [d]})$.

On the other hand, if $\text{range}(A) \neq \text{span}(\{u_s\}_{s \in [d]})$, there must exist a $v \in \text{span}(\{u_s\}_{s \in [d]})$, $v \neq 0$, such that $Av = 0$. Thus, we have $v^\mathsf{T} Av = 0$, which implies that

$$v^\mathsf{T} \left( \sum_{s=1}^d \lambda_s u_s u_s^\mathsf{T} \right) v = 0 \Leftrightarrow \sum_{s=1}^d \lambda_s (v^\mathsf{T} u_s)^2 = 0. \tag{52}$$

Since $\lambda_s > 0$ for $s \in [d]$, we must have $v^\mathsf{T} u_s = 0$ for all $s \in [d]$. Meanwhile, since $v \in \text{span}(\{u_s\}_{s \in [d]})$, it indicates that $v = 0$, which contradicts with the assumption that $v \neq 0$. ∎

**Lemma 10** *Assume* $A \in \mathbb{R}^{n \times n}$ *is a singular positive semi-definite matrix with* $d = \text{rank}(A) < n$. *Let* $u_0 \notin \text{range}(A)$. *Then, for any* $\lambda_0 > 0$, $u_0^\mathsf{T}(\lambda_0 u_0 u_0^\mathsf{T} + A)^\dagger u_0 = 1/\lambda_0$.

*Proof.* First, we decompose $A$ into $A = \sum_{s=1}^d \lambda_s u_s u_s^\mathsf{T}$, where $\{u_s\}_s$ are orthonormal eigenvectors of $A$, and $\{\lambda_s > 0\}_s$ are non-zero eigenvalues of $A$. Thus, $u_0 \notin \text{range}(A)$ indicates that $u_0, u_1, \ldots, u_d$ are linearly independent.

Since $X \succeq Y$ is equivalent to $X^\dagger \preceq Y^\dagger$, we have

$$u_t^\mathsf{T} \left( \sum_{s=0}^d \lambda_s u_s u_s^\mathsf{T} \right)^\dagger u_t \leq \frac{u_t^\mathsf{T}(u_t u_t^\mathsf{T})^\dagger u_t}{\lambda_t} = \frac{1}{\lambda_t} \tag{53}$$

holds for all $t \in \{0, 1, \ldots, d\}$.

Multiplying both sides of (53) by $\lambda_t$ and summing over $t$, we have

$$d + 1 \geq \sum_{t=0}^{d} \lambda_t u_t^\mathsf{T} \left( \sum_{s=0}^{d} \lambda_s u_s u_s^\mathsf{T} \right)^\dagger u_t = \mathrm{trace}\left( \left( \sum_{s=0}^{d} \lambda_s u_s u_s^\mathsf{T} \right)^\dagger \left( \sum_{t=0}^{d} \lambda_t u_t u_t^\mathsf{T} \right) \right) = d + 1.$$

Thus, Eqn. (53) must hold with equality for all $t \in \{0, 1, \ldots, d\}$. In particular, $u_0^\mathsf{T}(\lambda_0 u_0 u_0^\mathsf{T} + A)^\dagger u_0 = 1/\lambda_0$. ∎

---

**Algorithm 3** Block Coordinate Ascent (BCA)

---

1: **Input:** $\{\mathcal{A}_i\}_i, \{\mathcal{R}_a\}_a, \{e_{i,a}\}_{i,a}, \epsilon$.
2: For each $i \in [M], a \in \mathcal{A}_i$: $\pi_{i,a} \leftarrow \frac{1}{|\mathcal{A}_i|}$.
3: For each $a \in [K]$: $\tilde{V}_a \leftarrow \left( \sum_{i \in \mathcal{R}_a} \pi_{i,a} e_{i,a} e_{i,a}^\mathsf{T} \right)^\dagger$, $d_a \leftarrow \mathrm{rank}(\tilde{V}_a)$.
4: **while** $G(\pi) > \sum_{a \in [K]} d_a + \epsilon$ **do**
5:     **for** $i \in [M]$ **do** solve the following optimization problem:

$$\max_{\{\omega_a\}_{a \in \mathcal{A}_i}} \sum_{a \in \mathcal{A}_i} \log(1 + \omega_a e_{i,a}^\mathsf{T} \tilde{V}_a e_{i,a}), \quad \text{s.t.} \sum_{a \in \mathcal{A}_i} \omega_a = 0 \text{ and } -\pi_{i,a} \leq \omega_a \leq 1 - \pi_{i,a}. \quad (54)$$

6:         **for** $a \in \mathcal{A}_i$ **do**

$$\pi_{i,a} \leftarrow \pi_{i,a} + \omega_a, \quad \tilde{V}_a \leftarrow \tilde{V}_a - \frac{\omega_a \tilde{V}_a e_{i,a} e_{i,a}^\mathsf{T} \tilde{V}_a}{1 + \omega_a e_{i,a}^\mathsf{T} \tilde{V}_a e_{i,a}}.$$

7:         **end for**
8:     **end for**
9: **end while**

---

We are now ready to introduce the Block Coordinate Ascent (BCA) algorithm. As depicted in Algorithm 3, at each phase $p$, BCA aims to obtain a distribution $\pi$ by changing $\pi_i := \{\pi_{i,a}\}_{a \in \mathcal{A}_i}$ for a client $i$ while fixing the distributions for all the other clients $j \neq i$ in each step.

Consider that $\pi$ is changed to $\pi^+$ by redistributing $\pi_i$ to $\pi_i^+$. Thus, $\pi_i^+ = \pi_i + \omega$, where $\sum_{a \in \mathcal{A}_i} \omega_a = 0$, and $-\pi_{i,a} \leq \omega_a \leq 1 - \pi_{i,a}$.

Recall that

$$F(\pi) = \sum_{a \in [K]} \log \mathrm{Det}_{d_a} \left( \sum_{j \in \mathcal{R}_a} \pi_{j,a} e_{j,a} e_{j,a}^\mathsf{T} \right).$$

To simplify the notation, let $U_a := \sum_{j \in \mathcal{R}_a} \pi_{j,a} e_{j,a} e_{j,a}^\mathsf{T}$. Then, to maximize $F(\pi^+)$ is equivalent to maximizing the following:

$$F(\pi^+) - F(\pi) = \sum_{a \in \mathcal{A}_i} \log \frac{\mathrm{Det}_{d_a}\left( U_a + \omega_a e_{i,a} e_{i,a}^\mathsf{T} \right)}{\mathrm{Det}_{d_a}\left( U_a \right)} \quad (55)$$

$$= \sum_{a \in \mathcal{A}_i} \log \left( 1 + \omega_a e_{i,a}^\mathsf{T} U_a^\dagger e_{i,a} \right), \quad (56)$$

where Eqn. (56) follows from the low-rank updating formula (48) in Lemma 8 when $\mathrm{rank}(U_a + \omega_a e_{i,a} e_{i,a}^\mathsf{T}) = \mathrm{rank}(U_a)$.

After obtaining $\{\omega_a\}_{a \in \mathcal{A}_i}$, the distribution $\pi_i$ will be updated; Correspondingly, all involved $\{U_a^\dagger\}_{a \in \mathcal{A}_i}$ in the objective function (56) shall be updated as well. The low-rank updating formula (51) from Lemma 8 is then invoked to perform the updating efficiently. After that, BCA proceeds to the next block and repeats the procedure.

**Proposition 2** *The distribution vector $\pi$ after each block updating under BCA is always in $\mathcal{C}$.*

*Proof.* We prove this through induction. First, it is obvious that the initialization $\pi_{i,a} = \frac{1}{|\mathcal{A}_i|}$, for $i \in [M]$, $a \in \mathcal{A}_i$ ensures that $\pi_i$ is a valid distribution for every $i \in [M]$. Meanwhile, since $\pi_{i,a} = \frac{1}{|\mathcal{A}_i|}$ for $i \in [M]$, $a \in \mathcal{A}_i$, according to Lemma 9, we have $\text{rank}(U_a) = \text{rank}(\{e_{j,a}\}_{j \in \mathcal{R}_a})$. Thus, the rank-preserving condition is satisfied as well.

Next, we use $\pi$ and $\pi^+$ to denote the distribution before and after one block updating on $\pi_i$, respectively. Assume $\pi \in \mathcal{C}$, and we aim to show that $\pi^+ \in \mathcal{C}$ as well. It is straightforward to see that the conditions $\sum_{a \in \mathcal{A}_i} \omega_a = 0$ and $-\pi_{i,a} \le \omega_a \le 1 - \pi_{i,a}$ ensure that $\pi_i^+$ is still a valid distribution. It thus suffices to show that the rank-preserving condition is satisfied for $\pi^+$. We prove this through contradiction. Assume after the updating, $\text{rank}(U_a^+) < \text{rank}(U_a)$. We note that

$$U_a^+ = U_a + \omega_a e_{i,a} e_{i,a}^\intercal = (\pi_{i,a} + \omega_a)e_{i,a}e_{i,a}^\intercal + \sum_{j \in \mathcal{R}_a \setminus i} \pi_{j,a} e_{j,a} e_{j,a}^\intercal. \tag{57}$$

According to Lemma 9, if $\pi_{i,a} + \omega_a > 0$, we must have $\text{rank}(U_a^+) = \text{rank}(U_a)$. Thus, if $\text{rank}(U_a^+) < \text{rank}(U_a)$, we must have $\pi_{i,a} + \omega_a = 0$, and $e_{i,a} \notin \text{span}(\{e_{j,b}\}_{j \in \mathcal{R}_a \setminus \{i\}})$. Then, according to Lemma 10, we must have $e_{i,a}^\intercal U_a^\dagger e_{i,a} = 1/\pi_{i,a}$. Correspondingly, Eqn. (56) becomes $-\infty$. Since we can always perturb $\omega$ to make $\pi_{i,a}^+ \neq 0$ for every $a \in \mathcal{A}_i$, there exist feasible solutions making (56) greater than $-\infty$. Therefore, the distribution that makes $\text{rank}(U_a^+) < \text{rank}(U_a)$ cannot be the solution to (54) in the BCA algorithm. This indicates that the solution of BCA in each phase satisfies the rank-preserving condition. ∎

The convergence of BCA can be obtained by leveraging the result for the block coordinate minimization (BCM) algorithm introduced in (Calafiore and El Ghaoui, 2014). Given a function $f_0(x_1, \ldots, x_v)$ where $x_i \in \mathcal{X}_i$, $i = 1, \ldots, v$, the BCM algorithm generates $x^{(k)} := (x_1^{(k)}, \ldots, x_v^{(k)})$ as follows:

$$x_i^{(k+1)} = \arg\min_{y \in \mathcal{X}_i} f_0\left(x_1^{(k+1)}, \ldots, x_{i-1}^{(k+1)}, y, x_{i+1}^{(k)}, \ldots, x_v^{(k)}\right).$$

Then, the following theorem guarantees the convergence of the BCM algorithm.

**Theorem 4 (Theorem 12.4 in Calafiore and El Ghaoui (2014))** *Assume $f_0(x_1, \ldots, x_v)$ is convex and continuously differentiable on the feasible set $\mathcal{X} := \mathcal{X}_1 \otimes \ldots \otimes \mathcal{X}_v$. Moreover, let $f_0$ be strictly convex in $x_i$ when the other variable blocks $x_j, j \neq i$, are held constant. If the sequence $\{x^{(k)}\}$ generated by the BCM algorithm is well-defined, then every limit point of $\{x^{(k)}\}$ converges to an optimal solution of the optimization problem $\min_{x \in \mathcal{X}} f_0(x)$.*

Theorem 4 assures that every sequence $\{\pi^{(k)}\}$ generated by BCA converges to an optimal solution of (27), since $F(\pi^+) - F(\pi)$ is concave, continuously differentiable, and strictly concave in each $\pi_i$, $i \in [M]$ when the other $\pi_j$'s, $j \neq i$, are fixed.

## D  Proof of Theorem 1

We first present the complete version of Theorem 1.

**Theorem 5 (Complete version of Theorem 1)** *Consider time horizon $T$ that consists of $H$ phases with $f^p = cn^p$, where $c$ and $n > 1$ are fixed integers, and $n^p$ denotes the pth power of $n$. Let*

$$\alpha = \min\left\{\sqrt{2\log(KH/\delta) + d\log(ke)}, \sqrt{2\log(2MKH/\delta)}\right\}, \tag{58}$$

*where $k > 1$ is a number satisfying $kd \ge 2\log(KH/\delta) + d\log(ke)$. Then, with probability at least $1 - \delta$, the regret under Fed-PE is upper bounded as*

$$R(T) \le 4\alpha \frac{L}{\ell} \sqrt{dKM} \left(\frac{\sqrt{n^2 - n}}{\sqrt{n} - 1}\sqrt{T} + \frac{K}{\sqrt{cn} - \sqrt{c}}\right).$$

*Furthermore, by assuming $K = O(\sqrt{T})$, the cumulative regret scales as*

$$O\left(\frac{L}{\ell}\sqrt{dKMT(\log(K(\log T)/\delta) + \min\{d, \log M\})}\right)$$

*and the communication cost scales as $O(Md^2 K \log T)$.*

The proof of Theorem 5 relies on a "good" event that happens with high probability. We define the "bad" event as follows:

$$\mathcal{E}(\alpha) = \{\exists p \in [H], i \in [M], a \in \mathcal{A}_i^{p-1}, |\hat{r}_{i,a}^p - r_{i,a}| \geq u_{i,a}^p = \alpha \sigma_{i,a}^p\}, \tag{59}$$

where $\alpha$ is defined in (58) barring explicit explanations, and $\sigma_{i,a}^p := \|x_{i,a}\|_{V_a^p}/\ell$. We call $\mathcal{E}^c(\alpha)$ the "good" event.

## D.1 Bound the Probability of the Bad Event

Before we bound the probability of the bad event $\mathcal{E}(\alpha)$, let us first introduce some necessary notations and characterize the sub-Gaussianity of the locally estimated rewards.

Let $\mathcal{F}_p$ be the $\sigma$-algebra of the events happened in and before the arm elimination stage at phase $p$, i.e. $\mathcal{F}_p = \sigma\{a_{i,t} \in [K], y_{i,t} \in \mathbb{R} | t \in \cup_{q=0}^{p-1} \cup_{a \in \mathcal{A}_i^q} \mathcal{T}_{i,a}^q, i \in [M]\}$, with $\mathcal{F}_0 = \emptyset$. Note that $\mathcal{A}_i^p$ and $\mathcal{R}_a^p$ are $\mathcal{F}_p$-measurable.

Note that $\{a \in \mathcal{A}_i^p\}$ is equivalent to $\{i \in \mathcal{R}_a^p\}$. Throughout the analysis, we frequently exchange the order of double summations $\sum_{i \in [M]} \sum_{a \in \mathcal{A}_i^p}$ and $\sum_{a \in [K]} \sum_{i \in \mathcal{R}_a^p}$.

**Lemma 11** *At phase $p \in [H]$, for any client $i \in [M]$, arm $a \in \mathcal{A}_i^{p-1}$, $\hat{r}_{i,a}^p - r_{i,a}$ is a conditionally sub-Gaussian random variable, i.e. $\mathbb{E}[\exp(\lambda(\hat{r}_{i,a}^p - r_{i,a}))|\mathcal{F}_{p-1}] \leq \exp\left(\frac{\lambda^2(\sigma_{i,a}^p)^2}{2}\right)$.*

*Proof.* Let $\xi_{i,a}^{p-1}$ be the sum of the independent sub-Gaussian noise incurred during the collaborative exploration step in phase $p-1$, i.e. $\xi_{i,a}^{p-1} := \sum_{t \in \mathcal{T}_{i,a}^{p-1}} \eta_{i,t}$. Then, given $f_{i,a}^{p-1}$, $\xi_{i,a}^{p-1}$ is a conditionally $\sqrt{f_{i,a}^{p-1}}$-sub-Gaussian random variable.

Recall the definition of local estimators, and we have

$$\hat{\theta}_{i,a}^{p-1} = \left(\frac{1}{f_{i,a}^{p-1}} \sum_{t \in \mathcal{T}_{i,a}^{p-1}} y_{i,t}\right) \frac{x_{i,a}}{\|x_{i,a}\|^2} \tag{60}$$

$$= \left(x_{i,a}^\mathsf{T}\theta_a + \frac{\xi_{i,a}^{p-1}}{f_{i,a}^{p-1}}\right) \frac{x_{i,a}}{\|x_{i,a}\|^2} \tag{61}$$

$$= \frac{x_{i,a}x_{i,a}^\mathsf{T}}{\|x_{i,a}\|^2}\theta_a + \frac{x_{i,a}\xi_{i,a}^{p-1}}{f_{i,a}^{p-1}\|x_{i,a}\|^2}. \tag{62}$$

Since $\hat{r}_{i,a}^p = x_{i,a}^\mathsf{T}\hat{\theta}_a^p$, and $V_a^p = \left(\sum_{j \in \mathcal{R}_a^{p-1}} f_{j,a}^{p-1} \frac{x_{j,a}x_{j,a}^\mathsf{T}}{\|x_{j,a}\|^2}\right)^\dagger$, we have

$$\hat{r}_{i,a}^p - r_{i,a} = x_{i,a}^\mathsf{T}\hat{\theta}_a^p - x_{i,a}^\mathsf{T}\theta_a \tag{63}$$

$$= x_{i,a}^\mathsf{T}V_a^p\left(\sum_{j \in \mathcal{R}_a^{p-1}} f_{j,a}^{p-1}\hat{\theta}_{j,a}^{p-1}\right) - x_{i,a}^\mathsf{T}\theta_a \tag{64}$$

$$= x_{i,a}^\mathsf{T}V_a^p\left(\sum_{j \in \mathcal{R}_a^{p-1}} f_{j,a}^{p-1}\frac{x_{j,a}x_{j,a}^\mathsf{T}}{\|x_{j,a}\|^2}\theta_a + \sum_{j \in \mathcal{R}_a^{p-1}} \frac{x_{j,a}\xi_{j,a}^{p-1}}{\|x_{j,a}\|^2}\right) - x_{i,a}^\mathsf{T}\theta_a \tag{65}$$

$$= x_{i,a}V_a^p(V_a^p)^\dagger\theta_a + x_{i,a}^\mathsf{T}V_a^p\left(\sum_{j \in \mathcal{R}_a^{p-1}} \frac{e_{j,a}}{\|x_{j,a}\|}\xi_{j,a}^{p-1}\right) - x_{i,a}^\mathsf{T}\theta_a \tag{66}$$

$$= x_{i,a}^\mathsf{T}V_a^p\left(\sum_{j \in \mathcal{R}_a^{p-1}} \frac{e_{j,a}}{\|x_{j,a}\|}\xi_{j,a}^{p-1}\right), \tag{67}$$

where (66) is due to the definition of $V_a^p$ in Eqn. (4), and (67) due to the fact that $x_{i,a} \in \text{range}(V_a^p)$ and the property of pseudo-inverse specified in Definition 2.

Noe that Eqn. (67) is a linear combination of $\{\xi_{j,a}^{p-1}\}_{j \in \mathcal{R}_a^{p-1}}$. Thus, given $\mathcal{F}_p$, $\hat{r}_{i,a}^p - r_{i,a}$ is a conditionally sub-Gaussian random variable, whose parameter can be bounded as

$$\sum_{j \in \mathcal{R}_a^{p-1}} \left( x_{i,a}^\intercal V_a^p \frac{e_{j,a}}{\|x_{j,a}\|} \right)^2 f_{j,a}^{p-1} \leq \frac{1}{\ell^2} \sum_{j \in \mathcal{R}_a^{p-1}} x_{i,a}^\intercal V_a^p f_{j,a}^{p-1} e_{j,a} e_{j,a}^\intercal V_a^p x_{i,a} \tag{68}$$

$$= \frac{1}{\ell^2} x_{i,a}^\intercal V_a^p x_{i,a} = (\sigma_{i,a}^p)^2, \tag{69}$$

where Eqn. (68) comes from the bounded parameter assumption in Assumption 1 that $\|x_{i,a}\| \geq \ell$ for all $i, a$. ∎

Lemma 11 enables us to use concentration inequalities to bound the probability of the bad event $\mathcal{E}(\alpha)$ defined in (59) as follows.

**Lemma 12** *Under* `Fed-PE`*, we have* $\mathbb{P}[\mathcal{E}(\alpha)] \leq \delta$.

*Proof.* Let $\alpha = \min\{\alpha_1, \alpha_2\}$, where

$$\alpha_1 = \sqrt{2 \log(2MKH/\delta)}, \tag{70}$$

$$\alpha_2 = \sqrt{2 \log(KH/\delta) + d \log(ke)}. \tag{71}$$

As specified in Theorem 5, $k$ is a number that satisfies $k \geq \max\{\alpha_2^2/d, 1\}$. Note that this choice of $k$ ensures that $\alpha_2^2 \geq d$.

Based on the definition of $\mathcal{E}(\alpha)$ in (59), we have

$$\mathcal{E}(\alpha) = \mathcal{E}(\min\{\alpha_1, \alpha_2\}) \supset \mathcal{E}(\max\{\alpha_1, \alpha_2\}), \tag{72}$$

which implies that $\mathbb{P}[\mathcal{E}(\alpha)] = \max(\mathbb{P}[\mathcal{E}(\alpha_1)], \mathbb{P}[\mathcal{E}(\alpha_2)])$. Thus, it suffices to prove that $\mathbb{P}[\mathcal{E}(\alpha_i)] \leq \delta, \forall i \in \{1, 2\}$. In the following, we bound $\mathbb{P}[\mathcal{E}(\alpha_1)]$ and $\mathbb{P}[\mathcal{E}(\alpha_2)]$ separately.

*(i) Bound* $\mathbb{P}[\mathcal{E}(\alpha_1)]$. First, based on Lemma 11 and Hoeffding's inequality, we have

$$\mathbb{P}\left[|\hat{r}_{i,a}^p - r_{i,a}| \geq \alpha_1 \sigma_{i,a}^p | \mathcal{F}_{p-1}\right] \leq 2 \exp(-\alpha_1^2/2) = \frac{\delta}{MKH}. \tag{73}$$

Then, by applying the union bound,

$$\mathbb{P}[\mathcal{E}(\alpha_1)] = \mathbb{P}[\exists p \in [H], i \in [M], a \in \mathcal{A}_i^{p-1}, |\hat{r}_{i,a}^p - r_{i,a}| \geq \alpha_1 \sigma_{i,a}^p] \tag{74}$$

$$\leq \sum_{p \in [H]} \sum_{i \in [M]} \sum_{a \in \mathcal{A}_i^{p-1}} \mathbb{P}[|\hat{r}_{i,a}^p - r_{i,a}| \geq \alpha_1 \sigma_{i,a}^p | \mathcal{F}_{p-1}] \tag{75}$$

$$\leq HMK \frac{\delta}{MKH} = \delta. \tag{76}$$

*(ii) Bound* $\mathbb{P}[\mathcal{E}(\alpha_2)]$. Next, we aim to show that $\mathbb{P}[\mathcal{E}(\alpha_2)] \leq \delta$ is true. Our first observation is that

$$|\hat{r}_{i,a}^p - r_{i,a}| = \left| x_{i,a}^\intercal V_a^p \left( \sum_{j \in \mathcal{R}_a^{p-1}} \frac{e_{j,a}}{\|x_{j,a}\|} \xi_{j,a}^{p-1} \right) \right| \tag{77}$$

$$\leq \|x_{i,a}\|_{V_a^p} \cdot \left\| \sum_{j \in \mathcal{R}_a^{p-1}} \frac{e_{j,a}}{\|x_{j,a}\|} \xi_{j,a}^{p-1} \right\|_{V_a^p} \tag{78}$$

$$= \sigma_{i,a}^p \cdot \left\| \sum_{j \in \mathcal{R}_a^{p-1}} \frac{e_{j,a}}{\|x_{j,a}\|} \ell \xi_{j,a}^{p-1} \right\|_{V_a^p}, \tag{79}$$

where (78) is due to Cauchy-Schwarz inequality. Thus, if $\mathcal{E}(\alpha_2)$ happens, we must have that

$$\left\| \sum_{j \in \mathcal{R}_a^{p-1}} \frac{e_{j,a}}{\|x_{j,a}\|} \ell \xi_{j,a}^{p-1} \right\|_{V_a^p} \geq \alpha_2$$

hold for some phase $p$ and arm $a$.

We now analyze the term

$$X_{a,p} := \left\| \sum_{j \in \mathcal{R}_a^{p-1}} \frac{e_{j,a}}{\|x_{j,a}\|} \ell \xi_{j,a}^{p-1} \right\|_{V_a^p}^2$$

for a given phase $p$ and arm $a$. Note that

$$X_{a,p} = \sum_{i,j \in \mathcal{R}_a^{p-1}} \frac{\ell \xi_{i,a}^{p-1} e_{i,a}^{\mathsf{T}}}{\|x_{i,a}\|} \left( \sum_{k \in \mathcal{R}_a^{p-1}} f_{k,a}^{p-1} e_{k,a} e_{k,a}^{\mathsf{T}} \right)^{\dagger} \frac{\ell \xi_{j,a}^{p-1} e_{j,a}}{\|x_{j,a}\|}. \tag{80}$$

In the following, we aim to write $X_{a,p}$ in a matrix form. We note that $f_{i,a}^{p-1}$ may equal $0$ under the BCA algorithm, and if this happens, client $i$ does not pull arm $a$ during the collaborative exploration step in phase $p$, even though $a$ is in the active arm set. For this case, $\xi_{i,a}^{p-1} = 0$. Thus, we define a new random variable $\zeta_{i,a}$ as follows:

$$\zeta_{i,a} = \begin{cases} \dfrac{\ell \xi_{i,a}^{p-1}}{\sqrt{f_{i,a}^{p-1}} \|x_{i,a}\|}, & \text{if } f_{i,a}^{p-1} \neq 0, \\ 0, & \text{if } f_{i,a}^{p-1} = 0. \end{cases} \tag{81}$$

Then, $\{\zeta_{i,a}\}_{i \in \mathcal{R}_a^{p-1}}$ are conditionally independent 1-sub-Gaussian random variables.

Define vector $\zeta := (\zeta_{i,a})_{i \in \mathcal{R}_a^{p-1}}$ and matrix $A := (a_{i,j})_{i,j \in \mathcal{R}_a^{p-1}}$ where

$$a_{i,j} = \sqrt{f_{i,a}^{p-1}} e_{i,a}^{\mathsf{T}} \left( \sum_{k \in \mathcal{R}_a^{p-1}} f_{k,a}^{p-1} e_{k,a} e_{k,a}^{\mathsf{T}} \right)^{\dagger} e_{j,a} \sqrt{f_{j,a}^{p-1}}.$$

We can verify that $A$ is a symmetric matrix and

$$\sum_{k \in \mathcal{R}_a^{p-1}} a_{i,k} a_{k,j} = \sqrt{f_{i,a}^{p-1}} e_{i,a}^{\mathsf{T}} V_a^p \left( \sum_{k \in \mathcal{R}_a^{p-1}} f_{j,a}^{p-1} e_{k,a} e_{k,a}^{\mathsf{T}} \right) V_a^p e_{j,a} \sqrt{f_{j,a}^{p-1}}$$

$$= \sqrt{f_{i,a}^{p-1}} e_{i,a}^{\mathsf{T}} V_a^p (V_a^p)^{\dagger} V_a^p e_{j,a} \sqrt{f_{j,a}^{p-1}}$$

$$= \sqrt{f_{i,a}^{p-1}} e_{i,a}^{\mathsf{T}} V_a^p e_{j,a} \sqrt{f_{j,a}^{p-1}} = a_{i,j}.$$

Thus, $A^2 = A$, which implies that the eigenvalues of $A$ are either 1 or 0.

Meanwhile, we have

$$\text{trace}(A) = \sum_{i \in \mathcal{R}_a^{p-1}} a_{i,i} = V_a^p \sum_{i \in \mathcal{R}_a^{p-1}} f_{i,a}^{p-1} e_{i,a} e_{i,a}^{\mathsf{T}} = V_a^p (V_a^p)^{\dagger} = \text{rank}(V_a^p) = d_a^p.$$

Thus, the sum of the eigenvalues of $A$ is $d_a^p$. Combining with the fact that the eigenvalues of $A$ must be 1 or 0, we conclude that there are exactly $d_a^p$ eigenvalues equal to 1, and the rest eigenvalues are all 0. Therefore, $\text{rank}(A) = d_a^p \leq d$.

By the definition of $\zeta$ and $A^2 = A$, we have $X_{a,p} = \zeta^{\top} A \zeta = \|A\zeta\|^2$, where $\zeta$ is a conditionally 1-sub-Gaussian random vector, and $A$ is a $\mathcal{F}_{p-1}$-measurable matrix. Therefore, for any $\lambda \in (0, 1/2)$, we have

$$\mathbb{P}\left[ X_{a,p} \geq \alpha_2^2 | \mathcal{F}_{p-1} \right] = \mathbb{P}\left[ \|A\zeta\|^2 \geq \alpha_2^2 | \mathcal{F}_{p-1} \right] \tag{82}$$

$$= \mathbb{P}\left[ e^{\lambda \|A\zeta\|} \geq e^{\lambda \alpha_2^2} \big| \mathcal{F}_{p-1} \right] \tag{83}$$

$$\leq e^{-\lambda \alpha_2^2} \mathbb{E}\left[ e^{\lambda \|A\zeta\|} \big| \mathcal{F}_{p-1} \right] \tag{84}$$

$$\leq e^{-\lambda \alpha_2^2} \sqrt{\frac{1}{\det(I_d - 2\lambda A^2)}} \tag{85}$$

$$= e^{-\lambda\alpha_2^2}(1 - 2\lambda)^{-d_a^p/2} \tag{86}$$

$$\leq e^{-\lambda\alpha_2^2}(1 - 2\lambda)^{-d/2} \tag{87}$$

where (84) is due to Markov's inequality, (85) follows from Lemma 4, and (86) follows from the fact that the eigenvalues of $A$ are either 1 or 0 and there are exactly $d_a^p$ 1's.

By choosing $\lambda = \frac{\alpha_2^2 - d}{2\alpha_2^2} \in (0, \frac{1}{2})$, we have

$$\mathbb{P}\left[X_{a,p} \geq \alpha_2^2 | \mathcal{F}_{p-1}\right] \leq \left(\frac{\alpha_2^2}{d}\right)^{d/2} \exp\left(-\frac{\alpha_2^2 - d}{2}\right) \leq \frac{\delta}{KH}. \tag{88}$$

The last inequality is due to the following analysis:

$$\left(\frac{\alpha_2^2}{d}\right)^{d/2} \exp\left(-\frac{\alpha_2^2 - d}{2}\right) \leq \frac{\delta}{KH} \tag{89}$$

$$\Leftrightarrow \frac{d}{2}\left(2\log\alpha_2 - \log d\right) + \frac{d}{2} - \frac{1}{2}\left(\sqrt{2\log(KH/\delta) + d\log(ke)}\right)^2 \leq \log\left(\frac{\delta}{KH}\right) \tag{90}$$

$$\Leftrightarrow d\log\alpha_2 \leq \frac{d\log(dk)}{2} \tag{91}$$

$$\Leftrightarrow \alpha_2^2 \leq dk, \tag{92}$$

where (92) is assured by the definition of $k$.

Finally, the proof is completed by applying the union bound over phase $p$ and arm $a$. $\blacksquare$

### D.2 Bound the Regret under the Good Event

**Lemma 13** *If $\mathcal{E}^c(\alpha)$ occurs, we must have $a_i^* \in \mathcal{A}_i^p$, i.e. any optimal arm will never be eliminated.*

*Proof.* When $\mathcal{E}^c(\alpha)$ occurs, we have

$$\hat{r}_{i,\hat{a}_i}^p - \hat{r}_{i,a_i^*}^p \leq r_{i,\hat{a}_i} - r_{i,a_i^*} + u_{i,\hat{a}_i}^p + u_{i,a_i^*}^p \leq u_{i,\hat{a}_i}^p + u_{i,a_i^*}^p,$$

where $\hat{a}_i$ is the estimated optimal arm for client $i$ in phase $p$. Thus, under the elimination procedure in `Fed-PE`, $a_i^*$ will not be eliminated. $\blacksquare$

**Lemma 14** *If $\mathcal{E}^c(\alpha)$ occurs, the regret of `Fed-PE` in phase $p$ is upper bounded by $4\alpha\frac{L}{\ell}\sqrt{dKM}\frac{f^p + K}{\sqrt{f^{p-1}}}$.*

*Proof.* When $\mathcal{E}^c(\alpha)$ happens, $|\hat{r}_{i,a} - r_{i,a}| \leq u_{i,a}^p = \alpha\sigma_{i,a}^p = \frac{\alpha}{\ell}\|x_{i,a}\|_{V_a^p}$ holds for all $p \in [H], i \in [M], a \in \mathcal{A}_i^{p-1}$. Therefore, by the construction of $\mathcal{A}_i^p$, pulling any active arm $a \in \mathcal{A}_i^p$ in phase $p$ will incur a regret upper bounded by

$$\Delta_{i,a} = r_{i,a_i^*} - r_{i,a} \tag{93}$$

$$\leq \hat{r}_{i,\hat{a}_i} + u_{i,a_i^*}^p - \hat{r}_{i,a} + u_{i,a}^p \tag{94}$$

$$\leq u_{i,\hat{a}_i}^p + u_{i,a_i^*}^p + 2u_{i,a}^p \tag{95}$$

$$\leq 4\max_{a\in\mathcal{A}_i^p} u_{i,a}^p, \tag{96}$$

where (94) follows from the definition of $\mathcal{E}(\alpha)$, (95) follows from the arm elimination procedure under `Fed-PE`, and (96) is due to Lemma 13.

Under `Fed-PE`, the phase length is fixed as $f^p + K$ for $p \in [H]$. Then, given that the good event $\mathcal{E}^c(\alpha)$ happens, the total regret incurred during phase $p$, denoted as $R_p$, can be bounded as follows:

$$R_p \leq \sum_{i\in[M]} 4\max_{a\in\mathcal{A}_i^p} u_{i,a}^p(f^p + K) \tag{97}$$

$$\leq 4(f^p + K)\sqrt{M\sum_{i\in[M]}\max_{a\in\mathcal{A}_i^p}(u_{i,a}^p)^2} \tag{98}$$

$$\leq \frac{4\alpha L}{\ell}(f^p + K)\sqrt{M\sum_{i\in[M]}\max_{a\in\mathcal{A}_i^p}\frac{x_{i,a}^{\mathsf{T}}}{\|x_{i,a}\|_2}V_a^p\frac{x_{i,a}}{\|x_{i,a}\|_2}} \tag{99}$$

$$\leq \frac{4\alpha L}{\ell}(f^p + K)\sqrt{M\sum_{i\in[M]}\max_{a\in\mathcal{A}_i^p}e_{i,a}^{\mathsf{T}}\left(\sum_{j\in\mathcal{R}_a^p}\lceil\pi_{j,a}f^{p-1}\rceil e_{j,a}e_{j,a}^{\mathsf{T}}\right)^{\dagger}e_{i,a}} \tag{100}$$

$$\leq \frac{4\alpha L}{\ell}(f^p + K)\sqrt{\frac{M}{f^{p-1}}\sum_{i\in[M]}\max_{a\in\mathcal{A}_i^p}e_{i,a}^{\mathsf{T}}\left(\sum_{j\in\mathcal{R}_a^p}\pi_{j,a}e_{j,a}e_{j,a}^{\mathsf{T}}\right)^{\dagger}e_{i,a}} \tag{101}$$

$$\leq \frac{4\alpha L}{\ell}(f^p + K)\sqrt{\frac{dKM}{f^{p-1}}} \tag{102}$$

$$= \frac{4\alpha L}{\ell}\sqrt{dKM}\frac{f^p + K}{\sqrt{f^{p-1}}}. \tag{103}$$

where (98) is based on Cauchy-Schwarz inequality, (99) follows from the definition of $u_{i,a}^p := \alpha\|x_{i,a}\|_{V_a^p}/\ell$. We note that summation in (101) is exactly the solution to the multi-client G-optimal design in (6), which equals $\sum_a d_a^p \leq dK$ according to Lemma 6.

Note that the upper bound also holds for $p = 1$ when $f^0$ is defined as 1. This is because $V_a^0 := (\sum_{j\in[M]}e_{j,a}e_{j,a}^{\mathsf{T}})^{\dagger} \preceq (\sum_{j\in\mathcal{R}_a^p}\lceil\pi_{j,a}f^0\rceil e_{j,a}e_{j,a}^{\mathsf{T}})^{\dagger}$ for any $\pi$, although the central server does not utilize the G-optimal design to obtain $\pi^0$ during initialization. Thus, we still have (100) hold. ∎

**Corollary 1** Let $\sigma_{i,a}^p = \|x_{i,a}\|_{V_a^p}/\ell$. Then, under `Fed-PE`, for any $p \in [H]$, $a \in \mathcal{A}_i^p$, we have $\sum_{i\in[M]}\max_{a\in\mathcal{A}_i}(\sigma_{i,a}^p)^2 \leq \frac{dKL^2}{\ell^2 f^{p-1}}$.

*Proof.* Corollary 1 can be easily verified based on the fact that $\sum_{i\in[M]}\max_{a\in\mathcal{A}_i^p}(\sigma_{i,a}^p)^2 \leq \frac{L^2}{\ell^2}\sum_{i\in[M]}\max_{a\in\mathcal{A}_i^p}\frac{x_{i,a}^{\mathsf{T}}}{\|x_{i,a}\|_2}V_a^p\frac{x_{i,a}}{\|x_{i,a}\|_2}$ and the rest analysis follows the same argument as in (99)-(102). ∎

### D.3 Put Pieces Together

Finally, we are ready to prove Theorem 1. Recall the superscript $p$ of $n$ is the exponent. Then, with probability at least $1 - \delta$ and $n > 1$, the total regret over the $H$ phases can be bounded as:

$$R(T) = \sum_{p=1}^{H}R_p \leq \sum_{p=1}^{H}\frac{4\alpha L}{\ell}\sqrt{dKM}(\sqrt{c}n^{\frac{p+1}{2}} + Kn^{-\frac{p-1}{2}}/\sqrt{c}) \tag{104}$$

$$\leq \frac{4\alpha L}{\ell}\sqrt{dKM}\left(\sqrt{c}n\frac{n^{\frac{H+1}{2}} - \sqrt{n}}{\sqrt{n} - 1} + \frac{K}{\sqrt{c}n - \sqrt{c}}\right) \tag{105}$$

$$\leq \frac{4\alpha L}{\ell}\sqrt{dKM}\left(\frac{\sqrt{n^2 - n}}{\sqrt{n} - 1}\sqrt{cn\frac{n^H - 1}{n - 1}} + \frac{K}{\sqrt{c}n - \sqrt{c}}\right) \tag{106}$$

$$\leq \frac{4\alpha L}{\ell}\sqrt{dKM}\left(\frac{\sqrt{n^2 - n}}{\sqrt{n} - 1}\sqrt{T} + \frac{K}{\sqrt{c}n - \sqrt{c}}\right), \tag{107}$$

where (106) follows from that $n^{H/2} - 1 \leq \sqrt{n^H - 1}$, and (107) is due to the fact that

$$T = \sum_{p=1}^{H}f^p + KH \geq \sum_{p=1}^{H}cn^p = cn\frac{n^H - 1}{n - 1}.$$

Since

$$\alpha = O\left(\sqrt{\log(K(\log T)/\delta) + \min(d, \log M)}\right),$$

the regret scales in

$$O\left(\frac{L}{\ell}\sqrt{dKM(\log(K(\log T)/\delta) + \min(d, \log M)}\left(\sqrt{T} + K\right)\right).$$

When $K = O(\sqrt{T})$, the cumulative regret scales as

$$O\left(\frac{L}{\ell}\sqrt{dKMT(\log(K(\log T)/\delta) + \min(d, \log M)}\right).$$

**Remark 6** *We note the upload cost in Theorem 5 can be reduced by a factor of $d$ by sending scalars $\{\hat{\theta}_{i,a}^p/\bar{e}_{i,a}\}$ instead of vectors $\{\hat{\theta}_{i,a}^p\}$, as $\hat{\theta}_{i,a}^p$ is always in the same direction as $\bar{e}_{i,a}$. Similarly, for the download cost, if instead of broadcasting, the server calculates the projection of $\hat{\theta}_a^p$ along each direction $\bar{e}_{i,a}$, as well as $\|\bar{e}_{i,a}\|_{V_a^p}$, and send them back to client $i$, client $i$ can utilize those quantities instead of $\hat{\theta}_a^p$ and $V_a^p$ to obtain $\hat{r}_{i,a}^p$ and $u_{i,a}^p$. The corresponding download cost can then be reduced by a factor of $d^2$. Then, the overall communication cost would scale in $O(KM \log T)$.*

## E   Analysis of `Enhanced Fed-PE`

### E.1   Algorithm Details

As noted in Section 4.5, the motivation of `Enhanced Fed-PE` is to improve the efficiency of `Fed-PE` by leveraging all historical information to obtain more accurate estimates of $r_{i,a}$ and $u_{i,a}$ in each phase $p$. To achieve this goal, we keep the other parts of `Fed-PE` intact while only changing the arm elimination step as follows.

After calculating $\hat{r}_{i,a}^p$ according to Eqn. (2) in `Fed-PE` and obtaining $\sigma_{i,a}^p = \|x_{i,a}\|_{V_a^p}/\ell$, `Enhanced Fed-PE` aggregates estimates from previous phases to obtain a refined estimate of $r_{i,a}^p$, denoted as $\bar{r}_{i,a}^p$, and the corresponding confidence interval $\bar{u}_{i,a}^p$ as follows:

$$\bar{r}_{i,a}^p = \frac{\sum_{q=1}^p \hat{r}_{i,a}^q f^{q-1}}{\sum_{q=1}^p f^{q-1}}, \tag{108}$$

$$\bar{\sigma}_{i,a}^p = \sqrt{\frac{dK}{M} + \sum_{q=1}^p (\sigma_{i,a}^q)^2 (f^{q-1})^2}, \tag{109}$$

$$\alpha_{i,a}^p = \sqrt{\log\left(\frac{M^3 K(\bar{\sigma}_{i,a}^p)^2}{d\delta^2}\right)}, \tag{110}$$

$$\bar{u}_{i,a}^p = \frac{\alpha_{i,a}^p \bar{\sigma}_{i,a}^p}{\sum_{q=1}^p f^{q-1}}, \tag{111}$$

with $f^0 := 1$.

Denote $\bar{a}_i^p := \arg\max_{a \in \mathcal{A}_i^{p-1}} \bar{r}_{i,a}^p$. Then, the updating rule for the active arm set $\mathcal{A}_i^p$ is changed as follows

$$\mathcal{A}_i^p \leftarrow \left\{a \in \mathcal{A}_i^{p-1} \mid \bar{r}_{i,a}^p + \bar{u}_{i,a}^p \geq \bar{r}_{i,\bar{a}_i^p}^p - \bar{u}_{i,\bar{a}_i^p}^p\right\}. \tag{112}$$

### E.2   Performance of `Enhanced Fed-PE`

The performance of `Enhanced Fed-PE` is summarized in the following theorem.

**Theorem 6** *Let $S_{p-1} := \sum_{q=0}^{p-1} f^q$. Consider time horizon $T$ consisting of $H$ phases such that $HK \leq S_H$. Then, with probability at least $1 - \delta$, the total regret under* `Enhanced Fed-PE` *can be bounded as*

$$R(T) \leq \frac{4\sqrt{6}L}{\ell} \sum_{p=1}^{H} \frac{S_p - S_{p-1} + K}{\sqrt{S_{p-1}}} \sqrt{dKM \log \frac{LKMT}{\ell\delta}}. \tag{113}$$

*In particular, if $f^p \geq K$ and there exists a constant $c > 1$ such that $S_p \leq cS_{p-1}$ holds for all $p$, then, $\sum_{p=1}^{H} \frac{S_p - S_{p-1} + K}{\sqrt{S_{p-1}}} \leq 4\sqrt{cT}$, which indicates that*

$$R(T) \leq \frac{16\sqrt{6c}L}{\ell} \sqrt{dKMT \log \frac{LKMT}{\ell\delta}}.$$

In the following, we consider different selections of $f^p$ and evaluate the corresponding regret performance and communication costs. We fix the time horizon $T$, and assume $K \leq \sqrt{T}$. Note that $T = S_H + HK$.

**Uniform selection.** Let $f^1 = K - 1$, $f^p = K$, and $S_p = pK$. Let $H = \min\{p : 2pK \geq T\}$. We have

$$\sum_{p=1}^{H} \frac{S_p - S_{p-1} + K}{\sqrt{S_{p-1}}} = \sum_{p=1}^{H} \frac{2K}{\sqrt{pK}} \leq 2\sqrt{K} \int_0^{\frac{T}{2K}+1} \frac{1}{\sqrt{x}} dx = 4\sqrt{T/2 + K} \leq 4\sqrt{T}. \tag{114}$$

The communication cost with this phase length selection is $O(T/K)$.

**Exponential selection.** Choose $f^p = cn^p$, the same selection as in `Fed-PE`. Since $S_{p-1} \geq f^{p-1} = cn^{p-1}$, we have

$$\sum_{p=1}^{H} \frac{S_p - S_{p-1} + K}{\sqrt{S_{p-1}}} \leq \sum_{p=1}^{H} \frac{f^p + K}{\sqrt{f^{p-1}}} = O(\sqrt{T}), \tag{115}$$

and the communication cost is the same as that of `Fed-PE`, i.e. $O(\log T)$. We note that the regret under `Enhanced Fed-PE` scales in the same order as that under `Fed-PE`.

**Greedy selection.** Generate a sequence $\{\tilde{S}_p\}_{p=0}^{H}$ that satisfies the following equations

$$\tilde{S}_0 = 1, \quad \tilde{S}_p - \tilde{S}_{p-1} + K = 2\sqrt{T\tilde{S}_{p-1}}. \tag{116}$$

Since $K \leq \sqrt{T} \leq \sqrt{T\tilde{S}_{p-1}}$, we have

$$\tilde{S}_p + \sqrt{T\tilde{S}_{p-1}} \geq \tilde{S}_p - \tilde{S}_{p-1} + K = 2\sqrt{T\tilde{S}_{p-1}}. \tag{117}$$

Thus, $\tilde{S}_p \geq \sqrt{T\tilde{S}_{p-1}}$. Since $\tilde{S}_0 = 1$, $\tilde{S}_p \geq T^{1-\frac{1}{2^p}}$.

Let $H = \min\{p : \tilde{S}_p + pK \geq T\}$. In order to have $\tilde{S}_{H_0} + H_0 K \geq T$, we have

$$\tilde{S}_{H_0} + H_0 K \geq T \Leftrightarrow \sum_{p=2}^{H_0} (\tilde{S}_p - \tilde{S}_{p-1} + K) + \tilde{S}_1 + K \geq T \Leftarrow 2\sum_{p=2}^{H_0} \sqrt{T\tilde{S}_{p-1}} \geq T$$

$$\Leftarrow 2\sqrt{\tilde{S}_{H_0-1}} \geq \sqrt{T} \Leftarrow 2T^{\frac{1}{2}-\frac{1}{2^{H_0}}} \geq \sqrt{T} \Leftrightarrow \log 2 \geq \frac{1}{2^{H_0}} \log T \Leftarrow H_0 \geq \log_2 \log_2 T. \tag{118}$$

Thus, $H \leq \lceil \log_2 \log_2 T \rceil \in \{p : \tilde{S}_p + pK \geq T\}$.

Let $S_p = \lceil \tilde{S}_p \rceil$ for $p \in [H-1]$ and $S_H = T - HK$. Accordingly, $f^p = S_p - S_{p-1}$. Applying the equality in (117), we have

$$\sum_{p=1}^{H} \frac{S_p - S_{p-1} + K}{\sqrt{S_{p-1}}} = O(H\sqrt{T}) = O(\sqrt{T} \log\log T), \tag{119}$$

and communication cost under this choice of phase length is $O(\log \log T)$.

We summarize the results in Table 2. Note that we only list the scaling in $T$ and omit the other common factors in both regrets and communication costs. Among those three selections, exponential selection achieves the same regret performance as uniform selection, with a significantly reduced communication cost. On the other hand, the greedy selection achieves the lowest communication cost, with the price of slightly increased regret bound.

Table 2: Performance comparison with different phase length selection

| $f^p$ selection | Regret | Communication Cost |
|---|---|---|
| Uniform | $O(\sqrt{T \log T})$ | $O(T/K)$ |
| Exponential | $O(\sqrt{T \log T})$ | $O(\log T)$ |
| Greedy | $O(\sqrt{T \log T} \log \log T)$ | $O(\log \log T)$ |

### E.3 Proof of Theorem 6

Similar to the proof of Theorem 5, we define a "bad" event as follows:

$$\bar{\mathcal{E}} = \left\{ \forall p \in [H], i \in [M], a \in \mathcal{A}_i^{p-1}, |\bar{r}_{i,a}^p - r_{i,a}| \geq \bar{u}_{i,a}^p \right\}. \tag{120}$$

Then, we will first show that the probability of $\bar{\mathcal{E}}$ is upper bounded by $\delta$, and then analyze the regret when $\bar{\mathcal{E}}^c$ occurs.

**Lemma 15** *Under* `Enhanced Fed-PE`, *we have* $\mathbb{P}[\bar{\mathcal{E}}] \leq \delta$.

*Proof.* Lemma 11 has shown that each $\hat{r}_{i,a}^q - r_{i,a}$ is a conditionally sub-Gaussian random variable. Thus, $(\hat{r}_{i,a}^q - r_{i,a})f^{q-1}$ is a conditionally $\sigma_{i,a}^q f^{q-1}$-sub-Gaussian random variable. Letting $\sigma^2 = dK/M$ in Lemma 5, we have

$$\mathbb{P}\left[|\bar{r}_{i,a}^p - r_{i,a}| \geq \bar{u}_{i,a}^p | \mathcal{F}_{p-1}\right] \leq \frac{\delta}{MK}.$$

Lemma 15 then follows by applying the union bound over $i \in [M]$ and $a \in [K]$. ∎

**Lemma 16** *If* $\bar{\mathcal{E}}^c$ *happens, then the regret in phase* $p \in [H]$ *is upper bounded by* $\frac{4\sqrt{6}L}{\ell} \frac{f^p + K}{\sqrt{S_{p-1}}} \sqrt{dKM \log \frac{LKMT}{\ell\delta}}$.

*Proof.* First, we bound the terms $\bar{\sigma}_{i,a}^p$ and $\alpha_{i,a}^p$ defined in (109) and (111), respectively.

We note that Corollary 1 still holds under `Enhanced Fed-PE` due to the fact that the local estimates $\{\hat{\theta}_{i,a}^p\}$ only rely on the observations collected in phase $p$, the same as under `Ped-PE`.

Thus,

$$\bar{\sigma}_{i,a}^p = \sqrt{\frac{dK}{M} + \sum_{q=1}^p (\sigma_{i,a}^q)^2 (f^{q-1})^2} \tag{121}$$

$$\leq \sqrt{dK + \sum_{q=1}^p \frac{dKL^2(f^{q-1})^2}{f^{q-1}\ell^2}} \tag{122}$$

$$\leq \frac{L}{\ell}\sqrt{2dKT}. \tag{123}$$

Using (123), we can bound $\alpha_{i,a}^p$ as follows:

$$\alpha_{i,a}^p = \sqrt{\log \frac{M^3 K (\bar{\sigma}_{i,a}^p)^2}{d\delta^2}} \tag{124}$$

$$\leq \sqrt{\log \frac{2L^2 M^3 K^2 T}{\ell^2 \delta^2}} \tag{125}$$

$$\leq \sqrt{3 \log \frac{LKMT}{\ell \delta}} := \alpha_0. \tag{126}$$

Following similar argument as in Lemma 14, we have

$$R_p \leq \sum_{i \in [M]} 4 \max_{a \in \mathcal{A}_i^p} \bar{u}_{i,a}^p (f^p + K) \tag{127}$$

$$= 4(f^p + K) \sum_{i \in [M]} \max_{a \in \mathcal{A}_i^p} \alpha_{i,a}^p \frac{\bar{\sigma}_{i,a}^p}{S_{p-1}} \tag{128}$$

$$\leq 4\alpha_0 \frac{f^p + K}{S_{p-1}} \sum_{i \in [M]} \max_{a \in \mathcal{A}_i^p} \sqrt{\frac{dK}{M} + \sum_{q=1}^{p} (\sigma_{i,a}^q)^2 (f^{q-1})^2} \tag{129}$$

$$\leq 4\alpha_0 \frac{f^p + K}{S_{p-1}} \sum_{i \in [M]} \sqrt{\frac{dK}{M} + \sum_{q=1}^{p} \max_{a \in \mathcal{A}_i^q} (\sigma_{i,a}^q)^2 (f^{q-1})^2} \tag{130}$$

$$\leq \frac{4\alpha_0 (f^p + K)}{S_{p-1}} \sqrt{M \left( \frac{dK}{M} + \sum_{q=1}^{p} \sum_{i \in [M]} \max_{a \in \mathcal{A}_i^q} (\sigma_{i,a}^q)^2 (f^{q-1})^2 \right)}, \tag{131}$$

where in (129) we use $\alpha_0$ from Eqn. (126) to bound $\alpha_{i,a}^p$, and (131) follows from Cauchy-Schwarz inequality.

Then, we apply the result from Corollary 1 on all $q \in \{1, \dots, p\}$ and $R_p$ can be further bounded as

$$R_p \leq \frac{4\alpha_0 L}{\ell} \frac{f^p + K}{S_{p-1}} \sqrt{dK + M \sum_{q=1}^{p} \frac{dK(f^{q-1})^2}{f^{q-1}}} \tag{132}$$

$$\leq \frac{4\alpha_0 L}{\ell} \frac{f^p + K}{S_{p-1}} \sqrt{dK S_{p-1} + dKM S_{p-1}} \tag{133}$$

$$\leq \frac{4\sqrt{6} L}{\ell} \frac{f^p + K}{\sqrt{S_{p-1}}} \sqrt{dKM \log \frac{LKMT}{\ell \delta}}. \tag{134}$$

∎

Finally, we are ready to prove Theorem 6. The first part of the theorem follows directly from Lemma 16. The second part of Theorem 6 can be obtained by noticing that

$$\sum_{p=1}^{H} \frac{S_p - S_{p-1}}{\sqrt{S_{p-1}}} = \sum_{p=1}^{H} \left( \sqrt{\frac{S_p}{S_{p-1}}} + 1 \right) (\sqrt{S_p} - \sqrt{S_{p-1}}) \tag{135}$$

$$\leq 2\sqrt{c} \sum_{p=1}^{H} (\sqrt{S_p} - \sqrt{S_{p-1}}) \tag{136}$$

$$\leq 2\sqrt{cT}. \tag{137}$$

# F  Lower Bound Analysis

## F.1  Collinearly-dependent Policies

First, we state the definition of collinearity between a pair of vectors.

**Definition 4 (Collinear vectors)** *For a given set of vectors $\mathcal{X}$, two vectors $x, y \in \mathcal{X}$ are called* collinear *(denoted as $x \sim y$) if there exists a subset $S \subset \mathcal{X}$ such that the following conditions are satisfied: 1) $x \notin \mathrm{span}(S)$; and 2) $x \in \mathrm{span}(S \cup \{y\})$.*

Definition 4 indicates that two clients $i$ and $j$ are collinear if there exists an arm $a \in [K]$ such that the corresponding feacture vectors $x_{i,a}$ and $x_{j,a}$ are collinear given $\{x_{i,a}\}_{i \in [M]}$.

**Proposition 3** *The collinear relation on a given set of vectors $\mathcal{X}$ is an equivalence relation, i.e., for any $x, y, z \in \mathcal{X}$, we have*

1. Reflexivity: $x \sim x$.
2. Symmetry: *If $x \sim y$, then we must have $y \sim z$.*
3. Transitivity: *If $x \sim y$ and $y \sim z$, then, we must have $y \sim z$.*

*Proof.* We prove those three properties one by one in the following.

- *Proof of Reflexivity.* The reflexivity is obvious when we set $S = \emptyset$.
- *Proof of Symmetry.* Assume for a subset $S \subset \mathcal{X}$, we have $x \notin \mathrm{span}(S)$ and $x \in \mathrm{span}(S \cup \{y\})$. This implies that

$$y \notin \mathrm{span}(S). \tag{138}$$

  Meanwhile, we have

$$\mathrm{span}(S) \subset \mathrm{span}(S \cup \{x\}) \subseteq \mathrm{span}(S \cup \{y\}). \tag{139}$$

Since $\dim(\mathrm{span}(S \cup \{y\})) = \dim(\mathrm{span}(S)) + 1$, we must have

$$\mathrm{span}(S \cup \{x\}) = \mathrm{span}(S \cup \{y\}), \tag{140}$$

  which indicates that

$$y \in \mathrm{span}(S \cup \{x\}). \tag{141}$$

  Therefore, combining (138) and (141), we must have $y \sim x$.
- *Proof of Transitivity.* We consider the following possible cases.
  a) $\mathrm{span}(x), \mathrm{span}(y), \mathrm{span}(z)$ are not distinct. Following the proof of reflexivity and the symmetry property, we can easily show that $x \sim z$.
  b) $\mathrm{span}(x), \mathrm{span}(y), \mathrm{span}(z)$ are all distinct, and $\dim(\mathrm{span}(\{x, y, z\})) = 2$. For this case, we let $S = \{y\}$. Then, since $\mathrm{span}(x), \mathrm{span}(y), \mathrm{span}(z)$ are all distinct, we must have $x \notin \mathrm{span}(S)$. On the other hand, since $\dim(\mathrm{span}(\{x, y, z\})) = 2$, we must have $\mathrm{span}(\{x, y, z\}) = \mathrm{span}(\{y, z\})$. This implies that

$$x \in \mathrm{span}(\{x, y, z\}) = \mathrm{span}(\{y, z\}) = \mathrm{span}(S \cup \{z\}). \tag{142}$$

  Therefore, we must have $x \sim z$.
  c) $x, y, z$ are linearly independent. Let $S, T \subset \mathcal{X}$ be the minimal subsets such that $x \notin \mathrm{span}(S)$, $x \in \mathrm{span}(S \cup \{y\})$, and $z \notin \mathrm{span}(T)$, $z \in \mathrm{span}(T \cup \{y\})$. Then, we have the following subcases:
  c1) $y \notin \mathrm{span}(S \cup T)$. This implies that

$$x \notin \mathrm{span}(S \cup T). \tag{143}$$

  This is because $y \sim x$, thus $y \in \mathrm{span}(S \cup \{x\})$. If $x \in \mathrm{span}(S \cup T)$, then we must have $y \in \mathrm{span}(S \cup (S \cup T)) = \mathrm{span}(S \cup T)$.
  On the other hand, since $y \sim z$, we have $y \in \mathrm{span}(T \cup \{z\})$, which indicates that

$$x \in \mathrm{span}(S \cup \{y\}) \subset \mathrm{span}(S \cup T \cup \{z\}) = \mathrm{span}((S \cup T) \cup \{z\}). \tag{144}$$

  Combining (143) and (144), we have $x \sim z$.
  c2) $y \in \mathrm{span}(S \cup T)$. Since we assume both $S$ and $T$ are minimal, then, for any $s \in S$, $x \sim s$ as well, as

$$x \notin \mathrm{span}((S \backslash \{s\}) \cup \{y\}), \quad x \in \mathrm{span}(S \cup \{y\}). \tag{145}$$

  Let $u$ be a vector in $S$, and define $S_1 := S \backslash \{u\}$. For the remaining vectors in $S_1$, label them as $s_1, s_2, \ldots, s_m$. Then, we set $\mathcal{B}_1 := \{x, y, s_1, \ldots, s_m\}$ as the basis for $\mathrm{span}(S_1 \cup \{x, y\})$. Note that those vectors in $\mathcal{B}_1$ are now unit vectors with respect to $\mathcal{B}_1$.

Since $x \sim u$, we have $u \in \mathrm{span}(S_1 \cup \{x, y\})$. Denote the coordinates of $u$ relative to basis $\mathcal{B}_1$ as $[u]_{\mathcal{B}_1} := (u_1, u_2, \ldots, u_{m+2})^T$. Then, we must have $u_i \neq 0$ for any $i$, as otherwise $S$ cannot be minimal. This also implies that replacing any vector in $\mathcal{B}_1$ by $u$ also form a basis for $\mathrm{span}(S_1 \cup \{x, y\})$. We further consider two possible cases.

c2-i) $z \in \mathrm{span}(S \cup \{y\})$. Since $\mathrm{span}(S \cup \{y\}) = \mathrm{span}(S_1 \cup \{u, y\}) = \mathrm{span}(S_1 \cup \{x, y\})$, we can express the coordinates of $z$ relative to the selected basis $\mathcal{B}_1$ as $[z]_{\mathcal{B}_1} := (z_1, z_2, \ldots, z_{m+2})$. Let $i^*$ be the first non-zero coordinate with $z_{i^*} \neq 0$. Then, define a vector $w$ as

$$w = \begin{cases} u, & i^* = 1, \\ y, & i^* = 2, \\ s_{i^*-2}, & \text{otherwise.} \end{cases} \tag{146}$$

and $W = (S \cup \{y\}) \backslash \{w\}$.

Then, $x \notin \mathrm{span}(W)$ according to (145), and $x \in \mathrm{span}(S \cup \{y\}) = \mathrm{span}(W \cup \{z\})$ based on the definition of $w$ in (146). Thus, $x \sim z$.

c2-ii) $z \notin \mathrm{span}(S \cup \{y\})$. Assume $\dim(\mathrm{span}(S \cup T)) = n$. Since $z \in \mathrm{span}(T \cup \{y\})$, $y \in \mathrm{span}(S \cup T)$, we must have $z \in \mathrm{span}(S \cup \{y\} \cup T) = \mathrm{span}(S \cup T)$. While $\dim(\mathrm{span}(S \cup \{y\}))$ equals $m + 2$ as shown before, when $z$ is included, we must have $n > m + 2$. Set $\mathcal{B}_2 = \{x, y, s_1, s_2, \ldots, s_m, z, t_1, \ldots, t_{n-m-3}\}$, where $t_i$s are vectors from $T$ that are linearly independent with the other vectors in $\mathcal{B}_2$. Then $\mathcal{B}_2$ form a basis for $\mathrm{span}(S \cup T)$, and all the vectors in $\mathcal{B}_2$ are unit vectors with respect to $\mathcal{B}_2$. Denote $T_1 = \{t_1, \ldots, t_{n-m-3}\}$.

Since $z \in \mathrm{span}(\{y\} \cup T)$ and $z \notin \mathrm{span}(\{y\} \cup T_1)$, there must exist a vector $v \in T \backslash T_1$, whose coordinate vector with respect to $\mathcal{B}_2$ is denoted as $[v]_{\mathcal{B}_2} := (v_1, v_2, \ldots, v_n)$ with $v_{m+3} \neq 0$. If $v_1 \neq 0$, let $W = S_1 \cup T_1 \cup \{y, v\}$. Note that $S_1 \cup T_1 \cup \{y\}$ contains unit vectors in $\mathcal{B}$ except $x$ and $z$, while $v$ contains non-zero components along the dimension spanned by $x$ and $z$. Therefore, $x \notin \mathrm{span}(W)$, while $x \in \mathrm{span}(W \cup \{z\}) = \mathrm{span}(S \cup T)$. Thus, $x \sim z$.

If $v_1 = 0$, then, $v_2, v_3, \ldots, v_{2+m}$ cannot be all zero. Otherwise, $z \in \mathrm{span}(T_1 \cup \{v\}) \subset \mathrm{span}(T)$, which contradicts the assumption that $z \notin \mathrm{span}(T)$.

Let $i^*$ be the smallest index with $v_{i^*} \neq 0$, and

$$w = \begin{cases} y, & i^* = 2, \\ s_{i^*-2}, & \text{otherwise.} \end{cases} \tag{147}$$

Let $W = ((S \cup \{y\}) \backslash \{w\}) \cup T_1 \cup \{v\}$. Compared with $\mathcal{B}_2$, $W$ does not contain $x, z$ but $u, v$; Besides, one vector in $S \cup \{y\}$ is removed. If $x \in \mathrm{span}(W)$, then $\mathrm{span}(W) = \mathrm{span}(((S \cup \{x, y\}) \backslash \{w\}) \cup T_1 \cup \{v\}) = \mathrm{span}((S \cup \{x, y\}) \cup T_1 \cup \{v\}) = \mathrm{span}(S \cup T)$. However, $\dim(W) = n - 1$, which contradicts with the assumption that $\dim(\mathrm{span}(S \cup T)) = n$. Thus, we must have $x \notin \mathrm{span}(W)$.

Next, to show that $x \sim z$, it suffices to show that $x \in \mathrm{span}(W \cup \{z\})$. Through linear algebra, we can verify that $\mathrm{span}(W \cup \{z\}) = \mathrm{span}(S \cup T)$. Since $x \in \mathrm{span}(S \cup T)$, the proof is thus complete.

■

**Lemma 17** *For any given arm $a \in [K]$, partition feature vectors $\{x_{i,a}\}_{i \in [M]}$ into $L$ equivalence classes denoted as $S_1, S_2, \ldots, S_L$ based on the collinear relation. Then, different classes are linearly independent, i.e.,*

$$\mathrm{span}(S_1 \cup S_2 \ldots \cup S_i) \cap \mathrm{span}(S_{i+1}) = \{0\} \tag{148}$$

*for any $i = 1, \ldots, L - 1$.*

*Proof.* We prove it through contradiction. First, we assume $\dim(\mathrm{span}(S_1 \cup S_2 \ldots \cup S_i)) = n$, and assume $\{e_1, e_2, \ldots, e_n\} \subseteq \cup_{j=1}^{i} S_j$ is a basis for it.

Assume $\mathrm{span}(S_1 \cup S_2 \ldots \cup S_i) \cap \mathrm{span}(S_{i+1}) \neq \{0\}$. Then, there must exist $u \in \mathrm{span}(S_{i+1})$, $u \neq 0$, such that $u = \sum_i u_i e_i$. Then, we have $u \in \mathrm{span}(\{e_i | u_i \neq 0\})$. However, $u \notin \mathrm{span}(\{e_i | u_i \neq 0\} - \{e_j\})$ for any $j$ such that $u_j \neq 0$. Thus, $u \sim e_j$ for any $j$ such that $u_j \neq 0$. Based on Proposition 3, we must have $S_{i+1}$ equivalent to at least one of $S_1, S_2, \ldots, S_i$, which contradicts with the assumption on those equivalence classes.

Therefore, we must have $\text{span}(S_1 \cup S_2 \ldots \cup S_i) \cap \text{span}(S_{i+1}) = \{0\}$. Note that the labeling of the classes can be arbitrary, which indicates the property holds for any groups of the equivalence classes. ∎

**Lemma 18** *Let $X \in \mathbb{R}^{d \times m}$ and $Y \in \mathbb{R}^{d \times n}$ be two matrices such that $\text{span}(X) \cap \text{span}(Y) = \{0\}$. Then,*

$$u^{\mathsf{T}}(XX^{\mathsf{T}} + YY^{\mathsf{T}})^{\dagger}v = \begin{cases} u^{\mathsf{T}}(XX^{\mathsf{T}})^{\dagger}v, & \text{if } u, v \in \text{span}(X), \\ 0, & \text{if } u \in \text{span}(X), v \in \text{span}(Y), \\ u^{\mathsf{T}}(YY^{\mathsf{T}})^{\dagger}v, & \text{if } u, v \in \text{span}(Y). \end{cases} \quad (149)$$

*Proof.* Since $\text{span}(X) \cap \text{span}(Y) = \{0\}$, there exists an invertible matrix $A \in \mathbb{R}^{d \times d}$ such that

$$\text{span}(AX) = \text{span}(e_1, e_2, \ldots, e_{d_X}), \quad (150)$$
$$\text{span}(AY) = \text{span}(e_{d_X+1}, e_{d_X+2}, \ldots, e_{d_X+d_Y}), \quad (151)$$

where $d_X := \dim(\text{span}(X))$ and $d_Y := \dim(\text{span}(Y))$, and $e_i$'s are unit vectors in $\mathbb{R}^d$.

Then,

$$u^{\mathsf{T}}(XX^{\mathsf{T}} + YY^{\mathsf{T}})^{\dagger}v = (Au)^{\mathsf{T}}((AX)(AX)^{\mathsf{T}} + (AY)(AY)^{\mathsf{T}})^{\dagger}(Av). \quad (152)$$

Note that $(AX)(AX)^{\mathsf{T}}$ and $(AY)(AY)^{\mathsf{T}}$ are now block diagonal matrices whose non-zero blocks do not overlap with each other. To be more specific, they are in the following forms

$$(AX)(AX)^{\mathsf{T}} = \begin{bmatrix} \star & 0 & 0 \\ 0 & 0 & 0 \\ 0 & 0 & 0 \end{bmatrix}, \qquad (AY)(AY)^{\mathsf{T}} = \begin{bmatrix} 0 & 0 & 0 \\ 0 & \star & 0 \\ 0 & 0 & 0 \end{bmatrix}, \quad (153)$$

where $\star$ represents non-zero blocks.

Thus, if $u, v \in \text{span}(X)$, $Au$, $Av$ must be in the form of $\begin{bmatrix} \star \\ 0 \\ 0 \end{bmatrix}$, which verifies that

$$(Au)^{\mathsf{T}}((AX)(AX)^{\mathsf{T}} + (AY)(AY)^{\mathsf{T}})^{\dagger}(Av) = (Au)^{\mathsf{T}}((AX)(AX)^{\mathsf{T}})^{\dagger}(Av) = u^{\mathsf{T}}(XX^{\mathsf{T}})^{\dagger}v.$$

Similarly, if $u, v \in \text{span}(Y)$, we have

$$(Au)^{\mathsf{T}}((AX)(AX)^{\mathsf{T}} + (AY)(AY)^{\mathsf{T}})^{\dagger}(Av) = (Au)^{\mathsf{T}}((AY)(AY)^{\mathsf{T}})^{\dagger}(Av) = u^{\mathsf{T}}(YY^{\mathsf{T}})^{\dagger}v.$$

If $u \in \text{span}(X)$ and $v \in \text{span}(Y)$, $Au$, $Av$ must be in the form of $\begin{bmatrix} \star \\ 0 \\ 0 \end{bmatrix}$ and $\begin{bmatrix} 0 \\ \star \\ 0 \end{bmatrix}$, respectively. Thus, we must have

$$(Au)^{\mathsf{T}}((AX)(AX)^{\mathsf{T}} + (AY)(AY)^{\mathsf{T}})^{\dagger}(Av) = 0.$$

∎

**Remark 7** *We point out that Lemma 10 can be treated as a special instance of Lemma 18.*

**Theorem 7** *For the federated linear contextual bandits considered in this work, partition clients into equivalence classes where two clients $i$ and $j$ are in the same class if there exists an arm $a \in [K]$ such that $x_{i,a} \sim x_{j,a}$. Let $S$ be the class that includes client $i$, and $X \in \mathbb{R}^{d \times m}$ be a matrix with $\text{span}(X) \subseteq \text{span}(\{x_{j,a}\}_{j \in S})$. Let $Y \in \mathbb{R}^{d \times n}$ be another matrix with $\text{span}(X) \cap \text{span}(Y) = \{0\}$. Let $u_X, v_X \in \text{span}(X)$ and $v_Y \in \text{span}(Y)$. Then, for a given policy $\pi$, if the local estimates for any arm $a \in [K]$ are in the following forms*

$$\hat{r}_{i,a} = u_X^T(XX^{\mathsf{T}} + YY^{\mathsf{T}})^{\dagger}(v_X + v_Y) \quad (154)$$
$$\hat{\sigma}_{i,a} = u_X^T(XX^{\mathsf{T}} + YY^{\mathsf{T}})^{\dagger}u_X \quad (155)$$

*and the decisions are made based solely on those quantities besides other system parameters, then $\pi$ is a collinearly-dependent policy.*

*Proof.* Based on Lemma 18, we can show that

$$\hat{r}_{i,a} = u_X^T (XX^\intercal + YY^\intercal)^\dagger (v_X + v_Y) \tag{156}$$

$$= u_X^T (XX^\intercal + YY^\intercal)^\dagger v_X + u_X^T (XX^\intercal + YY^\intercal)^\dagger v_Y \tag{157}$$

$$= u_X^T (XX^\intercal)^\dagger v_X, \tag{158}$$

$$\sigma_{i,a} = u_X^T (XX^\intercal + YY^\intercal)^\dagger u_X = u_X^T (XX^\intercal)^\dagger u_X. \tag{159}$$

Since $\hat{r}_{i,a}$ and $\hat{\sigma}_{i,a}$ only depend on the feature vectors associated with clients in the same class $S$, and the decisions under $\pi$ are based on those estimates only, $\pi$ is a collinearly-dependent policy. ∎

**Corollary 2** `Fed-PE` *and* `Enhanced Fed-PE` *are both collinearly-dependent policies.*

Corollary 2 can be easily verified by checking the expressions of $\hat{r}_{i,a}^p$ and $\sigma_{i,a}^p$ under `Fed-PE` and `Enhanced Fed-PE`, respectively.

**Corollary 3** *LinUCB type of policies are collinearly-dependent policies if we include $d$ dummy clients with $d$ unit feature vectors in $\mathbb{R}^d$.*

*Proof.* For single-agent sequential linear contextual bandits with disjoint parameters, the LinUCB algorithm (Li et al., 2010) works as follows: at time $t$, for any incoming context $c_t = i$, the learner obtains estimated reward and uncertainty for each arm $a \in [K]$ as follows:

$$\hat{r}_{i,a}(t) = x_{i,a}^\intercal V_a^{-1}(t) \left( \sum_{s=1}^{t-1} \mathbb{1}\{a_s = a\} x_{c_s,a} y_s \right) \tag{160}$$

$$\hat{\sigma}_{i,a}(t) = \|x_{i,a}\|_{V_a^{-1}(t)}, \tag{161}$$

where

$$V_a(t) = \sum_{s=1}^{t-1} x_{c_s,a} x_{c_s,a}^\intercal \mathbb{1}\{a_s = a\} + \lambda I_d. \tag{162}$$

It then picks the arm with the highest upper confidence bound $\hat{r}_{i,a}(t) + \alpha \hat{\sigma}_{i,a}(t)$, where $\alpha$ is a real number.

If we view each context to be associated with a fixed client as in our federated linear contextual bandits setting, and treat the identity matrix $I_d$ as $\sum_{i=1}^d e_i e_i^\intercal$ where $e_i$'s are unit feature vectors associated with $d$ dummy clients, then, the estimate at client $i$ for arm $a$ only depends on clients with $x_{j,a} \sim x_{i,a}$. Thus, the decision at client $i$ only depends on other clients in the same equivalent class, and such LinUCB type of policies are collinearly-dependent. ∎

**Corollary 4** *Thompson sampling based policies with Gaussian priors are collinearly-dependent policies.*

*Proof.* To make the setting consistent, we first modify the Thompson sampling algorithm proposed for the shared parameter linear contextual bandits (Agrawal and Goyal, 2013b) to a disjoint parameter setting.

Similar to LinUCB, the learner would calculate the mean and variance of the Gaussian distribution for the parameter $\theta_a$ associated with each arm $a \in [K]$ according to

$$\hat{\theta}_a(t) = V_a^{-1}(t) \left( \sum_{s=1}^{t-1} \mathbb{1}\{a_s = a\} x_{c_s,a_s} y_s \right), \tag{163}$$

$$\hat{\sigma}_{i,a}(t) = \alpha^2 V_a^{-1}(t), \tag{164}$$

where $\alpha \in \mathbb{R}$ is a constant and $V_a^{-1}(t)$ is defined in the same way as in (162). It then samples $\theta_a(t)$ from the distribution $\mathcal{N}(\hat{\theta}_a(t), \hat{\sigma}_{i,a}(t))$, and then plays the arm $a_t := \arg\max_a x_{i,a}^\intercal \theta_a(t)$ if the incoming context $c_t = i$.

We note that under the Thompson sampling procedure, the reward associated with arm $a$ and incoming context $i$ at time $t$ is actually a Gaussian random variable with distribution $\mathcal{N}\left(x_{i,a}^{\mathsf{T}} V_a^{-1}(t)\left(\sum_{s=1}^{t-1} \mathbb{1}\{a_s = a\} x_{c_s,a_s} y_s\right), \alpha^2 x_{i,a}^{\mathsf{T}} V_a^{-1}(t) x_{i,a}\right)$. According to Theorem 7, since the mean and variance are in the form specified in (154)-(155), the distribution of the reward under the incoming context $i$ only depends on other contexts that in the same equivalence class.

If we associate each context with a fixed client, then, we can see that the decision at client $i$ only depends on other clients in the same equivalence class. Thus, such Thompson sampling type of policies are collinearly-dependent. ∎

### F.2 Proof of Theorem 2

We provide the detailed proof for the minimax regret bound in Theorem 2. For simplicity, we assume $M/d$ is an integer. We use $\mathbf{1}$ and $\mathbf{0}$ to denote the all-one and all-zero vectors, respectively.

First, divide $[M]$ into $d$ groups, each with $M/d$ clients. Let the $s$-th group be $G_s = \{(s-1)M/d + 1, \ldots, sM/d\}$, where $s \in \{1, 2, \ldots, d\}$. We thus have $[M] = \cup_{s=1}^d G_s$. Consider the case where $x_{i,a} = e_s$ for all $i \in G_s$ and $a \in [K]$, where $e_s$ is the $s$-th canonical basis vector.

We first choose $\theta = \{\Delta\mathbf{1}, \mathbf{0}, \ldots, \mathbf{0}\}$, i.e. $\theta_1 = (\Delta, \Delta, \ldots, \Delta)^{\mathsf{T}}$, and $\theta_a = (0, 0, \ldots, 0)^{\mathsf{T}}$ for any $a \geq 2$. ($\Delta$ will be specified later.) Then, the reward for client $i$ pulling arm $a_t$ at time $t$ would be $y_{i,t} := x_{i,a_t}^{\mathsf{T}} \theta_{a_t} + \eta_{i,t}$, where $\eta_{i,t}$ is independently drawn from a standard Gaussian distribution.

Consider a collinearly-dependent policy $\pi$. Denote $\mathbb{P}_\theta$ as the probability measure induced by $\theta = \{\theta_a\}_{a \in [K]}$ under policy $\pi$, and $\mathbb{E}_\theta$ as the corresponding expectation. Let $T_{i,a}$ be the number of times that client $i$ pulls arm $a$ within time horizon $T$. Note that $\sum_{a \in [K]} T_{i,a} = T$.

Define $\bar{T}_{s,a} := \sum_{i \in G_s} T_{i,a}$, i.e., the total number of times that arm $a$ has been pulled by the clients in group $s$ over time horizon $T$. Then, we have

$$\sum_{a \in [K]} \mathbb{E}_\theta[\bar{T}_{s,a}] = \frac{MT}{d}.$$

Thus, there must exist an arm $a_s^* > 1$ for each group $s$ such that $\mathbb{E}_\theta[\bar{T}_{s,a_s^*}] \leq \frac{MT}{d(K-1)}$.

Define a new instance $\tilde{\theta} = \{\tilde{\theta}_a\}_{a \in [K]}$, where

$$\tilde{\theta}_a = \theta_a + \sum_{s=1}^d 2\Delta e_{a_s^*} \mathbb{1}\{a_s^* = a\}, \quad \forall a \in [K].$$

Therefore, while arm 1 being optimal for all clients under instance $\theta$, the optimal arm of clients in group $G_s$ is arm $a_s^*$ under instance $\tilde{\theta}$.

Before we proceed, we introduce the following notations. Let $\mathcal{H}_t := \{a_{i,\tau}, y_{i,\tau}\}_{i \in [M], \tau \in [t]}$, $\mathcal{H}_{s,t} := \{a_{i,\tau}, y_{i,\tau}\}_{i \in G_s, \tau \in [t]}$, and $\mathcal{G}_t := \{a_{i,t}, y_{i,t}\}_{i \in [M]}$, $\mathcal{G}_{s,t} := \{a_{i,t}, y_{i,t}\}_{i \in G_s}$.

Then, we prove a useful lemma.

**Lemma 19** *Denote $D(P\|Q) := \int \log \frac{dP}{dQ} dP$ as the KL divergence between two distributions $P$ and $Q$, and $\mathcal{N}(\mu, \sigma^2)$ as a Gaussian distribution with mean $\mu$ and variance $\sigma^2$. Then,*

$$D(\mathbb{P}_\theta(\mathcal{H}_{s,T})\|\mathbb{P}_{\tilde{\theta}}(\mathcal{H}_{s,T})) = \mathbb{E}_\theta[\bar{T}_{s,a_s^*}] \cdot D(\mathcal{N}(0,1)\|\mathcal{N}(2\Delta, 1)).$$

*Proof.* Based on the definition of KL divergence, we have

$$D(\mathbb{P}_\theta(\mathcal{H}_{s,T})\|\mathbb{P}_{\tilde{\theta}}(\mathcal{H}_{s,T}))$$

$$= \mathbb{E}_\theta\left[\log \frac{\mathbb{P}_\theta(\mathcal{H}_{s,T})}{\mathbb{P}_{\tilde{\theta}}(\mathcal{H}_{s,T})}\right] \tag{165}$$

$$= \mathbb{E}_\theta\left[\log \frac{\int_{i \notin G_s} \mathbb{P}_\theta(\mathcal{H}_T)}{\int_{i \notin G_s} \mathbb{P}_{\tilde{\theta}}(\mathcal{H}_T)}\right] \tag{166}$$

$$= \mathbb{E}_\theta \left[ \log \frac{\int_{i \notin G_s} \mathbb{P}_\theta(\mathcal{H}_{T-1}) \mathbb{P}_\theta(\mathcal{G}_T | \mathcal{H}_{T-1})}{\int_{i \notin G_s} \mathbb{P}_{\tilde{\theta}}(\mathcal{H}_{T-1}) \mathbb{P}_{\tilde{\theta}}(\mathcal{G}_T | \mathcal{H}_{T-1})} \right] \tag{167}$$

$$= \mathbb{E}_\theta \left[ \log \frac{\int_{i \notin G_s} \mathbb{P}_\theta(\mathcal{H}_{T-1}) \prod_r \mathbb{P}_\theta(\mathcal{G}_{r,T} | \mathcal{H}_{r,T-1})}{\int_{i \notin G_s} \mathbb{P}_{\tilde{\theta}}(\mathcal{H}_{T-1}) \prod_r \mathbb{P}_{\tilde{\theta}}(\mathcal{G}_{s,T} | \mathcal{H}_{r,T-1})} \right] \tag{168}$$

$$= \mathbb{E}_\theta \left[ \log \frac{\int_{i \notin G_s} \mathbb{P}_\theta(\mathcal{H}_{T-1}) \prod_{r \neq s} \mathbb{P}_\theta(\mathcal{G}_{r,T} | \mathcal{H}_{r,T-1})}{\int_{i \notin G_s} \mathbb{P}_{\tilde{\theta}}(\mathcal{H}_{T-1}) \prod_{r \neq s} \mathbb{P}_{\tilde{\theta}}(\mathcal{G}_{r,T} | \mathcal{H}_{r,T-1})} + \log \frac{\mathbb{P}_{\tilde{\theta}}(\mathcal{G}_{s,T} | \mathcal{H}_{s,T-1})}{\mathbb{P}_{\tilde{\theta}}(\mathcal{G}_{s,T} | \mathcal{H}_{s,T-1})} \right] \tag{169}$$

$$= \mathbb{E}_\theta \left[ \log \frac{\int_{i \notin G_s} \mathbb{P}_\theta(\mathcal{H}_{T-1})}{\int_{i \notin G_s} \mathbb{P}_{\tilde{\theta}}(\mathcal{H}_{T-1})} \right] + \mathbb{E}_\theta \left[ \sum_{i \in G_s} \mathbb{1}\{a_{i,T} = a_s^*\} D(\mathcal{N}(0,1) \| \mathcal{N}(2\Delta, 1)) \right], \tag{170}$$

where in (165), we integrate over all variables that are associated with clients outside group $s$ to obtain the marginal distributions $\mathbb{P}_\theta(\mathcal{H}_{s,T})$ and $\mathbb{P}_{\tilde{\theta}}(\mathcal{H}_{s,T})$, respectively. Based on the assumption that $\pi$ is collinearly-independent, we can decompose the conditional distributions $\mathbb{P}_{\tilde{\theta}}(\mathcal{G}_T | \mathcal{H}_{T-1})$ and $\mathbb{P}_\theta(\mathcal{G}_T | \mathcal{H}_{T-1})$ into products of conditional distributions where only variables associated with clients in each group are dependent. Noticing that within group $s$, $\mathbb{P}_\theta$ and $\mathbb{P}_{\tilde{\theta}}$ only differs over arm $a_s^*$, based on which (170) is obtained.

Express the first term in (170) recursively, we eventually have

$$D(\mathbb{P}_\theta(\mathcal{H}_{s,T}) \| \mathbb{P}_{\tilde{\theta}}(\mathcal{H}_{s,T})) = \sum_{t \in [T]} \mathbb{E}_\theta \left[ \sum_{i \in G_s} \mathbb{1}\{a_{i,t} = a_s^*\} D(\mathcal{N}(0,1) \| \mathcal{N}(2\Delta, 1)) \right] \tag{171}$$

$$= \mathbb{E}_\theta[\bar{T}_{s,a_s^*}] \cdot D(\mathcal{N}(0,1) \| \mathcal{N}(2\Delta, 1)). \tag{172}$$

∎

Now we are ready to bound the regrets under policy $\pi$ and two instances $\theta$ and $\tilde{\theta}$. Since the selection of any sub-optimal arm will incur a regret of $\Delta$, we have

$$R(T; \pi, \theta) + R(T; \pi, \tilde{\theta}) \geq \Delta \frac{MT}{2d} \sum_{s=1}^d \left( \mathbb{P}_\theta \left[ \bar{T}_{s,1} \leq \frac{MT}{2d} \right] + \mathbb{P}_{\tilde{\theta}} \left[ \bar{T}_{s,1} > \frac{MT}{2d} \right] \right) \tag{173}$$

$$\geq \Delta \frac{MT}{4d} \sum_{s=1}^d \exp(-D(\mathbb{P}_\theta(\mathcal{H}_{s,T}) \| \mathbb{P}_{\tilde{\theta}}(\mathcal{H}_{s,T}))) \tag{174}$$

$$= \Delta \frac{MT}{4d} \sum_{s=1}^d \exp\left( -\mathbb{E}_\theta[\bar{T}_{s,a_s^*}] \cdot D(\mathcal{N}(0,1) \| \mathcal{N}(2\Delta, 1)) \right) \tag{175}$$

$$\geq \Delta \frac{MT}{4d} \sum_{s=1}^d \exp\left( \frac{MT}{d(K-1)} \cdot 2\Delta^2 \right) \tag{176}$$

$$= \frac{MT\Delta}{4} \exp\left( \frac{2MT\Delta^2}{d(K-1)} \right), \tag{177}$$

where (174) follows from the high probability Pinsker inequality, (175) is due to Lemma 19, and (176) follows from the definition of $\bar{T}_{s,a_s^*}$.

Let $\Delta := \sqrt{\frac{d(K-1)}{4MT}}$. Then, we have

$$\max\left( R(T; \pi, \theta) + R(T; \pi, \tilde{\theta}) \right) \geq \frac{\sqrt{d(K-1)MT}}{8\sqrt{e}} = \Omega(\sqrt{dKMT}).$$

# G  Fed-PE **for Shared Parameter Case**

## G.1  Algorithm Details

We slightly modify Fed-PE for the shared parameter case where $\theta_a = \theta, \forall a \in [K]$. Such minor tweaks, as opposed to a full-blown re-design, lead to a unified algorithmic framework that can be

flexibly applied to both disjoint and shared parameter cases. Similar to the disjoint parameter case, at the client side, each client $i$ treats $\{\theta_a\}_a$ separately, i.e., it does not aggregate rewards collected by pulling different arms to estimate the shared parameter $\theta$; Rather, at phase $p$, it will output $|\mathcal{A}_i^p|$ different estimates $\{\hat{\theta}_{i,a}\}_{a\in\mathcal{A}_i^p}$ and send them to the central server for aggregation.

The major difference occurs in the server side subroutine. Once different estimates $\{\hat{\theta}_{i,a}\}_{i,a}$ are received, the central server would aggregate them to obtain a global estimate of the shared parameter $\theta$. Specifically, the global aggregation step at the server side in (4) can be changed as follows: after obtaining

$$V_a^{p+1} \leftarrow \left( \sum_{i\in\mathcal{R}_a^p} f_{i,a}^p \frac{\hat{\theta}_{i,a}^p (\hat{\theta}_{i,a}^p)^{\mathsf{T}}}{\|\hat{\theta}_{i,a}^p\|^2} \right)^{\dagger}, \tag{178}$$

the server would calculate

$$V^{p+1} \leftarrow \left( \sum_{a\in\mathcal{A}^p} (V_a^{p+1})^{\dagger} \right)^{\dagger}, \tag{179}$$

$$\hat{\theta}^{p+1} \leftarrow V^{p+1} \left( \sum_{i\in[M]} \sum_{a\in\mathcal{A}_i^p} f_{i,a}^p \hat{\theta}_{i,a}^p \right). \tag{180}$$

Correspondingly, the multi-client G-optimal design at phase $p$ can be formulated as

$$\text{minimize } G(\pi) = \sum_{i=1}^{M} \max_{a\in\mathcal{A}_i^p} e_{i,a}^{\mathsf{T}} \left( \sum_{j\in[M]} \sum_{a\in\mathcal{A}_j^p} \pi_{j,a}^p e_{j,a} e_{j,a}^{\mathsf{T}} \right)^{\dagger} e_{i,a} \quad \text{s.t. } \pi^p \in \mathcal{C}^p. \tag{181}$$

Compared with the multi-client G-optimal design for the disjoint parameter case in Eqn. (6), the only difference is that (181) now has a double summation over both arms and clients inside $(\cdot)^{\dagger}$ in the objective function, while (6) only contains a summation over the clients. This is consistent with the different constructions of the "potential matrices" for those two cases.

Once the estimated shared parameter and potential matrix $(\hat{\theta}^p, V_a^p)$ is broadcast to the clients, they will use it to obtain estimates $\hat{r}_{i,a}^p$ and $u_{i,a}^p$, as in (2) under `Fed-PE`. The process then continues.

## G.2 Theoretical Analysis

We present the regret upper bound and analysis for the shared parameter case in this subsection.

**Theorem 8 (Complete version of Theorem 3)** *Consider time horizon $T$ that consists of $H$ phases with $f^p = cn^p$, where $c$ and $n > 1$ are fixed integers, and $n^p$ denotes the pth power of $n$. Let*

$$\alpha = \min(\sqrt{2\log(H/\delta) + d\log(ke)}, \sqrt{2\log(2MKH/\delta)}), \tag{182}$$

*where $k > 1$ is a number satisfying $kd \geq 2\log(H/\delta) + d\log(ke)$. Then, with probability at least $1 - \delta$, the regret of the adapted `Fed-PE` for the shared parameter case is upper bounded as*

$$R(T) \leq 4\alpha \frac{L}{\ell} \sqrt{dM} \left( \frac{\sqrt{n^2 - n}}{\sqrt{n} - 1} \sqrt{T} + \frac{K}{\sqrt{cn} - \sqrt{c}} \right).$$

*In particular, if $K = O(\sqrt{T})$, the regret upper bound scales in*

$$O\left( \sqrt{dMT(\log((\log T)/\delta) + \min\{d, \log MK\})} \right),$$

*and the communication cost scales in $O((KdM + d^2M)\log T)$.*

Following similar analysis as in the proof of Lemma 7 by replacing $v_{j,a}$ with $e_{j,a}$, we can show that the multi-client G-optimal design in (181) is equivalent to the following determinant optimization problem:

$$\text{maximize } F(\pi) = \log \text{Det} \left( \sum_{j\in[M]} \sum_{a\in\mathcal{A}_j^p} \pi_{j,a}^p e_{j,a} e_{j,a}^{\mathsf{T}} \right) \quad \text{s.t. } \pi^p \in \mathcal{C}^p. \tag{183}$$

and the optimal solution $(\pi^p)^*$ satisfies $G((\pi^p)^*) = \mathrm{rank}(\{e_{i,a}\}_{i \in [M], a \in \mathcal{A}_i}) \leq d$.

Note that compared with the optimal solution $\pi^*$ in Lemma 7 where $G(\pi^p) = \mathrm{rank}(\{v_{i,a}\}_{i \in [M], a \in \mathcal{A}_i}) \leq Kd$, the upper bound on $G(\pi)$ has been reduced by a factor of $K$, due to the dimension difference between $e_{i,a}$ and $v_{i,a}$. Besides, we also note that the block diagonal structure inside $\mathrm{Det}(\cdot)$ in (28) does not exist in (183) any more, which implies that the `BCA` algorithm in Algorithm 3 cannot be applied for this case. Designing alternative efficient optimization algorithms to solve (183) is one of our future steps.

With a little abuse of notation, in this section, we define

$$\alpha_2 := \sqrt{2 \log(H/\delta) + d \log(ke)} \tag{184}$$

$$\sigma_{i,a}^p := \|x_{i,a}\|_{V^p} / \ell \tag{185}$$

$$X_p := \left\| \sum_{i \in [M]} \sum_{a \in \mathcal{A}_i} \frac{e_{i,a}}{\|x_{i,a}\|} \ell \xi_{i,a}^{p-1} \right\|_{V^p}^2 \tag{186}$$

$$a_{(i,a),(j,b)} := \sqrt{f_{i,a}^{p-1}} e_{i,a}^{\mathsf{T}} \left( \sum_{k \in [M]} \sum_{a \in \mathcal{A}_k^{p-1}} f_{k,a}^{p-1} e_{k,a} e_{k,a}^{\mathsf{T}} \right)^{\dagger} e_{j,b} \sqrt{f_{j,b}^{p-1}} \tag{187}$$

$$\xi := \left( \frac{\ell \xi_{i,a}^{p-1}}{\|x_{i,a}\| \sqrt{f_{i,a}^{p-1}}} \right)_{i \in [M], a \in \mathcal{A}_i^{p-1}} \in \mathbb{R}^{\sum_{i \in [M]} |\mathcal{A}_i^{p-1}|} \tag{188}$$

$$A := (a_{(i,a),(j,b)}) \in \mathbb{R}^{\left( \sum_{i \in [M]} |\mathcal{A}_i^{p-1}| \right) \times \left( \sum_{i \in [M]} |\mathcal{A}_i^{p-1}| \right)}. \tag{189}$$

It can be verified that $X_p = \|A\xi\|^2$, $A^2 = A$, and $\mathrm{trace}(A) \leq d$.

Then, following the same definition of "bad" event $\mathcal{E}(\alpha)$ by using the new set of variables, we can show that Lemma 11 still holds by slightly modifying the analysis in Appendix D. For the probability of the bad event, we can show that $\mathbb{P}[\mathcal{E}(\alpha_1)] \leq \delta$ similarly as in the proof of Lemma 12, while $\mathbb{P}[\mathcal{E}(\alpha_2)] \leq \delta$ can be derived using the new definitions of $X_p$, $A$, $\xi$. In particular, we can show the following lemma.

**Lemma 20** *Under the adapted* `Fed-PE` *algorithm for the shared parameter case, for any $p \in [H]$, we have* $\mathbb{P}[X_p \geq \sqrt{2 \log(H/\delta) + d \log(ke)} | \mathcal{F}_{p-1}] \leq \delta/H$.

$\mathbb{P}[\mathcal{E}(\alpha_2)] \leq \delta$ then follows by applying a union bound over all phases $p \in [H]$.

Then, in order to bound the regret when the good event happens, we follow the same steps as in the proof of Lemma 14, except that we utilize the following lemma to remove the factor of $\sqrt{K}$ in the upper bound.

**Lemma 21** *Under the adapted* `Fed-PE` *algorithm for the shared parameter case, for any $p \in [H]$, we have* $\sum_{i \in [M]} \max_{a \in \mathcal{A}_i^p} \frac{x_{i,a}^{\mathsf{T}}}{\|x_{i,a}\|_2} V^p \frac{x_{i,a}}{\|x_{i,a}\|_2} \leq \frac{d}{f^{p-1}}$.

Lemma 21 is a direct consequence of the multi-agent G-optimal design for the shared parameter case, as elaborated earlier. Thus, when we put all pieces together, the regret is reduced by a factor of $\sqrt{K}$ compared with Theorem 1, while the other factors remain the same.

## H  Experiment Details and Additional Results

**Experiment Environment.** The experiments in both Section 6 and this section are carried out using a MacBook Pro with a 2.3GHz Quad-Core Intel Core i5 CPU and 8GB 2133 MHz LPDDR3 memory. The programming language is Python3 with packages Numpy, Scipy, and Pandas. All experiments are computationally light – the running time ranges from less than a minute to slightly more than 1 hour.

**Regret versus System Parameters.** In addition to the results reported in the main paper, we also plot the regret with varying $M, K$ and $d$ under `Enhanced Fed-PE` in Fig. 2. The experimental setup

remains the same as the synthetic one in Section 6, except that $T$ is set as $2^{16}$. We see that the per-client regret is proportional to the number of arms and the context dimensions, and inversely proportional to the number of clients, which is reasonable.

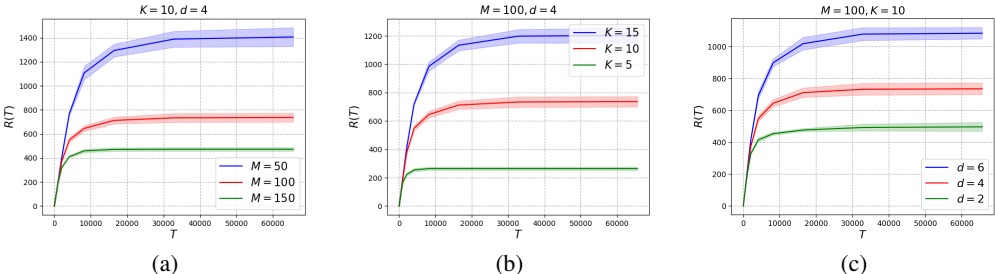

Figure 2: Per-client pseudo-regret over $T$, with varying $M, K$, and $d$. Shaded area indicates the standard deviation.

**Sparsity Level.** A sparse solution $\pi$ of the optimization problem (6) has many zero entries, which is desirable because it means each client only needs to explore a few arms in one phase even if the active arm set is very large. Define the `Sparsity Level` of $\pi$ as:

$$\texttt{Sparsity Level} = \frac{\#\text{the number of non-zero entries of } \pi}{M}.$$

The smaller the `Sparsity Level` is, the sparser the solution is. Since there are $M$ linear equality constraints, the number of non-zero entries in $\pi$ should be at least $M$. Thus, the minimum value of `Sparsity Level` is 1. Moreover, if the elimination procedure works well, we expect to eventually observe only $M$ non-zero entries in $\pi$, and constant regret under a specific realization of the bandits problem. Therefore, `Sparsity Level` also characterizes how many arms that one client pulls on average within a phase. Besides, a smaller `Sparsity Level` indicates less communication cost per client.

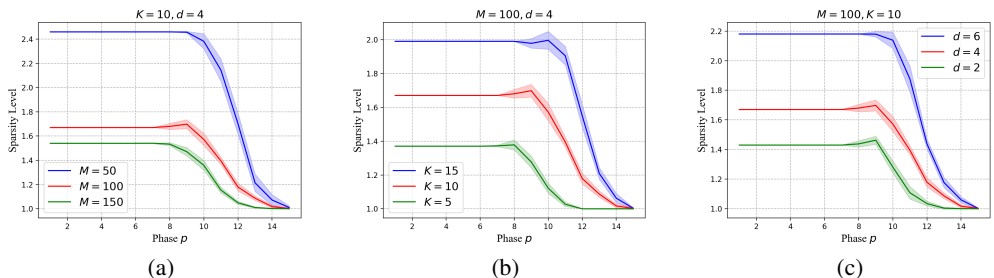

Figure 3: Sparsity level over phase index $p$, with varying $M, K$ and $d$. Shaded area indicates the standard deviation.

In Figure 3, we plot the `Sparsity Level` as a function of the phase index $p$. The result suggests that our algorithm assigns each client only approximately 2 arms (on average) to explore. We also see that the `Sparsity Level` and the communication per client are proportional to $K$ and $d$, and inversely proportional to $M$.

**Number of Iterations.** To measure the efficiency of the BCA algorithm, we define the `Number of Iterations` as the minimum number $k$ such that $G(\pi^{(k)}) \le d_a^p + \epsilon$, where the notations remain the same as in the Appendix C.2. In the experiment, we choose $\epsilon = 0.1$.

In Figure 4, we observe that the `Number of Iterations` is inversely proportional to $M$, and proportional to $K$ and $d$, which suggests that increasing the number of clients will not affect the efficiency significantly, although we should bear in mind that one iteration includes $M$ block updates.

**Different Phase Length Selection.** In Figure 5, we plot the regret performances of the three selections of $f^p$ introduced in Appendix E.2. Specifically, we construct the "greedy" phase length the

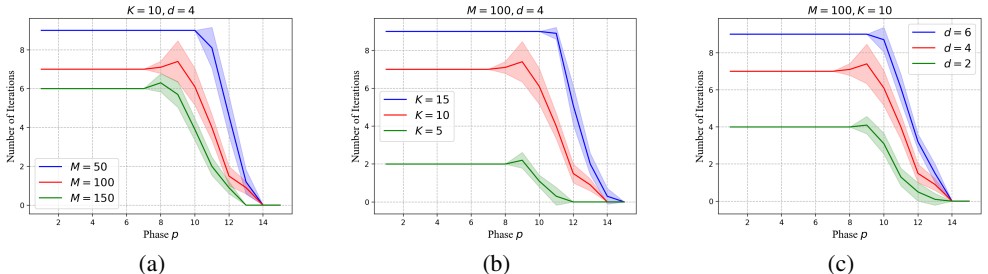

(a)                                        (b)                                        (c)

Figure 4: Number of iterations over phase index $p$, with varying $M, K$, and $d$. Shaded area indicates the standard deviation.

same as in Appendix E.2. The "exponential" selection is $f^p = 2^p$. For the "uniform" selection, we set $f^p = 100$ rather than 10 for time-saving. Note that the greedy selection leads to $\log_2 \log_2 T - 1 = 3$ phases, while the exponential selection consists of 15 phases, and uniform has 655 phases. As we observe, the uniform selection results in the lowest regret, while the exponential selection has slightly worse regret performance. The regret under the greedy selection is much worse than those under the other selections. The results suggest that exponential selection has the best tradeoff between regret and communication costs, corroborating the theoretical results in Table 2.

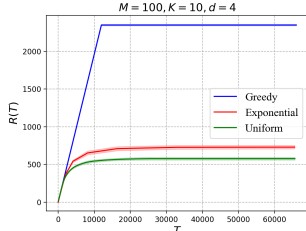

Figure 5: Regret with different selection of $f^p$. Shaded area indicates the standard deviation.