# OpenReview forum: "Federated Linear Contextual Bandits"
_NeurIPS.cc/2021/Conference — NeurIPS 2021 Poster_

### Official Review · Reviewer_LC6L · 2021-07-08

**Rating:** 7
**Confidence:** 3

**Summary:**

I think the paper sufficiently characterizes the federated linear contextual bandits problem (i) it proposes an algorithm for both the shared parameter and disjoint parameter setting (ii) proves regret guarantees and (iii) characterizes a reasonable class of policies for which a lower bound holds; this lower bound shows that the achieved regret guarantees are near optimal. The communication cost of the algorithm is also reasonable w.r.t to prior work. I think the writing of the paper can be improved in some areas and some technical points need to be explained more, but over all I think this is a good paper.

**Ethical Concerns:**

I do not have any ethical concerns regarding this paper.

**Limitations And Societal Impact:**

I think the authors have adequately addressed the limitations and societal impact.

**Main Review:**

Pros.

1. The paper characterizes the problem almost completely providing algorithms, regret upper bounds and lower bounds for a class of policies.
2. The algorithm uses the well known D-optimal design phased algorithm from Lattimore and Szepesvari, however I think several small innovations are required to make it work in the federated setting where the context vectors are different for each agent and can potentially span different subspaces of R^d. I find the discussion in section 4.1 useful.
3. It seems that the paper also develops a new block coordinate ascent algorithm to solve the multi-agent G-optimal design problem (it is a quite natural extension to G-optimal design). Unfortunately I did not check the derivation as it has been pushed to the appendix, but this also seems like a good contribution.
4. I am quite intrigued by the class of policies in the lower bound. It seems that if we stay within this class the regret bounds are almost tight. I would still need to gain more intuition for this result (see below).

Cons and Clarification needed.

1. In the introduction it would be better if the authors choose one of the applications for example the content recommendation one and maps each of the problem details like thetas, x's and rewards to real world objects in that problem. In my experience this always makes it easier to understand the relation of the problem to the applications.

2. The model is listed as one of the contributions. Is the model in the FedUCB paper different from the shared parameter setting? -- please clarify this. I know that they are interested in DP but my question is wrt the model alone.

3. It seems that because of the differential privacy constraint the FedUCB paper achieves a worse dependency of d^{3/4} but table 1 lists a sqrt{d} dependence for this paper. Am I misunderstanding the result of the FedUCB paper?

4.  I am not an expert in federated learning so this might be a naive question. In line 139 it says that the communication cost is measured in terms of number of scalars communicated. But it seems that if the size of each such scalar is unlimited one might be able to encode a lot of information in each of them using a coding scheme. Should this not be restricted?

5. I understand from context what f_p is but can you please point me to where this is first defined in the paper? I seem to have missed it.

6.  In the original G-optimal design based algorithm if the elimination criterion is in terms of optimality gap being larger than 2^{-l} in phase l. However, here it is changed to UCB < LCB kind of a condition. What is the intuition or reason behind this? Is this mentioned in the paper?

7.  I think a little more discussion is needed about the "rank-preserving" condition. To me it seems that we are just ensuring that the rank is same, however can the subspace spanned by the LHS and the RHS be completely different?

8. I am not being able to gain a lot of intuition about collinearly-dependent policies from the text description. I think it is really important to explain this part well. I am confused by the fact that in def 1 we require this to hold for just "an arm a" and also one specific subset "S". However, in the next paragraph it is claimed that "Intuitively, for
two clients that are not collinear, their local observations on ""any"" arm a cannot be utilized to improve
each other’s knowledge of their own local models". Please add more details about why this follows.

9. Each client initially starts by pulling each arm once. This might be reasonable for the disjoint parameter case. For the shared parameter setting this is not a good idea when K is very large. For instance the original G-optimal algorithm starts by pulling a set of arms of size only d(d+1)/2. I think this needs to be incorporated in the shared param setting.

**Time Spent Reviewing:**

4

---

> ### Author Response · Authors · 2021-08-10
> **Response to Reviewer LC6L**
>
> We thank the reviewer for the careful reading of our work and thoughtful comments. We present our responses in detail below.
>
> - We thank the reviewer for the suggestion of better connecting our model with practical applications. We will make the mapping more explicit in the next version. Taking the content recommendation scenario as an example, the arms correspond to different contents for recommendation, and the reward of recommending an item $a$ to a user $i$ corresponds to the user's satisfaction level (e.g., rating) on the recommendation. It is well known that user rating in such applications can be approximated as the inner product of a vector $\theta_a$, which depends on the characteristics of the content and is common for all users, and another feature vector $x_{i,a}$, which generally depends on the user's preference as well as the item.
>
> - The model considered in this paper is different from existing works. As described in Section 3, our local bandits model is essentially a single-player stochastic multi-armed bandits model where the reward of pulling an arm is an i.i.d. random variable. This is due to the assumption that each client is associated with a single fixed context, which corresponds to the profile of the local user.
>
>    Besides, to be consistent with the federated learning framework, our model also requires that only "model" updates, i.e., the estimates about the common parameters $[\theta_a]_{a\in[K]}$ instead of the raw data are communicated between the server and the clients.
>
>    The most significant differences between our model and the bandits model considered in the FedUCB paper are as follows: 1) In the latter paper, each client can see a set of varying contexts instead of a single fixed context as in our model. 2) Only model updates are communicated in our model, while FedUCB requires the clients to send (privatized) Gram matrices and reward vectors to the server. As elaborated in Section 4.1, the single fixed context assumption and the communication requirement actually make the problem more challenging, as now local estimates only contain information along the dimension of $x_{i,a}$. Such mis-alignment of the local estimates is a unique feature of our model and how to handle the challenges brought by such geometric structure is one of our major contributions.
>
> - According to Theorem 1 of Dubey and Pentland (2020), FedUCB achieves regret of $\tilde{O}(\sqrt{dMT}(\log T + Md^{1/4}\sqrt{1/\varepsilon}))$ with $(\varepsilon,\delta)$-differential privacy guarantee. When **DP is enforced** and $\varepsilon$ is sufficiently small, it renders a dependency of $d^{3/4}$. For a fair comparison with our result, we would set FedUCB to work with **no DP guarantee**, i.e., $\varepsilon\rightarrow\infty$. Then, the second term diminishes, leading to a regret bound in $\tilde{O}(\sqrt{dMT})$. We will update the table and clarify that the result is for the "no DP" setting.
>
> -  As stated in the paper, this definition of communication cost is the same as Wang et al. (2020). Each one of the communicated scalars (integer or real number) represents an individual piece of information. For example, in the upload phase each client sends an estimate of the projection of $\theta_a$, which is a $d$-dimensional vector containing $d$ scalars. This is a meaningful first-order measure of communication cost. Just as in federated learning, it is possible to design advanced quantization or coding schemes to reduce the communication load and improve its efficiency (e.g., for the server broadcast communication), but such refinement is out of the scope of the current paper.
>
> - $f^p$ is related to the length of phase $p$. Specifically, the length of phase $p$ is $f^p+K$, where the specific forms of $f^p$ are provided in the complete version of Theorem 1 (Theorem 5) in Appendix D and Appendix E. We will clarify the definition of $f^p$ in the revised version of the paper.
>
> - The original G-optimal design involves a single player, and the arm elimination criterion admits a simple form in term of the optimality gap being larger than $2^{-l}$; however, in our problem, we have a multi-client version of G-optimal design, where the optimality gap for individual clients are actually different, as a result of the heterogeneous feature vectors. Therefore, by changing the criterion to be in the form of UCB $<$ LCB, it naturally accommodates such heterogeneity in the optimality gaps and provides strong probability guarantees of the correct arm elimination. We will explain this in the next version of the paper.
>
> - We note that the subspace spanned by the LHS of the "rank-preserving" condition is always a {subset} of that spanned by the RHS. Thus, once the rank is preserved, the subspaces spanned by the LHS and the RHS are same. We will explain this in the revision.
>
> - Definition 1 contains two parts. The first part defines the collinear relationship between two clients: as long as there exists an arm satisfying the described conditions, the two clients are collinear. The second part defines collinearly-dependent policies: only when two clients are collinear, their decisions may rely on each other's local observations and become dependent; i.e., if two clients are **not** collinear, their decisions **must be** independent with each other under collinearly-dependent policies. The intuition for collinearly-dependent policies is that, for two clients that are **not** collinear, we could not find any arm such that the local observations obtained by pulling the arm can be utilized to improve the other client's estimates of her local rewards. Thus, their decision making process should be independent with each other.  We will elaborate this in the revision.
>
> - In the original G-optimal design, it only involves a single player, which admits a sparse solution involving only approximately $d(d+1)/2$ arms. However, in the federated multi-client setting considered in our work,  we have not been able to guarantee the existence of such sparse solutions. In fact, it is not even clear whether such sparsity enabled efficient arm-pulling strategy always exists in our setting. Exploiting such additional structure of the optimal solution to improve the sampling and communication efficiency is one of our future steps.

---

> > ### Comment · Reviewer_LC6L · 2021-08-18
> > **Thanks for the response**
> >
> > Thanks for the response. I understand some of the points better now. I had given a score of 7 and I will keep that score.

---

### Official Review · Reviewer_MvqL · 2021-07-15

**Rating:** 6
**Confidence:** 4

**Summary:**

This paper discusses the federated multi-armed bandit problem with linear reward functions. The formulation utilizes a linear structure with agent-dependent contexts to model differences between agent reward functions. The authors propose an algorithm based on a multi-agent G-optimal design with competitive theoretical performance under heterogeneity. The authors additionally establish minimax rates under a reasonable collinearity assumption. The algorithms FedPE and Enhanced FedPE additionally provide competitive experimental performance on both simulated and real-world benchmarks. In summary, the paper is a good contribution, however, there are some issues that I would like to see addressed as well.

**Limitations And Societal Impact:**

One limitation that the authors have to be clear about is that while this setting is indeed linear, it assumes a multi-armed environment with a fixed set of arms, which makes it inapplicable for many real-world federated learning settings such as federated recommendation systems, where the set of contexts changes with time and G-optimal design is not applicable. The authors should address this in their paper.

I do not foresee any direct societal consequences from this paper, but the authors do not discuss this either.

**Main Review:**

Originality: I believe the primary novelty of the paper is to present a linear formulation of heterogeneity across agents for the federated multi-armed bandit problem and then proposing a variant of G-optimal design. Currently, the paper is positioned (and compared against) federated contextual bandit algorithms (where the decision set changes with each iteration), whereas IMHO the paper is more suitable as a context-free formulation that essentially uses this linear formulation to account for heterogeneity within the different agents. The subsequent contributions (algorithm design, minimax rates with collinearity) all follow more smoothly from a conceptual POV if viewed from this lens, instead of a "contextual" premise. I have discussed the "contextual" aspect more in the "Issues/Concerns" section below as well.

Given the above consideration, I believe the paper has a good set of contributions from an algorithmic perspective with accompanying theoretical guarantees (based on a skim of the appendix, the proof technique is primarily a modification of the existing G-optimal route). I believe that a log T-style regret bound can also be derived by assuming a minimum gap of $\Delta$ between the rewards of the best and second-best arms for all agents, allowing us to compare more directly with multi-armed algorithms. Furthermore, the current analysis provides a worst-case regret bound, which incurs a factor of $O(\sqrt{K})$ regardless of the parameters $\theta$, but it would be interesting if the authors proposed a hybrid "smooth" model, where one can interpolate between "shared" parameters and distinct parameters with an interpolating regret bound.

Clarity: The paper is well-written and easy to understand, although I would recommend proofreading to iron out typos and some grammatical issues (minor).

Significance: There is a lot of interest in the federated/distributed bandit case, and this paper will be relevant to that sub-community.

Issues/Concerns:
- "Contextual" bandits setting: The authors incorrectly describe their setting as "contextual" whereas their setting is clearly the stochastic linear bandit setting. This is not "contextual" as the set of contexts are fixed and do not vary with time (Lattimore and Szepesvari, 2020, Chapter 20 clearly makes this distinction by calling it "Stochastic Linear Bandits" instead of "contextual"). This may seem like a nitpick, but it makes the setting considered in the paper much clearer.

- Comparison with Dubey and Pentland (2020): The regret bound presented in the former paper (Theorem 1 and Corollary 1) involves an $\mathcal O(M\sqrt{T}/\varepsilon)$ term that arises due to the $(\varepsilon, \delta)$-DP privacy guarantee, and not because of inefficiency in communication. A fair comparison would therefore be to set their FedUCB algorithm to work with no DP guarantee, i.e., $\varepsilon \rightarrow \infty$, in which case the bound is $\widetilde{\mathcal O}(\sqrt{dMT})$. Furthermore, I do not see any comparisons with FedUCB in the experiments?

- Multi-armed bandit baselines/literature review: I feel that the federated (context-free) bandit literature is also relevant as both baselines (say federated UCB1 without contexts) and part of the related work.


**Time Spent Reviewing:**

7

---

> ### Author Response · Authors · 2021-08-10
> **Response to Reviewer MvqL**
>
> We thank the reviewer for the careful reading of our work and the insightful comments. Below we present our responses.
>
> - We thank the reviewer for the insightful discussion of the "contextual" bandits setting.  To our understanding, contextual bandits refers to the setting where the reward of pulling the same arm depends on the context. Conventional contextual bandits is defined with respect to a single player, where the time-varying context can be interpreted as different incoming user profiles. In contrast, we consider a multi-client model, where each client is associated with a fixed user profile. The variation of contexts is captured over clients as opposed to over time.  Although the set of clients remains fixed through the learning process, the reward of pulling the same arm still **varies across clients**, and that is consistent with the contextual bandits philosophy. That's why we cast the problem in the contextual bandits framework. Actually, for the disjoint parameter setting where the arm parameters $\theta_a$ are different for different $a$, **the local bandits model is not a conventional stochastic linear bandits**. Thus we feel that calling the model as "federated linear bandits" may be a bit misleading for this setting. We will clarify our model to avoid any confusion in the revision.
>
> - We thank the reviewer for the suggestion of having a $\log T$-style gap-dependent bound. In this paper, we have focused exclusively on analyzing the gap-independent regret bound and comparing against other state-of-the-art regret bounds as shown in Table 1. Switching to gap-dependent regret bound is conceptually possible but would require different treatment in the analysis.
>
>    We also thank the reviewer for suggesting the "hybrid" model. It would be an interesting direction to present a unified framework to capture both shared and disjoint parameter settings and the smooth transition in between. Intuitively, this interpolation would enable us to replace the $\sqrt{Kd}$ factor in Theorem 1 with a "smoothness"-dependent factor that varies in between $\sqrt{d}$ and $\sqrt{Kd}$. We believe our algorithm can be modified to handle this hybrid model and we will rigorously analyze its regret bound in the future work.
>
> - We thank the reviewer for the suggested comparison with Dubey and Pentland (2020). We agree that a fair comparison would be to set $\varepsilon\rightarrow\infty$. We will update the results in Table 1 in the revision. Besides, we will adapt FedUCB and compare with it in the experiments as well.
>
> - To the best of our knowledge, we have included all federated (context-free) bandits literature in the related work, but we are open to suggestions if we miss any. For the baseline comparison in the experiment, we have included a context-free single-player method (local UCB) for the disjoint parameter case. However, a fair comparison with context-free federated bandit algorithms would require similar information exchange protocols between clients and the server, which is difficult to unify for context-free and context-based setups.
>
> - We will clarify the limitations of our current model in the revision, and investigate possible extensions to  handle time-varying clients/arms in the future. We note that the G-optimal design naturally allows the set of clients/arms to vary due to arm elimination, which makes such extension possible.

---

> > ### Comment · Reviewer_MvqL · 2021-08-23
> > **Thanks for the reponse**
> >
> > Thank you to the authors for responding to my concerns. I have decided to keep my score at 6.

---

### Official Review · Reviewer_N57f · 2021-07-18

**Rating:** 5
**Confidence:** 3

**Summary:**

The paper studies a federated setup where each client faces a stochastic contextual bandit.  The parameters of the bandits are coupled across the clientsm. The clients share their local estimates with the server which aggregate them and share back with the clients. The client are hetogenous and the server leverages the geomereic structure of the linear rewars to facilitates the clients learn the optimal arms.  Authors propose a collaborative algorithm called Fed-PE that do not require the nodes to exchange the local feature vectors thus preserving the privacy of the nodes. The authors consider the cases where the paramers are disjoint across the arms or coupled. The performance of the algorithms are valued on both synthetic and real datasets.

**Limitations And Societal Impact:**

Yes

**Main Review:**

The problem of contextual bandits in a federated setup is interesting. The authors' attempt to model this problem and analysis is appreciable.
However, the way the problem is motivated and it is modeled is not consistent.

--In the motivational applications of personalized content recommendation and personalized online education, the contexts are dervied from the user profile, and for different users, the context can be different. But in the model, contexts are fixed. Should not the context be allowed to change in each round?

--Is $f^p$ input to the algorithm? How is it set?

--Line 198:  It is unlcear why $\overline{e}_{i,a}$ in range$(x_{i,a})$

----

Post rebuttal:

The authors said "we consider a multi-client model, where each client is associated with a fixed user profile. The variation of contexts is captured over clients as opposed to overtime. we consider a multi-client model, where each client is associated with a fixed user profile."

This part of the motivation to model is not satisfying. If there is one learner for each profile, one has to consider a large number of learners as the number of profiles can be arbitrarily large. Also, in practice, I do not anticipate there will be one learner for each profile in applications like personalized content recommendations. The whole point of learning in personalized content recommendations is to figure out optimal action for (possibly) unseen profiles. This motivation of one learner per profile does not sound unsatisfying.

I keep my score unchanged.

**Time Spent Reviewing:**

6

---

> ### Author Response · Authors · 2021-08-10
> **Response to Reviewer N57f**
>
> We thank the reviewer for the detailed comments. Please find our clarifications below.
>
> - As the reviewer correctly noted, we do assume that the contexts are derived from the user profile, and vary for different users but not over time. Conventional contextual bandits is defined with respect to a single player, where the time-varying context can be interpreted as different incoming user profiles. In contrast, we consider a multi-client model, where each client is associated with a fixed user profile. The variation of contexts is captured over clients as opposed to over time.  Although the set of clients remains fixed through the learning process, the reward of pulling the same arm still **varies across clients**, and that is consistent with the contextual bandits philosophy. We will clarify this in the revision.
>
> - $f^p$ is configurable parameter of the algorithm, and is related to the length of phase $p$. Specifically, the length of phase $p$ is $f^p+K$, where the specific forms of $f^p$ are provided in the complete version of Theorem 1 (Theorem 5) in Appendix D and Appendix E. We will clarify the definition of $f^p$ in the revised version of the paper.
>
> - Regarding your Line 198 question, this is because the local estimate hat($\theta_{i,a}$) lies in the range of $x_{i,a}$, according to Eqn. (3) in Algorithm 1. Thus, the normalized version of the local estimate, overline($e_{i,a}$), lies in the range of $x_{i,a}$ as well.

---

> > ### Comment · Reviewer_ZDt5 · 2021-08-20
> > **Thanks for your response**
> >
> > I thank the authors for their response. I have decided to keep my score unchanged.

---

### Official Review · Reviewer_ZDt5 · 2021-07-22

**Rating:** 7
**Confidence:** 3

**Summary:**

This paper considers a linear contextual bandits problem in a federated setting where different users have different contexts but the rewards are generated through common global parameters. The goal is to minimise the regret for each user while sharing the exploration across users. The paper proposes an algorithm termed Fed-PE with regret bounded as O(\sqrt{dMT}), where d is the dimension of contexts, M is the number of users and T is the time-horizon. A matching lower bound is also presented showing that this regret is tight. The main challenge in this setting is to \emph{balance} the exploration across users as contexts might be very diverse. In order to address this challenge the paper proposes a new G-optimal design which optimises for balanced exploration across users.

**Limitations And Societal Impact:**

The paper argues that their algorithm is "private to some extent" as the context vectors are not sent to server, but I am not quite satisfied with this claim as a simple transformation of the context vectors is anyways sent to the server.

**Main Review:**

Even though there is a burgeoning literature on federated/distributed settings for linear bandits, the setting considered in this paper is new, interesting and well motivated. I think the technical contributions of this paper are also sound. I like the idea of G-optimal design and its integration with a policy that only updates O(log T) times.

Additional comments:

1. Arm Elimination: The current algorithm successively eliminates arms over time for each user. Is it possible to design an algorithm in this setting that does not eliminate arms over time similar to the UCB algorithm for non-contextual bandits?

2. Shared contexts across users: The current algorithm does not take advantage of the fact that some contexts across users might be similar. For example, there might be just m different "generic types" of users and each user might be from one of these types. In this case the dependence on M may be replaced with the dependence on m. Is it possible to do this in a more data dependent way, i.e. by designing an algorithm that automatically detects shared contexts?



**Time Spent Reviewing:**

5

---

> ### Author Response · Authors · 2021-08-10
> **Response to Reviewer ZDt5**
>
> We would like to thank the reviewer for the thoughtful comments and valuable suggestions. Below, we present our responses.
>
> - We would like to point out the the UCB type of approach has been adopted in the FedUCB algorithm in Dubey and Pentland (2020). Although FedUCB considers additional differential privacy constraint, it would also work when differential privacy is not imposed. However, as pointed out in our response to reviewer LC6L, there still exist non-trivial differences between our model and the model considered in Dubey and Pentland (2020). Therefore, how to develop a UCB type of algorithm in our setting while approaching the optimal scaling of the regret is one of our future steps.
>
> - We note that the regret bound is with respect to the total regret for $M$ clients, and it is unlikely that the scaling in $M$ can be further improved. To see this, consider the extreme case where all clients see the same local MAB instance, i.e., all clients belong to the same "generic type", and $m=1$. If we allow perfect communication after each synchronous arm pulling of the clients, the problem becomes a batched multi-armed bandits problem where each batch contains $M$ arm pulling. According to Gao et al. (2019), the lower bound for batched MAB scales in $\Omega(\sqrt{KMT})$. Therefore, we believe the dependence on $M$ is fundamental.
>
>     With that being said, we do agree with the reviewer that leveraging similarity among users may lead to improved scaling on the overall regret. Actually, our algorithm can automatically leverage the geometric structure of the local estimates (feature vectors) to coordinate the actions of users and ensure near-optimal regret performance. In other words, the potential gain from the similarity in user contexts can be leveraged by the G-optimal design adaptively. This can be manifested from Property 3) of the G-optimal design in Lemma 6, i.e.,  $G(\pi^*) = \sum_{a\in[K]} d_a$, where $d_a$ is the the dimension of the subspace spanned by the feature vectors $\{x_{i,a}\}$. Thus if there are $m$ "generic types" of users, $d_a$ is upper bounded by $\min(\{m,d\})$, and $G(\pi^*)\leq K\min(\{m,d\})$. We will then need to replace the factor of $\sqrt{dK}$ in the regret bounds by $\sqrt{K\min(m,d)}$.
>
>
> - We agree with the reviewer that although the exact context vectors are not sent to the server, its directional information is revealed to the server. As noted in Section 7, how to obfuscate the directional information without significantly impacting the learning regret is one of our future steps.
>
>
>   We do want to emphasize that the raw observations of the rewards are never transmitted, thus ensuring the privacy of such data. Besides, knowing the directional information is far from revealing private user preference. This is because the **magnitude** information of the context vectors, which is hidden from the server, is highly correlated with users' personal preferences. For example, $x_{i,a}$, $\frac{1}{4}x_{i,a}$ and $-x_{i,a}$ represent very different personal preference information. However, under our algorithm, the server can only obtain the same directional information of those feature vectors but not the exact magnitudes, thus could not infer the user preferences. That's why we think it protects the privacy of such context vectors to certain extent.

---

### Decision · Program_Chairs · 2021-09-27

**Decision:**

Accept (Poster)

**Comment:**

During discussion, all reviewers agreed that though the motivation for the precise setting is unclear, the paper makes good technical contributions and so I am recommending the paper for acceptance. However, I ask the authors to incorporate a discussion on the assumptions of each client serving one user and possible extension to potential new users entering the system.